# Variational Online Mirror Descent for Robust Learning in Schrödinger Bridge

## Abstract

Schrödinger bridge (SB) has evolved into a universal class of probabilistic generative models. In practice, however, estimated learning signals are often uncertain, and the reliability promised by existing methods is often based on speculative optimal case scenarios. Recent studies regarding the Sinkhorn algorithm through mirror descent (MD) have gained attention, revealing geometric insights into solution acquisition of the SB problems. In this paper, we propose a variational online MD (OMD) framework for the SB problems, which provides further stability to SB solvers. We formally prove convergence and a regret bound for the novel OMD formulation of SB acquisition. As a result, we propose a simulation-free SB algorithm called Variational Mirrored Schrödinger Bridge (VMSB) by utilizing the Wasserstein-Fisher-Rao geometry of the Gaussian mixture parameterization for Schrödinger potentials. Based on the Wasserstein gradient flow theory, the algorithm offers tractable learning dynamics that precisely approximate each OMD step. In experiments, we validate the performance of the proposed VMSB algorithm across an extensive suite of benchmarks. VMSB consistently outperforms contemporary SB solvers on a range of SB problems, demonstrating the robustness predicted by our theory.

## 1 Introduction

Schrödinger bridge (SB; Schrödinger, 1932) has emerged as a universal class of probabilistic generative models. Nevertheless, learning methods of SB remain somewhat *atypical*, each requiring a sophisticated approach to derive a solution. This excessive algorithmic specialization in existing methods arises from the absence of a unified perspective in statistical learning theory, often causing SB studies to rest on idealized or even wishful assumptions. However, since the Schrödinger bridge problem (SBP) is fundamentally an infinite-dimensional distributional problem—where estimated optimal probabilistic models inherently exhibit approximation errors—establishing robust theoretical guarantees is both essential and challenging. This work aims to address the SBP in a learning theoretical direction by emphasizing *robustness*, a critical property ensuring stability as well as reliability of solution against uncertainty and distributional shifts (Xu et al., 2008; Duchi & Namkoong, 2021). The collective perspective of considering the SBP as an ordinary instance of optimization opens new avenues for algorithmic advancements of probabilistic generative models, particularly within the context of the learning theory and stability improvements of probabilistic diffusion models.

One of fundamental approaches to formalizing a machine learning problem is to leverage the duality between probability distributions and model parameters. When such problems are situated within a dually-flat geometric structure induced by a Bregman divergence, mirror descent (MD; Nemirovsky & Yudin, 1983) offers a geometrically natural way to discretize the parametric updates. For parameters $\{w_t\}_{t=1}^T$ and a convex function $\Omega$, an update of MD for a cost function $F$ is derived as the following equation

$$\nabla\Omega(w_{t+1}) = \nabla\Omega(w_t) - \eta_t\nabla F(w_t), \tag{1}$$

where a gradient operation denoted as $\nabla\Omega(\cdot)$ creates a transformation that links a parametric space to a dual space. This work draws inspiration from recent research that reinterprets the Sinkhorn algorithm for training SB models as MD with respect to log-Schrödinger potentials (Peyré et al., 2019; Léger, 2021).

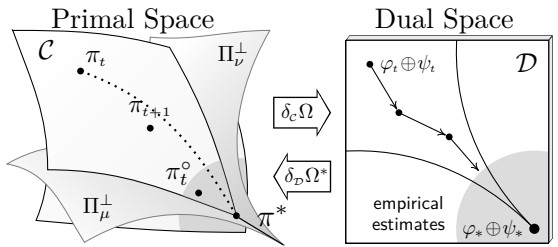

Figure 1: A schematic illustration. The primal and dual spaces $(\mathcal{C}, \mathcal{D})$ retain bidirectional maps $(\delta_\mathcal{C}\Omega, \delta_\mathcal{D}\Omega^*)$. $\Pi_\nu^\perp$ and $\Pi_\mu^\perp$ indicate projection spaces of $\gamma_1\pi = \mu$ and $\gamma_2\pi = \nu$, respectively. The current $\pi_t$ performs an online learning update following a "unreliable" leader $\pi_t^\circ$ in a region shaded in gray.

Table 1: A technical overview. Combining chracteristics of existing methods, VMSB offers a simulation-free SB solver that produces iterative OMD solutions. Our VMSB additionally provides a strong theoretical guarantee of convergence based on a regert analysis.

|  | Iterative | Simulation-free | Regret analysis |
|---|:---:|:---:|:---:|
| DSB (De Bortoli et al.) | ✓ | ✗ | ✗ |
| DSBM (Shi et al.) | ✓ | ✗ | ✗ |
| LightSB (Korotin et al.) | ✗ | ✓ | ✗ |
| LightSB-M (Gushchi et al.) | ✗ | ✓ | ✗ |
| **VMSB (ours)** | ✓ | ✓ | ✓ |

Notably, Aubin-Frankowski et al. (2022) rigorously extended the classical MD framework by reformulating it with directional derivatives and first variations, thereby establishing its concrete probabilistic foundations. Since prior analyses on Schrödinger bridge have predominantly focused on optimal case scenarios with a fixed cost function $F$, we argue that theoretical improvements are required for worst case scenarios through the use of online convex learning methods (Zinkevich, 2003), which address unknown sequences of convex cost functions $\{F_t\}_{t=1}^\infty$. Although online mirror descent (OMD; Srebro et al., 2011; Lei & Zhou, 2020) has demonstrated effectiveness in classical online learning settings, adapting OMD arguments to the SB context requires developing a novel theory and computational tools to guarantee robust algorithmic performance.

Meanwhile, bridging theory and practice necessitates a clever computation strategy of the MD update rule to overcome the non-trivial challenges posed by the distributional characteristics of optimal transport. In general, one can consider constrained distributional optimization problems with generalized gradient dynamics on the space of distributions endowed with the Wasserstein metric. Leveraging the Wasserstein gradient flow discovered by Jordan, Kinderlehrer, and Otto (JKO; Jordan et al., 1998), the desired dynamics of minimizing a cost functional $F : \mathcal{P}_2(\mathcal{X}) \to \mathbb{R}$ can be modeled, where $\mathcal{P}_2(\mathcal{X})$ denotes the set of probability distributions with finite second-order moments. Despite the extensive theoretical findings of Wasserstein gradient flows regarding OT problems (Ambrosio et al., 2005a; Santambrogio, 2015; Villani, 2021), the computational challenges remain. The established methods are commonly based on numerical methods of solving partial differential equations (PDEs; Carlier et al., 2017; Carrillo et al., 2023), whose exhaustive amount of computations make them unsuitable for systems with high dimensional probability densities.

A favored strategy to mitigate the issue is to narrow down the solution space into a subset of tractable distributions, often referred to as taking a *variational* form (Paisley et al., 2012; Blei et al., 2017). For example, mean-field formulations of SB (Liu et al., 2022; Claisse et al., 2023) are variational approximations. Unfortunately, it does not faithfully yield an analytical submanifold and is obligated to physically simulate among particles. Recently, a Gaussian mixture parameterization of the Schrödinger potentials has been proposed by Korotin et al. (2024). The simulation-free LightSB solver is simple yet general, with the guarantee of universal approximation for SB. The expressiveness of the solver coincides with geometric properties of Gaussian variational inference and mixture models (Chen et al., 2018b; Daudel et al., 2021; Lambert et al., 2022; Diao et al., 2023). However, its shortcoming—as well as other *simulation-free* solvers (Tong et al., 2024b; Gushchin et al., 2024a)—is the uncertainty of data-driven learning signals of non-convex objectives. This reveals room for improvement with the rich geometric properties of SB in a variational form.

In this paper, we explore a novel formulation of SB acquisition through the lens of dualistic geometry for ensuring further stability of learning. As illustrated in Fig. 1, we investigate primal and dual spaces $(\mathcal{C}, \mathcal{D})$. Under the entropic regularization, the transformations between coupling $\pi_t$ and dual potential $\varphi_t \oplus \psi_t$ are uniquely defined, and first variation operators $(\delta_\mathcal{C}, \delta_\mathcal{D})$ transform such elements (Aubin-Frankowski et al., 2022). For online learning, we postulate optimization errors of an SB solver, and propose a variational online mirror descent (VOMD) framework to formally reduce the errors in terms of regrets. To this end, we propose

a robust simulation-free SB algorithm called Variational Mirrored Schrödinger Bridge (VMSB). To solve SB in a robust manner, the variational method is based on gradient flows with respect to the Wasserstein-Fisher-Rao (WFR) geometry. The proposed VMSB offers a tractable approximation of mirror descent updates by formulating iterative subproblems by Wasserstein gradient flows. We show that this allows us to perform a complete form of online learning which is tolerant of unreliable empirical estimates of arbitrary data-driven SB solvers. The experiments indicate that VMSB outperforms existing methods in various benchmarks.

**Our contributions.** We aim to build a novel algorithm grounded in statistical learning theory, formally derived from the geometric perspective for SBPs. To the best of our knowledge, VMSB is the first VOMD-based SB algorithm and inherits the theoretical essence of OMD. Table 1 shows that VMSB is a simulation-free solver equipped with a rigorous convergence analysis for general learning situations of probabilistic generative models. We verify the validity of our core theoretical principles through extensive suite of SB benchmarks, including real-time online learning, standard benchmarks in optimal transport, and tasks involving the translation of one image to another. Our main contributions are summarized below:

- We develop a robust SB learning algorithm built upon the online mirror descent formulation, whose rules follow Wasserstein-2 dynamics derived from local MD objectives. Under mild assumptions, we formally prove convergence of the proposed algorithm in general online learning scenarios (§ 4).
- We introduce a simulation-free SB method leveraging the Wasserstein-Fisher-Rao geometry, ensuring asymptotic stability within Wasserstein gradient flows. The resulting VMSB algorithm admits closed-form dynamics, enabling accurate and computationally efficient implementation using LightSB (§ 5).
- We validate our algorithm across diverse SB problem settings, highlighting the effectiveness of our VOMD-based framework in contexts including online learning, classical optimal transport benchmarks, and image-to-image translation tasks. Empirical results consistently demonstrate that our proposed methods outperform existing SB solvers, strongly supporting our theoretical claims (§ 6).

## 2 Related Work

**Simulation-free SB.** The Schrödinger bridge problems are originated from a physical formulation for evolution of a dynamical system between measures (Léonard, 2012; Pavon & Wakolbinger, 1991). The study of SB has gained popularity due to its connection to entropy-regularized optimal transport (EOT; Peyré et al., 2019; Nutz, 2021). Its association with optimal transport suggests various applications across various fields related to machine learning, such as image processing, natural language processing, and control systems (Caron et al., 2020; Liu et al., 2023; Alvarez-Melis & Jaakkola, 2018; Chen et al., 2022). Historically, the most representative algorithm for SBP is Sinkhorn (Kullback, 1968), there has been progress in training SB with nonlinear networks (Vargas et al., 2021; De Bortoli et al., 2021) by "simulating" a half-bridge of forward and backward diffusion at each time. An SB solver is called as simulation-free (Tong et al., 2024a;b) if the solver is trained without samples from the simulation of SB diffusion processes. LightSB (Korotin et al., 2024) is a special type of simulation-free solver using the maximum likelihood method of Gaussian mixture models (GMMs). Building upon these advancements, our approach focuses on enhancing simulation-free SB solvers by leveraging geometric insights derived from the generalized dual geometry inherent to the SBP.

**MD and Sinkhorn.** The Bregman divergence (Bregman, 1967) is a family of statistical divergence that is particularly useful when analyzing constrained convex problems (Beck & Teboulle, 2003; Boyd & Vandenberghe, 2004; Hiriart-Urruty & Lemaréchal, 2004). Notably, Léger (2021) and Aubin-Frankowski et al. (2022) adopted the Bregman divergence into entropic optimal transport and SB problems with probability measures, and the studies revealed that Sinkhorn can be considered to be mirror descent with a constant step size $\eta \equiv 1$. In statistical geometries, the Bregman divergence is a first-order approximation of a Hessian structure (Shima & Yagi, 1997; Butnariu & Resmerita, 2006), which interprets MD as natural discretization on a gradient flow. Deb et al. (2023) introduced Wasserstein mirror flow, and the results include a geometric interpretation of Sinkhorn for unconstrained OT, *i.e.*, when $\varepsilon \to 0$ from the entropic regularization setup. Karimi et al. (2024) formulated a *half-iteration* of the Sinkhorn algorithm into a mirror flow, *i.e.*, $\eta_t \to 0$.

**Wasserstein gradient flows** have attracted considerable attention, as their intrinsic geometry is governed by the Wasserstein-2 metric. (Ambrosio et al., 2005a; Villani, 2009; Santambrogio, 2017). Otto (2001) introduced a formal Riemannian structure to interpret various evolutionary equations as gradient flows with

the Wasserstein space, which is closely related to our variational approach. The mirror Langevin dynamics is an early work describing the evolution of the Langevin diffusion (Hsieh et al., 2018), and was later incorporated in the geometry of the Bregman Wasserstein divergence (Rankin & Wong, 2023). We relate our methodology with recent approaches of variational inference on the Bures–Wasserstein space (Lambert et al., 2022; Diao et al., 2023). Utilizing Bures–Wasserstein geometry, the Wasserstein-Fisher-Rao geometry (Liero et al., 2016; Chizat et al., 2018; Liero et al., 2018; Lambert et al., 2022) additionally provides "liftings," which yield an interaction among measures.

**Learning theory.** Suppose we have time-varying costs $\{F_t\}_{t=1}^{\infty}$. We generally referred to learning through these signals as *online learning* (Fiat & Woeginger, 1998). Our interest lies in temporal costs defined in a probability space, where following the ordinary gradient may not be the best choice due to the geometric constraints (Amari, 2016; Amari & Nagaoka, 2000). In this sense, we primarily relate our work to OMD, an online learning form of mirror descent (Srebro et al., 2011; Raskutti & Mukherjee, 2015; Lei & Zhou, 2020). The OMD algorithm provides a generalization of robust learning by seeking solutions that are optimal in a worst case sense, ensuring performance guarantees under adversarial or uncertain conditions (Xu et al., 2008; Zinkevich, 2003; Madry et al., 2017). Another relevant design of the online algorithm is the follow-the-regularized-leader (FTRL; McMahan, 2011). OMD focuses on scheduling proximity of updates through $\{\eta_t\}_{t=1}^{T}$, whereas FTRL minimizes historical losses with a fixed proximity term.

## 3 Generalization of Schrödinger Bridge Problems

**Notation.** Let $\mathcal{P}(\mathcal{S})$ ($\mathcal{P}_2(\mathcal{S})$) denote the set of (absolutely continuous) Borel probability measures on $\mathcal{S} \subseteq \mathbb{R}^d$ (with a finite second moment). For marginals $\mu, \nu \in \mathcal{P}_2(\mathcal{S})$, $\Pi(\mu, \nu)$ denotes the set of couplings (Peyré et al., 2019). For a coupling $\pi$, we often use a shorthand notation $\vec{\pi}^x$ ($\overleftarrow{\pi}^y$) to denotes a conditional distribution for a sample data $\vec{\pi}(\cdot|x)$ (or $\overleftarrow{\pi}(\cdot|y)$; see Fig. 2). We use $\mathrm{KL}(\cdot\|\cdot)$ to denote the KL functional and assume $+\infty$ if an argument is not absolutely continuous. We employ a notation $\mathbb{P}([0,1], \mathcal{S})$ to denote a set of path measures from the time interval $[0,1]$.

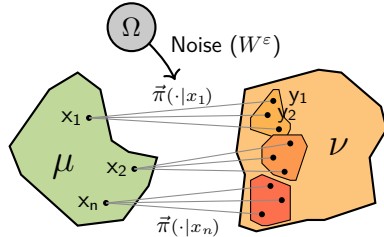

Figure 2: The SB problem.

**The static SB problem and Sinkhorn.** We first introduce a problem definition of a static variant of Schrödinger bridge problems. For a regularization coefficient $\varepsilon \in \mathbb{R}^+$, the *static* SB problem, or the entropic optimal transport problem with a quadratic cost function $c(x,y) = \frac{1}{2}\|x - y\|^2$, is defined as optimization of finding a unique minimizer of optimal coupling $\pi^*$ with repect to the following objective

$$\mathrm{OT}_\varepsilon(\mu, \nu) \coloneqq \inf_{\pi \in \Pi(\mu,\nu)} \iint_{\mathcal{S}\times\mathcal{S}} c(x,y)\, d\pi(x,y) + \varepsilon \mathrm{KL}(\pi \| \mu \otimes \nu) \tag{2}$$

where $\mu \otimes \nu$ denotes, the product of marginals and $\mathrm{KL}(\cdot\|\cdot)$ in the regularization term denotes the Kullback-Leibler (KL) functional, or relative entropy, defined as

$$\mathrm{KL}(\pi \| \mu \otimes \nu) \coloneqq \begin{cases} \iint_{\mathcal{S}\times\mathcal{S}} \log \frac{d\pi}{d(\mu \otimes \nu)}\, d\pi, & \pi \ll \mu \otimes \nu, \\ \infty, & \pi \not\ll \mu \otimes \nu. \end{cases}$$

It is well studied that the EOT problem (2) can be translated to a singular form of divergence minimization problem (Nutz, 2021), *i.e.*, $\mathrm{KL}(\pi\|\mathcal{R})$ where the reference measure $\mathcal{R}$ is of *Gibbs parameterization*, satisfying the Radon-Nikodym derivative of $\frac{d\mathcal{R}}{d(\mu \otimes \nu)} = e^{-c_\varepsilon}$ with $c_\varepsilon = c/\varepsilon$. We are particularly interested in its dual aspect, which contains a pair of *log-Schrödinger potentials* as an alternative representation for EOT solutions.

**Lemma 1** (Duality; Nutz, 2021). *Suppose existence of optimal coupling $\pi^* \in \Pi(\mu, \nu)$ and log-Schrödinger potentials $(\varphi^*, \psi^*) \in L^1(\mu) \times L^1(\nu)$. Then, the dual problem of (2) is found without gap by*

$$\inf_{\pi \in \Pi(\mu,\nu)} \mathrm{KL}(\pi\|\mathcal{R}) = \sup_{\varphi \in L^1(\mu), \psi \in L^1(\nu)} \mu(\varphi) + \nu(\psi) - \iint_{\mathcal{S}\times\mathcal{S}} e^{\varphi \oplus \psi}\, d\mathcal{R} + 1, \tag{3}$$

*where $\mu(\varphi) \coloneqq \int_{\mathcal{S}} \varphi\, d\mu$; $\nu(\psi) \coloneqq \int_{\mathcal{S}} \psi\, d\nu$; the operator $\oplus$ indicates the direct sum of two potentials; the symbol $\mathcal{R}$ denotes the reference of Gibbs measure: $d\mathcal{R} = e^{-c_\varepsilon} d(\mu \otimes \nu)$ with $c_\varepsilon(x,y) \coloneqq \frac{1}{2\varepsilon}\|x - y\|^2$.*

By the duality, the Lebesgue-integrable functions $(\varphi^*, \psi^*) \in L^1(\mu) \times L^1(\nu)$ represent dual solution of (3), associated with $\pi^*$ by $d\pi^* = e^{\varphi^* \oplus \psi^* - c_\varepsilon} d(\mu \otimes \nu)$, $(\mu \otimes \nu)$-almost surely. Apparently, the result is a general form of Legendre duality from convex optimization, and the realization that every element of the dual space can be split into two separate potential functions has inspired a *halfway* projection method known as the Sinkhorn algorithm (Cuturi, 2013). The Sinkhorn algorithm is defined as following alternating updates:

$$\psi_{2t+1}(y) = -\log \int_{\mathcal{S}} e^{\varphi_{2t}(x) - c_\varepsilon(x,y)} \mu(dx), \qquad \varphi_{2t+2}(x) = -\log \int_{\mathcal{S}} e^{\psi_{2t+1}(y) - c_\varepsilon(x,y)} \nu(dy), \qquad (4)$$

where each update is called a iterative proportional fitting (IPF; Kullback, 1968) procedure. Based on the geometric understanding of Bregman gradient descent and flow (Léger, 2021; Karimi et al., 2024), the static SB problem and Sinkhorn reveal a profound connection to the information geometry incurred by the KL functional. Of course, this implies that that the Sinkhorn algorithm is not the only way of finding the solution space. Akin to differential geometry, JKO discovered that each density can be analyzed using the standard geometry of the Wasserstein-2 distance, defined as (Jordan et al., 1998; Villani, 2009)

$$W_2(\mu, \nu) := \inf_{\pi \in \Pi(\mu, \nu)} \left( \mathbb{E}_{(x,y) \sim \pi} \|x - y\|^2 \right)^{\frac{1}{2}}, \qquad (5)$$

where $\pi$ is a joint distribution that has $\mu$ and $\nu$ as its marginals. In essence, the distance measures the minimum cost to transform the distribution $\mu$ into $\nu$, and provides an $L^2$-like metric that quantifies the discrepancy between two distributions by considering the distance that mass must travel. Our computation strategy utilizes the theory of Wasserstein gradient flows whose technical machinery is in Appendix B.

**The dynamic SB problem.** An alternative perspective on the Schrödinger bridge is the interpretation in terms of fluid dynamics. In this setup, we consider the Wiener process $W^\varepsilon$ with volatility $\varepsilon \in \mathbb{R}^+$ be the reference for Schrödinger bridge. The *dynamic* SBP (Vargas et al., 2021) aims to find a process $\mathcal{T}^*$ such that

$$\mathcal{T}^* := \underset{\mathcal{T} \in \mathcal{Q}(\mu, \nu)}{\arg\min} \, \mathrm{KL}(\mathcal{T} \| W^\varepsilon), \qquad (6)$$

where $\mathcal{Q}(\mu, \nu) \subset \mathbb{P}([0,1], \mathcal{S})$ is the set of processes with marginals $\mu$ and $\nu$. The SB process $\mathcal{T}^*$ is uniquely described by a stochastic differential equation (SDE): $dX_t = g^*(t, X_t) + dW_t^\varepsilon$ in $t \in [0,1)$, governed by a drift function $g^*$ along with some noise generated by $W_t^\varepsilon$. In the dynamic SBP, addressing $\mathcal{T}^*$ involves continuous-time log-Schrödinger potentials $\{\varphi_t\}_{t \in [0,1]}$, $\{\psi_t\}_{t \in [0,1]}$, and the time-dependent density $\rho_t$, satisfying the following differential equations for $\rho_0 = \mu$ and $\rho_1 = \nu$ (Gigli & Tamanini, 2020)

$$\begin{cases} \partial_t \varphi_t = \frac{1}{2} |\nabla \varphi_t|^2 + \frac{\varepsilon}{2} \Delta \varphi_t, \\ -\partial_t \psi_t = \frac{1}{2} |\nabla \psi_t|^2 + \frac{\varepsilon}{2} \Delta \psi_t, \end{cases} \qquad \begin{cases} -\partial_t \rho_t + \nabla \cdot (\nabla \varphi_t \rho_t) = \frac{\varepsilon}{2} \Delta \rho_t, \\ \partial_t \rho_t + \nabla \cdot (\nabla \psi_t \rho_t) = \frac{\varepsilon}{2} \Delta \rho_t, \end{cases} \qquad (7)$$

where the operators $(\nabla, \nabla\cdot, \Delta)$ denote gradient, divergence, and Laplacian, respectively. The pair of PDEs on the left are commonly known as Hamilton-Jacobi-Bellman equations; also, the pair on the right together indicates the forward and backward Fokker-Planck equations, by the fact that potential gradients reconstruct the score function $\nabla \varphi_t + \nabla \psi_t = \varepsilon \nabla \log \rho_t$. Applying the nonlinear Feynman-Kac lemma (Yong & Zhou, 1999) transforms Eq. (7) into stochastic differential equations called SB-FBSDE (Chen et al., 2022; see Appendix C). The fundamental equivalence between static and dynamic SBPs (Pavon & Wakolbinger, 1991; Léonard, 2012) allows us to consider the optimal coupling $\pi^*$ when finding the SB process $\mathcal{T}^*$, vice versa.

**The Bregman divergence.** To advance the discussion, one needs an alternative notion of gradients in order to generalize the Bregman divergence, since the topological space of SB does not ensure Gâteaux differentiability (Aubin-Frankowski et al., 2022; Definition 4 in Appendix A) In this context, we reintroduce the following definitions of *directional derivatives* (Aliprantis & Border, 2006) and *first variations* (Aubin-Frankowski et al., 2022). We also refer to Definition 7.12 of Santambrogio (2015) for alternative description.

**Definition 1** (Directional derivative). Given a locally convex topological vector space $\mathcal{M}$, the directional derivative of $F$ in the direction $\xi$ is defined as $d^+F(x; \xi) = \lim_{h \to 0^+} \frac{F(x + h\xi) - F(x)}{h}$.

**Definition 2** (First variation). Given a topological vector space $\mathcal{M}$ and a convex constraint $\mathcal{C} \subseteq \mathcal{M}$, for a function $F$ and $x \in \mathcal{C} \cup \mathrm{dom}(F)$, define the first variation of $F$ over $\mathcal{C}$ to be an element $\delta_\mathcal{C} F(x) \in \mathcal{M}^*$,

where $\mathcal{M}^*$ is the topological dual of $\mathcal{M}$, such that it holds for all $y \in \mathcal{C} \cup \text{dom}(F)$ and $v = y - x \in \mathcal{M}$: $\langle \delta_c F(x), v \rangle = d^+ F(x; v)$. $\langle \cdot, \cdot \rangle$ denotes the duality product of $\mathcal{M}$ and $\mathcal{M}^*$.

Following Karimi et al. (2024), this work considers the generalized Bregman divergence defined with a weak notion of the directional derivative. We explicitly set the Bregman potential $\Omega(\cdot) = \text{KL}(\cdot \| e^{-c_\varepsilon} \mu \otimes \nu)$ in the SB problems for the rest of the paper, which enforces the Gibbs parameterization for the OT couplings.

**Definition 3** (Bregman divergence). Let a convex functional $\Omega : \mathcal{M} \to \mathbb{R} \cup \{+\infty\}$ be a Bregman potential. Define the Bregman divergence associated with $\Omega$ as

$$D_\Omega(x\|y) \coloneqq \Omega(x) - \Omega(y) - d^+\Omega(y; x - y), \tag{8}$$

for every pair of elements in the domain $x, y \in \mathcal{M}$.

The definition preserves the essential role of the Bregman divergence as the first-order Taylor expansion of $\Omega$, allowing us to utilize properties similar to those used in the classical setting.

**Asymptotically log-concave distributions.** Lastly, our theoretical analysis works with a certain form of *measure concentration* property, and we formally address this property with asymptotically strong log-concave (alc) distributions. This section rigorously states the resulting assumption which is later used in address the desired properties of OMD, analogous to strong convexity assumption in the classical literature. Let us consider the following informal definition of asymptotically strong log-concave distributions

$$\mathcal{P}_{\text{alc}}(\mathbb{R}^d) \coloneqq \left\{ \zeta(\mathrm{d}x) = \exp\big(-U(x)\big)\mathrm{d}x \,:\, U \in C_2\big(\mathbb{R}^d\big),\ U \text{ is asymptotically strongly convex} \right\}, \tag{9}$$

where Appendix A contains a formal version on asymptotical convexity. Note that asymptotically log-concave functions satisfy a certain form of logarithmic Sobolev inequality (LSI; Gross, 1975). The condition can be an extension of Sobolev space (Adams & Fournier, 2003) for informational geometric problems. The simplest case of such condition for the Gaussian measure is represented as follows.

**Remark 1** (LSI for the standard Gaussian). Suppose that $f$ is a nonnegative function, integrable with respect to a measure $\gamma$, and that the entropy is defined as $\text{Ent}_\gamma(f) = \int_{\mathbb{R}^d} f \log f \mathrm{d}\gamma - \left(\int_{\mathbb{R}^d} f \mathrm{d}\gamma\right) \log\left(\int_{\mathbb{R}^d} f \mathrm{d}\gamma\right)$. the log Sobolev inequality when $\gamma$ is the standard Gaussian measure reads $\text{Ent}_\gamma(f) \leq \frac{1}{2} \int_{\mathbb{R}^d} \frac{|f|^2}{f} \mathrm{d}\gamma$.

Historically, the log Sobolev inequality condition arises from the implication of satisfying the Talagrand's inequality for bounding the Wasserstein-2 distance, and is closely related to measure concentration (Otto & Villani, 2000). The important extension of asymptotically strong log-concave distributions for Schrödinger bridge $d\pi = e^{\varphi \oplus \psi - c_\varepsilon} d(\mu \otimes \nu)$, $(\mu \otimes \nu)$-a.s. is that induced SB model also satisfies asymptotically strongly log-concaveness and the LSI condition (Conforti, 2024). For a representative model related to our work, the Gaussian mixture parameterization (Korotin et al., 2024) is a representative model that our theoretical analysis holds, because Gaussian mixture weights does not alter the asymptotic characteristic.

**Remark 2** (Conforti, 2024). Let $\mu, \nu \in \mathcal{P}_{\text{alc}}(\mathbb{R}^d)$ with finite entropy on Lebesgue measures and $\pi \in \mathcal{C}$ be a coupling in the static Schrödinger bridge problem. Then, for a quadratic cost function, the coupling distribution is also asymptotically log-concave and satisfies a form of logarithmic Sobolev inequality.

Let us suppose that a parameterized SB model $d\pi_t = e^{\varphi_t \oplus \psi_t - c_\varepsilon} d(\mu \otimes \nu)$ obeys the following constraints for marginals and potentials:

$$\mathcal{C} \coloneqq \left\{ \pi : (\mu, \nu) \in \mathcal{P}_2(\mathbb{R}^d) \cap \mathcal{P}_{\text{alc}}(\mathbb{R}^d),\ (\varphi, \psi) \in L^1(\mu) \times L^1(\nu),\ \text{and}\quad \varphi, \psi \in C^2(\mathbb{R}^d) \cap \text{Lip}(\mathcal{K}) \right\}, \tag{10}$$

where $\text{Lip}(\mathcal{K})$ denotes a set of functions with $\mathcal{K}$-Lipschitz continuity. Using the disintegration theorem for probability measures (Léonard, 2014), we assume the boundedness of Bregman divergence between two transport plans using derivatives of first variations with a positive constraint $\omega > 0$ by the following assumption.

**Assumption 1** (LSI for couplings). Let us suppose $\Omega = \text{KL}(\pi\|\mathcal{R})$ for a reference measure $\mathcal{R}$. Suppose that arbitrary $\pi, \bar{\pi} \in \mathcal{C}$ satisfy a type of logarithmic Sobolev inequality for relative entropy (KL divergence) is upper bounded by the relative Fisher information (Gross, 1975) for some $\bar{\omega} \in \mathbb{R}_+$ as

$$\text{KL}(\pi\|\mathcal{R}) \leq \frac{1}{2\bar{\omega}} \iint_{\mathbb{R}^d \times \mathbb{R}^d} \left| \nabla \log \frac{d\pi(x, y)}{d\mathcal{R}(x, y)} \right|^2 \pi(\mathrm{d}x, \mathrm{d}y). \tag{11}$$

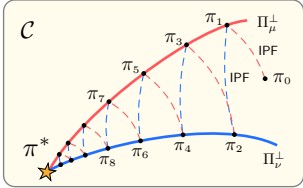 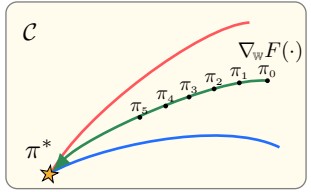 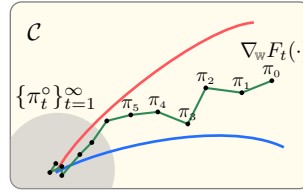

Figure 3: Learning for an SB model $\{\pi_t\}_{t=1}^{\infty}$ in the primal space $\mathcal{C}$ (see Fig. 1 for the details). Left: Sinkhorn (Lemma 2). Middle: Wasserstein gradient descent in the distributional space $\mathcal{C}$ for fixed $F$ (Lemma 3). Right: Variational online mirror descent with sequence of convex costs using uncertain estimates $\{\pi_t^{\circ}\}_{t=1}^{\infty}$.

By equivalence of the first variations of Bregman divergences (explained in Lemma 6), we obtain $\mathrm{KL}(\pi\|\mathcal{R}) = D_{\Omega}(\pi\|\bar{\pi})$. Applying the Hölder's inequality to Eq. (11), assume there exists a constant $\omega > 0$ such that

$$D_{\Omega}(\pi\|\bar{\pi}) \leq \frac{1}{2\omega}\big\|\nabla(\delta_{c}\Omega(\pi) - \delta_{c}\Omega(\bar{\pi}))\big\|_{L^2(\pi)}^2 \tag{12}$$

for the Bregman potential $\Omega = \mathrm{KL}(\cdot\|e^{-c_{\varepsilon}}\mu \otimes \nu)$ and the first variation $\delta_{c}$. We call the condition as $\mathrm{LSI}(\omega)$.

Since, as illustrated in Fig. 1, $\varphi_t \oplus \psi_t = \delta_{c}\Omega(\pi_t)$ for arbitrary $\pi_t$, the assumption geometrically enforces a quadratic upper bound on the Bregman divergence in terms of dual space gradients. In general, the LSI condition also has often been used to analyze the convergence of partial differential equations (Malrieu, 2001). To make our analysis on improvement (Lemma 14) and a solid regret bound of OMD (Lemma 16), this work finds that Assumption 1 is necessary to ensure a certain asymptotical type of measure concentration.

## 4 Learning Schrödinger Bridge via Online Mirror Descent

The goal in this section is to derive an OMD update rule for SB, and analyze its convergence. To accomplish this, we postulate on the existence of temporal estimates and an online learning problem. Our analysis suggests that applying an MD approach can reduce the uncertainty of these estimates.

### 4.1 Sinkhorn and Wasserstein descent as mirror descent algorithms

We start with a novel characterization of Sinkhorn as an online form of mirror descent as illustrated in the left side of Fig. 3, which will lead to a better understanding of the online mirror descent framework. OMD updates are determined by the first order approximation of costs $F_t$ and proximity of previous iterate with respect to a Bregman divergence (Beck & Teboulle, 2003). Using the first variation $\delta_{c}$ in Definition 2 instead of standard gradient $\nabla$, the proximal form of OMD is derived as (Karimi et al., 2024)

$$\pi_{t+1} = \underset{\pi \in \mathcal{C}}{\arg\min}\Big\{\langle \delta_{c}F_t(\pi_t), \pi - \pi_t\rangle + \tfrac{1}{\eta_t}D_{\Omega}(\pi\|\pi_t)\Big\}, \tag{13}$$

where $F_t$ denotes a temporal cost function for SB models in $\mathcal{C}$. In Eq. (13), the updates are determined by the first order approximation of $F_t$ and proximity of previous iterate $\pi_t$ with respect to the Bregman divergence (Beck & Teboulle, 2003). In contrast to the 'half-bridge' interpretation provided by Karimi et al. (2024), the online MD iteration (13) involves a temporal cost $F_t$, which offers more general reinterpretation of the Sinkhorn algorithm. Using the feasible model space $\mathcal{C}$ in (10), IPF projections (4) are reformulated as following subproblems of alternating Bregman projections:

$$\underset{\pi \in \Pi_{\mu}^{\perp}}{\arg\min}\big\{\mathrm{KL}(\pi\|\pi_{2t}) : \pi \in \mathcal{C}, \gamma_2\pi = \nu\big\}, \qquad \underset{\pi \in \Pi_{\nu}^{\perp}}{\arg\min}\big\{\mathrm{KL}(\pi\|\pi_{2t+1}) : \pi \in \mathcal{C}, \gamma_1\pi = \mu\big\}, \tag{14}$$

where $\gamma_1\pi(x) := \int \pi(x,y)\,\mathrm{d}y$ and $\gamma_2\pi(y) := \int \pi(x,y)\,\mathrm{d}x$ denotes the marginalization operations, and the symbols $(\Pi_{\mu}^{\perp}, \Pi_{\nu}^{\perp})$ denote the Sinkhorn projection spaces that preserve the property of marginals. As a generalized optimization problem in $\mathcal{C}$, one can consider a temporal cost $\widetilde{F}_t(\pi) := a_t\mathrm{KL}(\gamma_1\pi\|\mu) + (1 - a_t)\mathrm{KL}(\gamma_2\pi\|\nu)$ with sequence $\{a_t\}_{t=1}^{\infty} = \{0, 1, 0, 1, \dots\}$. By construction, we show that an online form of MD for $\widetilde{F}_t$ with a constant step size $\eta_t \equiv 1$ matches the Sinkhorn.

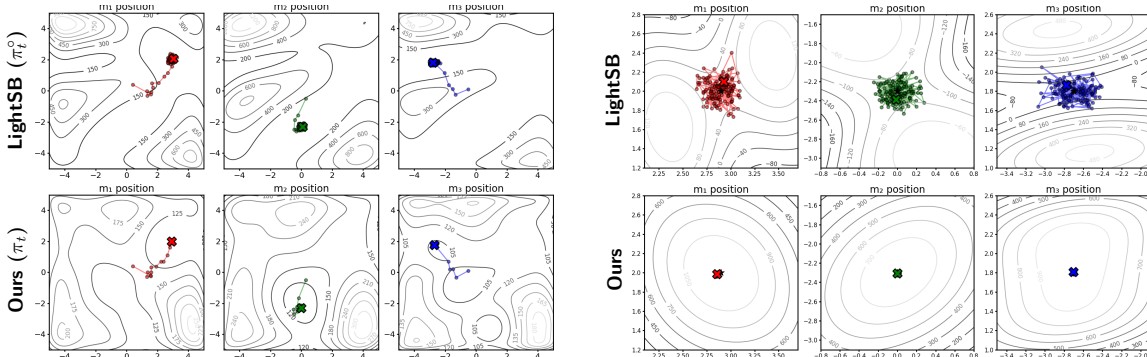

Figure 4: Loss landscapes and gradient dynamics in a 2D problem. Left: In an early stage, parameters of three modalities $\{m_k\}_{k=1}^3$ (mean estimations) for both LightSB (top) and VMSB (bottom) methods approach the optimality with different costs. Right: When magnified the landscapes in the late stages (10 times), while LightSB is vibrant, whereas our method emits strictly convex landscape and stable dynamics.

**Lemma 2** (Sinkhorn)**.** *For* $\Omega(\cdot) = \mathrm{KL}(\pi \| e^{-c_\varepsilon} \mu \otimes \nu)$, *iterates from* $\pi_{t+1} = \arg\min_{\pi \in \mathcal{C}} \big\{ \langle \delta_\mathcal{C} \widetilde{F}_t(\pi_t), \pi - \pi_t \rangle + D_\Omega(\pi \| \pi_t) \big\}$ *are equivalent to estimates from* $(\varphi_t, \psi_t)$ *of* (4), *for every update step* $t \in \mathbb{N}_0$.

The proof is in Appendix A. Consequently, we established that the Sinkhorn algorithm corresponds to an instance OMD; however, its inherent structure limits flexibility on step sizes and other underlying assumptions, making it challenging to analyze directly using standard OMD theoretical arguments. Instead, one can alternatively consider OMD by recovering a "static" objective, namely $F(\cdot) := \mathrm{KL}(\cdot \| \pi^*)$, where the KL functional is originated from the formal definition of SBP (Vargas et al., 2021; Chen et al., 2022). The following lemma shows that the MD updates for the static cost correspond to discretization of a Wasserstein gradient flow for SB models, a Riemannian steepest descent in the model space of $\mathcal{C}$.

**Lemma 3** (MD in the Wasserstein space)**.** *Suppose that* $F(\pi) := \mathrm{KL}(\pi \| \pi^*)$ *for* $\pi \in \mathcal{C}$. *The MD formulation of* $F$ *corresponds to a discretization of a geodesic flow such that* $\lim_{\eta_t \to 0^+} \frac{\pi_{t+1} - \pi_t}{\eta_t} = -\nabla_\mathbb{W} F(\pi_t)$, *where* $\nabla_\mathbb{W}$ *denotes the Wasserstein-2 gradient operator.*

According to Lemma 3, gradients of $F$ are tangential to the geodesic curve from $\pi_0$ to $\pi^*$ (green line in the middle of Fig. 3) in terms of the Wasserstein-2 metric $W_2$. Hence, the geometric interpretation allows us to consider the static cost $F$ as the ground-truth cost for optimization in our variational OMD framework. However, the ideal case falls short in practice since $\pi^*$ is inherently unknown. Therefore, we postulate on an online learning problem that nonstationary estimates $\{\pi_t^\circ\}_{t=1}^\infty$ are offered instead of $\pi^*$ as learning signals, making an optimization process with $F_t(\cdot) := \mathrm{KL}(\cdot \| \pi_t^\circ)$. As shown in in the right side of Fig. 3, we focus on the online learning setting where $\{\pi_t^\circ\}_{t=1}^\infty$ are fundamentally uncertain with perturbation, since the optimal coupling $\pi^*$ is not accessible during the training time.

## 4.2 Online mirror descent for Schrödinger bridges: theoretical analysis

From a learning theoretic standpoint, an apparent yet understated premise is that an SB algorithm does not retain the global target $\pi^*$ in practice. The global objective $F$ (also $\widehat{F}_t$) are fundamentally unknown, but are instead inferred, imposing innate uncertainty of optimization. Instead, we postulate on an online learning problem that nonstationary ergodic estimates $\{\pi_t^\circ\}_{t=1}^\infty$ are offered instead of $\pi^*$ (gray region in Fig. 1). Let $\Omega^*$ be the Fenchel conjugate of $\Omega + i_\mathcal{C}$ with the convex indicator[1] $i_\mathcal{C}$. For the space $\mathcal{D} := \delta_\mathcal{C} \Omega(\mathcal{C})$, a directional derivative $\delta_\mathcal{D}$ of $\Omega^*$ exists by the Danskin's theorem (Danskin, 1967; Bernhard & Rapaport, 1995), such that

$$\delta_\mathcal{D} \Omega^*(\varphi \oplus \psi) = \arg\max_{\pi \in \mathcal{C}} \big\{ \langle \varphi \oplus \psi, \pi \rangle - \Omega(\pi) \big\}. \tag{15}$$

Note that $(\delta_\mathcal{C} \Omega, \delta_\mathcal{D} \Omega^*)$ form bidirectional maps; a direct sum of potentials $\varphi \oplus \psi \in \mathcal{D}$ represent an element of the generalized dual space. The key assumption is that the learning target $\pi_t^\circ$ is asymptotically mean

---

[1] Defined as $i_\mathcal{C}(x) = 0$ if $x \in \mathcal{C}$ and $+\infty$ otherwise.

stationary (Gray & Kieffer, 1980) for the dual space, which have been used to analyze stochastic dynamics. Since iterates are updated through dual parameters in MD, we refer to the process as being dually stationary.

**Assumption 2** (Dually stationary process). Suppose that $\pi_{\mathcal{D}}^{\circ} \in \mathcal{C}$ exists, which is the primal representation of asymptotic dual average $\pi_{\mathcal{D}}^{\circ} := \delta_{\mathcal{D}}(\lim_{t \to \infty} \mathbb{E}_t[\delta_{\mathcal{C}} \Omega(\pi_t^{\circ})])$, where the notation $\mathbb{E}_t$ denotes the time-average.

Plugging $F_t(\cdot) = \mathrm{KL}(\cdot \| \pi_t^{\circ})$ to (13) from § 4.1, we achieve a distinct problem setup. Fig. 4 demonstrates toy experiment regarding our online learning hypothesis. OMD decomposes the global problem into local convex problems, and prevented iterates from being vibrant by stopping at a single point $\pi_{\mathcal{D}}^{\circ}$. This verifies that OMD stabilizes learning of $\pi_t$, even when the reference $\pi_t^{\circ}$ tends to inherently have some perturbation.

For stability, we state two conditions for OMD step sizes $\{\eta_t\}_{t=0}^{\infty}$, which will be justified in Theorem 1 and Proposition 1. Fig. 5 shows a plot of a well-known example: harmonic progression $\frac{1}{a+td}$ for $a \in \mathbb{R}$ and $d \in \mathbb{R}^+$ with respect to $t$. The sequence satisfies both conditions of the following assumption.

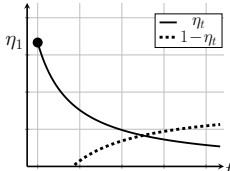

Figure 5: A sequence example of $\eta_t$ and $1 - \eta_t$.

**Assumption 3** (Step sizes). Assume two conditions for step sizes $\{\eta_t\}_{t=0}^{\infty}$. (a) *Convergent sequence & divergent series:* $\lim_{t \to \infty} \eta_t = 0$ and $\sum_{t=1}^{\infty} \eta_t = \infty$. (b) *Convergent series for squares:* $\sum_{t=1}^{\infty} \eta_t^2 < \infty$.

Using the conditions above, we firstly argue that OMD for the temporal cost $\mathrm{KL}(\cdot \| \pi_t^{\circ})$ with respect to the Bregman potential $\Omega = \mathrm{KL}(\cdot \| e^{-c_\varepsilon} \mu \otimes \nu)$ requires step size scheduling for the sake of convergence. The following theorem states that convergence in the case of $\pi^* = \pi_{\mathcal{D}}^{\circ}$ is assured when $\eta_t$ follows Assumption (3a). In contrast to the well-known linear convergence guarantees for the Sinkhorn algorithm under bounded costs with fixed marginals (Carlier, 2022), our OMD-based analysis establishes sublinear convergence rates, accommodating scenarios involving unbounded and non-stationary costs.

**Theorem 1** (Step size considerations). *Suppose the idealized case of $\pi^* = \pi_{\mathcal{D}}^{\circ}$. Then, for $\{\pi_t\}_{t_1}^T \subset \mathcal{C}$ we get $\lim_{T \to \infty} \mathbb{E}_{1:T}[D_\Omega(\pi_{\mathcal{D}}^{\circ} \| \pi_T)] = 0$ if and only if Assumption (3a) is satisfied. Furthermore, if the step size is in the form of $\eta_t = \frac{2}{t+1}$, then $\mathbb{E}_{1:T}[D_\Omega(\pi^* \| \pi_t)] = \mathcal{O}(1/T)$.*

Therefore, we can guarantee the ideal convergence in the SB learning when the scheduling of $\eta_t$ follows the step size assumptions. Next, we argue that general convergence toward $\pi_{\mathcal{D}}^{\circ}$ is guaranteed under Assumption (3b). Given the convex nature of SB cost functionals, we argue that this convergence toward $\pi_{\mathcal{D}}^{\circ}$ is beneficial as long as $\pi_t^{\circ}$ is trained to approximate $\pi^*$ and remain bounded. Therefore, we argue that the convergence of SB is beneficial and address the following statement.

**Proposition 1** (Convergence). *Suppose that $\pi^* \neq \pi_{\mathcal{D}}^{\circ}$, hence $\inf_{\pi \in \mathcal{C}} \mathbb{E}[F_t(\pi)] > 0$. If the step sizes $\{\eta_t\}_{t=0}^{\infty}$ satisfies Assumption 3, then $\lim_{t \to \infty} \mathbb{E}_{1:t}[D_\Omega(\pi_{\mathcal{D}}^{\circ} \| \pi_t)]$ converges to $0$ almost surely.*

Lastly, assume that the log Sobolev inequality in Assumption 1 holds with continuity of potentials. We establish an online learning regret bound of $\mathcal{O}(\sqrt{T})$ for certain instance of step sizes, demonstrating that imposing specific measure-theoretic properties in SBPs generalizes classical OMD results (Nesterov, 2009; Srebro et al., 2011; Orabona & Pál, 2018; Lei & Zhou, 2020). The analysis on our general online learning setup is compatible with these results by using the dual norm $\|\hat{g}_t\|$ defined by the first variation $\delta_{\mathcal{C}}$.

**Proposition 2** (Regret bound). *Assume $\varphi, \psi \in C^2(\mathbb{R}^d) \cap \mathrm{Lip}(\mathcal{K})$, Assumption 1 holds, and the given costs $\{F_t\}_{t=1}^T$ are bounded. For arbitrary $u \in \mathcal{C}$ and a total step $T$, define $D^2 = \max_{1 \leq t \leq T} D_\Omega(u \| \pi_t)$. (a) When the number of time step is known a priori, the regret is bounded to $2D\sqrt{2\omega^{-1}\mathcal{K}T}$ for a constant step size $\eta \equiv \frac{D\sqrt{\omega}}{\sqrt{2\mathcal{K}T}}$. (b) For an adaptive scheduling $\eta_t = D\sqrt{\omega}/\sqrt{2\sum_{i=1}^t \|\hat{g}_i\|^2}$ the regret is bounded to $2D\sqrt{2\omega^{-1}\sum_{t=1}^T \|\hat{g}_t\|^2}$ where $\hat{g}_t = \delta_{\mathcal{C}} \Omega(\pi_t) - \delta_{\mathcal{C}} \Omega(\pi_t^{\circ})$.*

Note that although our analysis establishes a rigorous connection between SB and OMD, it inherits certain limitations from classical regret analyses. For instance, sublinear regrets in Proposition 2 relies on an additional boundedness assumption on costs, and there exist some cases of Assumption 3 that may yield asymptotically linear regret (Orabona & Pál, 2018). Addressing these limitations may involve advanced hybrid OMD methodologies which are actively being studied, such as dual averaging (Fang et al., 2022) or FTRL (Chen & Orabona, 2023). As exploring (as well as computing) such extensions for SBPs falls beyond

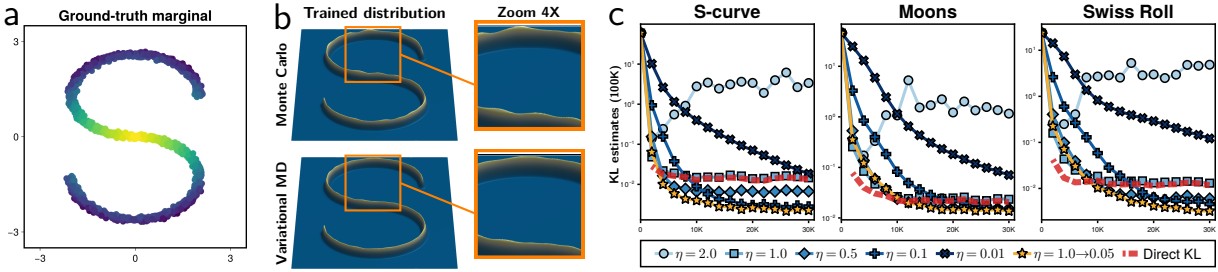

Figure 6: Variational MD with synthetic datasets. (a) A distribution is accessible by finite batch data. (b) 3D surfaces of $(\vec{\pi}_T^\circ, \vec{\pi}_T)$ trained by Monte Carlo method for KL (top) and variational MD (bottom) show that the MD results in more stable outcomes. (c) The plots show the estimated $\mathrm{KL}(\vec{\pi}_t \| \vec{\pi}^*)$ with different step size scheduling (5 runs), with red dashed baselines $\mathrm{KL}(\vec{\pi}_t^\circ \| \vec{\pi}^*)$.

our scope, from now on we focus on practical implementations of OMD that demonstrate theoretical convergence (Proposition 1) over a relatively long horizon of several hundred steps. Hence, we adopt Assumption 3 and provide corresponding experimental evidence to substantiate its validity in practical scenarios.

## 4.3 Online mirror descent updates using Wasserstein gradient flows

The remaining challenge is practicality of our OMD theory, namely the accurate computability of each critical point for (13). To resolve this issue, this work newly presents an approximation method using Wasserstein gradient flows (Jordan et al., 1998) through an equivalence property of first variations. Suppose we expand a subinterval $[t, t+1)$ for each OMD step (13) into continuous dynamics of $\rho(\tau) \in \mathcal{C}$ for a $\tau \in [0, \infty)$. By Otto's calculus on the Wasserstein space, known as the Otto calculus (Otto, 2001; see Appendix B.1), one can describe the dynamics of $\rho_\tau$ for minimizing a strictly convex functional $\mathcal{E}_t : \mathcal{C} \to \mathbb{R}$ as the PDE

$$\partial_\tau \rho_\tau = -\nabla_{\mathbb{W}} \mathcal{E}_t(\rho), \tag{16}$$

where $\nabla_{\mathbb{W}}$ denotes the Wasserstein-2 gradient operator $\nabla_{\mathbb{W}} := \nabla \cdot \left( \rho \nabla \frac{\delta}{\delta \rho} \right)$. In this work, we adopt the Wasserstein gradient flow theory (Jordan et al., 1998) to efficiently perform OMD where the equilibrium indicates the subsequent iterate $\pi_{t+1}$. Note that Wasserstein gradient flows are asymptotically stable by the LaSalle's invariance principle (Carrillo et al., 2023). We present a simple and exact closed-form expression for the VOMD update. Note that the cost $F_t(\cdot) = \mathrm{KL}(\cdot \| \pi_t^\circ)$ satisfies the 1-relative-smoothness and 1-strong-convexity relative to $\Omega$ (see Definition 6; Aubin-Frankowski et al., 2022). Then, a first variation of the OMD problem can be decomposed into multiple variations of another problem with similar characteristics (*e.g.*, equilibrium, smoothness, and convexity). We present the following theorem for the computation of OMD.

**Theorem 2** (Dynamics equivalence in first variation)**.** *Consider the Wasserstein gradient dynamics of* (16) *which solves a local update of* (13)*. The gradient dynamics of updates are equivalent to that of a linear combination of KL functionals such that for any $\rho_\tau \in \mathcal{C}$*

$$\eta_t \delta_c \mathcal{E}_t(\rho_\tau) = \delta_c \big\{ \eta_t \mathrm{KL}(\rho_\tau \| \pi_t^\circ) + (1 - \eta_t) \mathrm{KL}(\rho_\tau \| \pi_t) \big\} \quad \forall \rho_\tau \in \mathcal{C}, \tag{17}$$

*and the PDE* (17) *converges to a unique critical point of subsequent OMD iterate* (13) *as $\tau \to \infty$.*

*Sketch of Proof.* We identify $\delta \mathcal{E}_t$ as a dynamics that reaches an equilibrium solution for

$$
\begin{aligned}
&\underset{\pi \in \mathcal{C}}{\text{minimize}} \; \big\langle \delta_c F_t(\pi_t), \pi - \pi_t \big\rangle + \tfrac{1}{\eta_t} D_\Omega(\pi \| \pi_t) \\
&\iff \quad \underset{\pi \in \mathcal{C}}{\text{minimize}} \; \eta_t \underbrace{D_\Omega(\pi \| \pi_t^\circ)}_{\text{empirical estimates}} + (1 - \eta_t) \underbrace{D_\Omega(\pi \| \pi_t)}_{\text{proximity}},
\end{aligned}
\tag{18}
$$

and then the equivalence of first variation for recursively defined Bregman divergences is applied (Lemma 6). At a glance, Eq. (18) appears analogous to the interpolation search between two points, where the influence of $\pi_t^\circ$ is controlled by $\eta_t$. We leave the full version of proof in Appendix A.5. $\qquad \square$

Theorem 2 holds practical importance for OMD computation, since following the argument allows us to perform gradient-based updates without directly constructing a desired Bregman divergence. That is, updates can be drawn based on a linear combination of gradient flows $\eta_t \nabla_{\mathbb{W}} \mathrm{KL}(\rho_\tau \| \pi_t^\circ) + (1 - \eta_t) \nabla_{\mathbb{W}} \mathrm{KL}(\rho_\tau \| \pi_t)$, where such expression has been extensively studied both theoretically and computationally (Carrillo et al., 2023; Lambert et al., 2022). Therefore, we can utilize interpolation of Wasserstein gradient flows for performing updates and utilize a certain variational class for reducing the computational cost. Fig. 6 shows our actual experiments using GMMs. Let a reference estimation be fitted using a Monte Carlo method, and our model be trained through a variational OMD method which is explained in the subsequent section. We initially observed that the VOMD method provides stability improvement when $\eta < 1$. In contrast, the condition of $\eta > 1$ performed worse than the Monte Carlo method and $\eta = 1$ showed almost equivalent performance. Furthermore, the performance of VOMD was greatly improved by choosing a harmonic step size scheduling in the interval $[1.0, 0.05]$. All of these results on variational approximation precisely matches our analysis.

## 5 Algorithm: Variational Mirrored Schrödinger Bridge

In this section, we propose variational mirrored Schrödinger bridge, a simulation-free method that offers iterative MD updates for parameterized SB models with mixture models, using the Wasserstein-Fisher-Rao geometry. We provide a variational interpretation for LightSB models and draw a practical VOMD updates algorithm that closely resembles ordinary machine learning methods.

### 5.1 Gaussian mixture parameterization for the Schrödinger bridge problem

In order to verify theoretical claims in practice, this section presents a computational extension of our VOMD theory, which operates within a particular subspace of the Wasserstein geometry introduced in § 4.3. Recently, Korotin et al. (2024) proposed the GMM parameterization, which provides theoretically and computationally desirable models for our variational OMD approach. The parameterization considers the *adjusted* Schrödinger potential $u^*(x) := \exp(\varphi^*(x) - \|x\|^2 / 2\varepsilon)$ and $v^*(y) := \exp(\psi^*(y) - \|y\|^2 / 2\varepsilon)$ such that we have a proportional property $\pi^*(y|x) \propto \exp(\langle x, t \rangle / \varepsilon) v^*(y)$. With a finite set of parameters $\theta \triangleq \{\alpha_k, m_k, \Sigma_k\}_{k=1}^K$ for weights $\alpha_k > 0$, means $m_k \in \mathbb{R}^d$ and covariances $\Sigma_k \in \mathbf{S}_{++}^d$, Korotin et al. (2024) proposed to approximate the adjusted Schrödinger potential $v_\theta$ and conditional probability density $\vec{\pi}_\theta$

$$v_\theta(y) := \sum_{k=1}^K \alpha_k \mathcal{N}(y | m_k, \varepsilon \Sigma_k), \qquad \vec{\pi}_\theta^x(y) := \frac{1}{z_\theta^x} \sum_{k=1}^K \alpha_k^x \mathcal{N}(y | m_k^x, \varepsilon \Sigma_k), \tag{19}$$

where GMM component for $\vec{\pi}_\theta^x$ is conditioned by an input $x$: $m_k^x := m_k + \Sigma_k x$, $\alpha_k^x := \alpha_k \exp\left(\frac{x^\top \Sigma_k x + \langle m_k, x \rangle}{2\varepsilon}\right)$, $z_\theta^x := \sum_{k=1}^K \alpha_k^x$ (see Proposition 3.2 of Korotin et al.). For this parameterization, the closed-from expression of SB process $\mathcal{T}_\theta$ is given as the following SDE for $t \in [0, 1)$:

$$\mathcal{T}_\theta : \mathrm{d}X_t = g_\theta(t, X_t)\,\mathrm{d}t + \sqrt{\varepsilon}\,\mathrm{d}W_t,$$
$$g_\theta(t, x) := \varepsilon \nabla \log \mathcal{N}(x | 0, \varepsilon(1-t)I_d) \sum_{k=1}^K \alpha_k \mathcal{N}(m_k | 0, \varepsilon \Sigma_k) \mathcal{N}(m_k(t, x) | 0, A_k(t)), \tag{20}$$

where $m_k(t, x) \triangleq \frac{x}{\varepsilon(1-t)} + \frac{1}{\varepsilon} \Sigma_k^{-1} m_k$ and $A_k(t) \triangleq \frac{t}{\varepsilon(1-t)} I_d + \frac{1}{\varepsilon} \Sigma_k^{-1}$. Therefore, the LightSB parameterization represent both static and dynamic SB models and arbitrary SB solvers can be applied without restrictions. We utilize the GMM parameterization for our computational algorithm for three key reasons. Firstly, the parameterization induce the universal approximation property for both $\vec{\pi}_\theta$ and $\mathcal{T}_\theta$ (Korotin et al., 2024). Secondly, GMMs are asymptotically log-concave (see Lemma 4), which is a fundamental assumption for our theory. Lastly, the GMM parameterization makes the computation of Wasserstein gradient flows with respect to the KL divergence tractable, which in turn enables us to apply a canonical treatment to computational convex optimization problems endowed with a Riemannian-like geometry.

### 5.2 Computation of VOMD in the Wasserstein-Fisher-Rao geometry

---

**Algorithm 1** Variational Mirrored SB (VMSB).

---

**Input:** SB models $(\vec{\pi}_\theta, \vec{\pi}_\phi)$ parameterized by Gaussian mixtures, step sizes $(\eta_1, \eta_T)$, $n_y, B \in \mathbb{N}$.
1: **for** $t \leftarrow 1$ **to** $T$ **do**
2:     Acquire $\phi_t$ with an external data-driven SB solver.
3:     $\theta_t \leftarrow \theta$, $\eta_t \leftarrow 1/\big(\eta_1^{-1} + (\eta_T^{-1} - \eta_1^{-1})(t^{-1}/T^{-1})\big)$
4:     **for** $n \leftarrow 1$ **to** $N$ **do**
5:         $\{x_i\}_{i=1}^B \leftarrow$ sample mini batch data from $\mu$.
6:         $\frac{\partial \mathcal{L}}{\partial \theta} \leftarrow \frac{1}{B}\sum_{i=1}^B \eta_t \texttt{WFRgrad}(\theta; \phi_t, x_i, n_y) + (1 - \eta_t)\texttt{WFRgrad}(\theta; \theta_t, x_i, n_y)$
7:         Update $\theta$ with the gradient $\frac{\partial \mathcal{L}}{\partial \theta}$.
8:     **end for**
9: **end for**
**Output:** Trained SB model $\vec{\pi}_\theta$.

---

For tractable computation, we formally derive a particular variant of Wasserstein gradient flow for the GMM parameterization. The space of Gaussian parameters $\mathbb{R}^d \times \mathbf{S}_{++}^d$, endowed with the Wasserstein-2 metric $W_2$, is formally recognized as the Bures–Wasserstein (BW) geometry (Bures, 1969; Bhatia et al., 2019; Lambert et al., 2022) $\texttt{BW}(\mathbb{R}^d) \subseteq \mathcal{P}_2(\mathbb{R}^d)$. The Wasserstein-Fisher-Rao (WFR) geometry, equivalently characterized by the spherical Hellinger–Kantorovich distance, extends this setting by considering liftings of positive, complete, and separable measures while preserving total mass (Liero et al., 2018; Chizat et al., 2018; Lu et al., 2019). Building upon the BW space, the Wasserstein-Fisher-Rao geometry of GMMs, namely $\mathcal{P}_2(\texttt{BW}(\mathbb{R}^d))$, naturally provides liftings of Gaussian particles satisfying distributional consistency. In this work, we introduce the following proposition, which refines and extends the results from Lambert et al. (2022, § 6) specifically enhancing their framework through the introduction of freely trainable GMM weights $\alpha_k$.

**Proposition 3** (WFR gradient dynamics). *Suppose a time-varying GMM model $\rho_{\theta_\tau}$ with the parameter $\theta_\tau = \{\alpha_{k,\tau}, m_{k,\tau}, \Sigma_{k,\tau}\}_{k=1}^K$ at time $\tau$. Let $y_{k,\tau} \sim \mathcal{N}(m_{k,\tau}, \Sigma_{k,\tau})$ denote a sample from the k-th Gaussian particle of $\rho_{\theta_\tau}$. Then, the WFR dynamics $\nabla_{\texttt{WFR}}\mathrm{KL}(\rho_{\theta_\tau}\|\rho^*)$ wrt $\dot{\theta}_\tau = \{\dot{\alpha}_{k,\tau}, \dot{m}_{k,\tau}, \dot{\Sigma}_{k,\tau}\}_{k=1}^K$ are given as*

$$
\dot{\alpha}_{k,\tau} = -\bigg(\mathbb{E}\bigg[\log\frac{\rho_{\theta_\tau}}{\rho^*}(y_{k,\tau})\bigg] - \frac{1}{z_\tau}\sum_{\ell=1}^K \alpha_\ell \mathbb{E}\bigg[\log\frac{\rho_{\theta_\tau}}{\rho^*}(y_{\ell,\tau})\bigg]\bigg)\alpha_{k,\tau},
$$
$$
\dot{m}_{k,\tau} = -\mathbb{E}\bigg[\nabla\log\frac{\rho_{\theta_\tau}}{\rho^*}(y_{k,\tau})\bigg], \quad \dot{\Sigma}_{k,\tau} = -\mathbb{E}\bigg[\nabla^2\log\frac{\rho_{\theta_\tau}}{\rho^*}(y_{k,\tau})\bigg]\Sigma_{k,\tau} - \Sigma_{k,\tau}\mathbb{E}\bigg[\nabla^2\log\frac{\rho_{\theta_\tau}}{\rho^*}(y_{k,\tau})\bigg],
$$
(21)

*for $\tau \in [0, \infty)$, where $z_\tau := \sum_{k=1}^K \alpha_k$; $\nabla$ and $\nabla^2$ denote gradient and Hessian with respect to $y_{k,\tau}$.*

Appendices A.6 and B contain the complete theory. Proposition 3 argues that the one parameter family $\theta_\tau$ predicts a gradient-based algorithm of $\nabla_{\texttt{WFR}}\mathrm{KL}(\rho_{\theta_\tau}\|\rho^*)$, and thus Eq. (21) can be directly used for training GMM models. Recall that GMMs have a closed form expression of likelihoods, which means each log likelihood difference can be calculated without errors. Given that the target has the identical number of Gaussian particles, both Eq. (21) and its approximation using finite samples strictly induce zero gradients at the equilibrium. Hence, we argue that the simulation-free algorithm VMSB will result in more robust and stable outcomes than standard data-driven SB learning. Our VOMD framework can be implemented in modern deep learning libraries and, when coupled with advanced optimizers that provide inherently more stable gradient estimates (e.g., adaptive learning rate schedules), the proposed method empirically find convergence even faster than the conservative rates predicted by theory.

### 5.3 Algorithmic considerations

Algorithm 1 outlines the overall procedure. The proposed VMSB algorithm requires SB parameters $\theta$ and $\phi$, which represents $\vec{\pi}_t$ and $\vec{\pi}_t^\circ$ from the theoretical framework in § 4.2. The target model $\vec{\pi}_\phi$ is independently fitted using an arbitrary SB solver. By the results of analysis, one can schedule of the step size $\eta_t$ with a harmonic progression satisfying Assumption 3; thus, we propose to schedule by the series for $1 \geq \eta_1 \geq \eta_T > 0$ as in Line 3 of the algorithm. In our settings, the hyperparameters are set $\eta_1 = 1$ and $\eta_T \in \{0.05, 0.01\}$

which varies depending on each length of training. The algorithm can also put "warm up" steps leveraged by a existing solver, and start from $\theta = \phi_t$ enforcing $\eta_t \equiv 1$ for a certain period of the early stage.

The VMSB algorithm is essentially designed to perform the following approximation of the WFR gradient operation (21) in Proposition 3, approximated by the following equation with finite data samples $\{x_i\}_{i=1}^B \sim \mu$

$$\tfrac{1}{B} \sum_{i=1}^B \mathtt{WFRgrad}(\theta; \phi, x_i, n_y) \approx \nabla_{\mathtt{WFR}} \mathrm{KL}(\vec{\pi}_\theta \| \vec{\pi}_\phi),$$

where each expectation is estimated using $n_y$ samples from each Gaussian particle. Following Theorem 2, we propose to update the SB model $\vec{\pi}_\theta$ with $\eta_t \mathtt{WFRgrad}(\theta; \phi_t, x_i, n_y) + (1 - \eta_t)\mathtt{WFRgrad}(\theta; \theta_{t-1}, x_i, n_y)$ at each VOMD iteration $t$ (see Line 6). When $\mu$ is a zero-centered distribution, we set $B = 1$ and $x = 0$ for the fast training time. This trick is equivalent to training the adjusted Schrödinger potential (Korotin et al., 2024) $v_\theta := \sum_{k=1}^K \alpha_k \mathcal{N}(y|m_k, \varepsilon\Sigma_k) \propto \pi_\theta(\cdot|x=0)$ directly, which enables the VMSB algorithm to run efficiently for certain tasks. We argue that our design provides a simple yet faithful realization of OMD updates, yielding a procedure that closely resembles classical gradient descent in machine learning. Although our methodology and computational strategy build on well-established ideas (Lambert et al., 2022; Aubin-Frankowski et al., 2022; Karimi et al., 2024), we deliberately integrate them into a unified framework to verify our novel online learning theory for the SB problem. As a result, VMSB emerges as a robust solver that embeds OMD within a variational formulation, offering both rigorous theoretical guarantees and clear computational advantages.

## 6 Experimental Results

**Experiment goals.** We aimed to test our online learning hypothesis and verify that the VMSB effectively induces OMD updates. Since our theoretical claims are intended to be highly versatile, consistent performance improvements for each setting coincides with the generality of the proposed VOMD method. We delineate our objectives as follows: ① We aimed to affirm our online learning hypothesis by demonstrating consistent improvements. ② We sought to corroborate our theoretical results, aiming for stable performance that consistently exceeds that of benchmarks. ③ We aimed to verify that our algorithm effectively induces OMD by the Wasserstein gradient flow. To achieve these goals, we validate our algorithm across diverse SB problem settings, including online learning scenarios, classical OT benchmarks, and image translation tasks.

**Baselines and VMSB variants.** Korotin et al. (2024) proposed a streamlined, simulation-free solver referred to as **LightSB** that optimizes $\phi$ through Monte Carlo approximation of $\mathrm{KL}(\vec{\pi}^* \| \vec{\pi}_\phi)$. As an alternative, **LightSB-M** (Gushchin et al., 2024a) reformulated the reciprocal projection from DSBM (Shi et al., 2023) to a projection method termed *optimal projection*, establishing approximated bridge matching for the path measure $\mathcal{T}_\phi$. Applying Algorithm 1, we derived two distinct methods called **VMSB** and **VMSB-M** ($\vec{\pi}_\theta$), trained upon LightSB and LightSB-M solvers ($\vec{\pi}_\phi$), respectively. Since the theoretical arguments imply that the algorithm is agnostic to targets, the performance benefits of VMSB variants from their references support the generality of our claims. Additionally, we adopted VMSB on *hybrid* settings, leveraging networks or embeddings for complex problems. We refer readers to Appendix D for additional experimental setups.

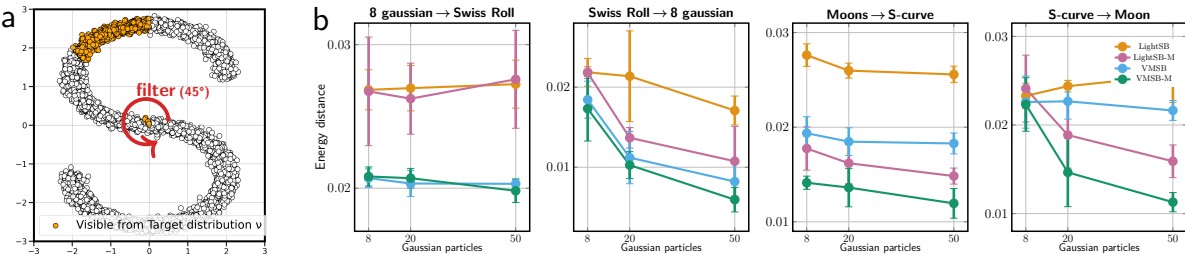

Figure 7: Online SBPs for synthetic dataset streams. (a) We designed an online learning problem with a rotating filter where an algorithm is allowed to observe the data in $y \sim \nu$ only 12.5% at a time. (b) The plots show that our VMSB and VMSB-M show consistent improvements from their references regarding the ED metric with 95% confidence intervals for 5 runs with different seeds.

Table 2: EOT benchmark scores with cB$\mathbb{W}_2^2$-UVP ↓ (%) between the optimal coupling $\pi^*$ and the learned model $\pi_\theta$ (5 runs). Results of classical EOT solvers marked with † are taken from (Korotin et al., 2024), and ‡ from (Gushchin et al., 2024a). Additionally, LightSB-EMA indicates a hybrid approach using the exponential moving average techniques (EMA; Morales-Brotons et al., 2024) for LightSB parameters (*decay* = 0.99). Our VMSB and VMSB-M results are highlighted in bold when VMSB methods exceed their reference algorithm.

| Type | Solver | $\varepsilon = 0.1$ | | | | $\varepsilon = 1$ | | | | $\varepsilon = 10$ | | | |
|---|---|---|---|---|---|---|---|---|---|---|---|---|---|
| | | $d = 2$ | $d = 16$ | $d = 64$ | $d = 128$ | $d = 2$ | $d = 16$ | $d = 64$ | $d = 128$ | $d = 2$ | $d = 16$ | $d = 64$ | $d = 128$ |
| Classical solvers (best; Korotin et al.)[†] | | 1.94 | 13.67 | 11.74 | 11.4 | 1.04 | 9.08 | 18.05 | 15.23 | 1.40 | 1.27 | 2.36 | 1.31 |
| Bridge-M | DSBM (Shi et al.)[‡] | 5.2 | 10.8 | 37.3 | 35 | 0.3 | 1.1 | 9.7 | 31 | 3.7 | 105 | 3557 | 15000 |
| Bridge-M | SF$^2$M-Sink (Tong et al.)[‡] | 0.54 | 3.7 | 9.5 | 10.9 | 0.2 | 1.1 | 9 | 23 | 0.31 | 4.9 | 319 | 819 |
| rev. KL | LightSB (Korotin et al.) | 0.007 | 0.040 | 0.100 | 0.140 | 0.014 | 0.026 | 0.060 | 0.140 | 0.019 | 0.027 | 0.052 | 0.092 |
| Bridge-M | LightSB-M (Gushchin et al.) | 0.017 | 0.088 | 0.204 | 0.346 | 0.020 | 0.069 | 0.134 | 0.294 | 0.014 | 0.029 | 0.207 | 0.747 |
| EMA | LightSB-EMA | 0.005 | 0.040 | 0.078 | 0.149 | 0.012 | 0.022 | 0.051 | 0.127 | 0.017 | 0.021 | 0.025 | 0.042 |
| Var-MD | VMSB (ours) | **0.004** | **0.012** | **0.038** | **0.101** | **0.010** | **0.018** | **0.044** | **0.114** | **0.013** | **0.019** | **0.021** | **0.040** |
| Var-MD | VMSB-M (ours) | **0.015** | **0.067** | **0.108** | **0.253** | **0.010** | **0.019** | **0.094** | **0.222** | **0.013** | **0.029** | **0.193** | **0.748** |

## 6.1 Online SB learning for synthetic data streams

To validate our online learning hypothesis, we first considered two-dimensional synthetic SB problems for data streams depicted in Fig. 7 (a). We applied an angle-based rotating filter, making the marginal as a data stream where only 12.5% (or 45-degree angle) of the total data is accessible for each step $t$. We trained conditional models $\vec{\pi}_\theta$ with VMSB and $\vec{\pi}_\phi$ with other baseline SB solvers for the 2D problem, respectively. Fig. 7 (b) shows the plots of squared energy distance (ED), which is a special instance of squared maximum mean discrepancy (MMD), approximating the $L^2$ distance between distributions: $\mathrm{ED}(P, Q) = \int (P(x) - Q(x))^2 \mathrm{d}x$ (Rizzo & Székely, 2016). In our ED evaluation, the VMSB algorithm achieved a strictly lower divergence than the LightSB and LightSB-M solvers for various numbers of Gaussian particles $K$. Therefore, we conclude that these results aligned with our hypothesis and theory of online mirror descent.

## 6.2 Quantitative Evaluation

**EOT benchmark.** Next, we considered the EOT benchmark proposed by Gushchin et al. (2024b), which contains 12 entropic OT problems with different volatility and dimensionality settings. Table 2 shows that LightSB and VMSB methods outperforms other method in terms of the cB$\mathbb{W}_2^2$-UVP metric as previously reported by Korotin et al. (2024) and Gushchin et al. (2024a). We also observed that a hybrid approach combining LightSB and the exponential moving average (EMA; Morales-Brotons et al., 2024) named as LightSB-EMA was affective for improving stability. Among 24 different settings, our MD approach exceeded the reference model and the EMA method in 23 settings in terms of the cB$\mathbb{W}_2^2$-UVP metric (Gushchin et al., 2024b). Our replication of LightSB/LightSB-M achieved better performance than originally reported results, and our method accordingly reached the state-of-the-art performance in this benchmark with stability, which represents strong evidence of Proposition 1. Among all cases, the only exception was LightSB-M, which had the highest dimension and volatility. We suspected that the drift form Eq. (20), which is proportional to $\varepsilon$, may have violated our assumptions Assumption 2 and the boundedness assumption during the training. Thus, we conclude that our variational MD training is effective in various EOT setups. Tables 9 and 10 in the appendix show comprehensive statistics on this benchmark with a wider range of SB solvers.

**SB on single cell dynamics.** We evaluated VMSB on unpaired single-cell data problems in the high-dimensional single cell dynamics experiment (Tong et al., 2024a). The dataset provided single cell data from four donors on days 2, 3, 4, and 7, describing the gene expression levels of distinct cells. Given samples collected on two different dates, the task involves performing inference on temporal evolution, such as interpolation and extrapola-

Table 3: Energy distance on the MSCI dataset (95% confidence interval, ten trials with instances of two setups). Results marked with ‡ are from (Gushchin et al., 2024a).

| Type | Solver | $d = 50$ | $d = 100$ | $d = 1000$ |
|---|---|---|---|---|
| Sinkhorn | Vargas et al. (2021)[†] | 2.34 | 2.24 | 1.864 |
| Bridge-M | DSBM (Shi et al.)[‡] | 2.46 ± 0.1 | 2.35 ± 0.1 | 1.36 ± 0.04 |
| Bridge-M | SF$^2$M-Sink (Tong et al.)[‡] | 2.66 ± 0.18 | 2.52 ± 0.17 | 1.38 ± 0.05 |
| rev. KL | LightSB | 2.31 ± 0.08 | 2.15 ± 0.09 | 1.264 ± 0.06 |
| Bridge-M | LightSB-M | 2.30 ± 0.08 | 2.15 ± 0.08 | 1.267 ± 0.06 |
| Var-MD | VMSB (ours) | **2.28 ± 0.09** | **2.13 ± 0.09** | **1.260 ± 0.06** |
| Var-MD | VMSB-M (ours) | **2.26 ± 0.10** | **2.12 ± 0.09** | **1.265 ± 0.05** |

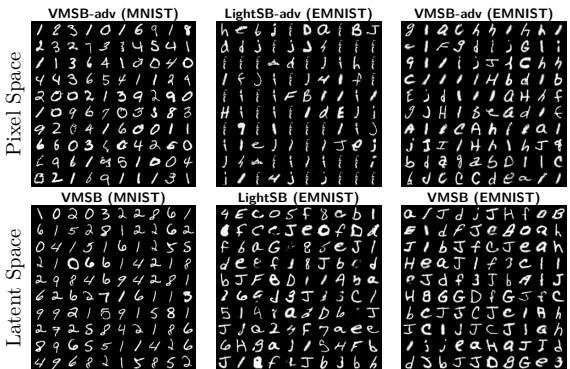

Figure 8: Generated MNIST/EMNIST samples. Top: Raw pixel SB results. Bottom: Latent SB results.

Table 4: FID and MSD scores in EMNIST-to-MNIST translation tasks. Hyperparameters between LightSB and VMSB are shared. We examined the scores with five runs for the ALAE case.

| | Method | FID | MSD |
|---|---|---|---|
| U-net | SF$^2$M-Sink | 23.215 | 0.456 |
| | DSBM-IPF | 15.211 | 0.352 |
| | DSBM-IMF | 11.429 | 0.373 |
| Pixel | LightSB-adv | 20.017 | 0.362 |
| | VMSB-adv (ours) | 15.471 | 0.356 |
| ALAE | LightSB | $9.183_{\pm 0.569}$ | $0.371_{\pm 0.018}$ |
| | VMSB (ours) | $\mathbf{8.774_{\pm 0.065}}$ | $0.365_{\pm 0.002}$ |

tion of PCA projections with $\{50, 100, 1000\}$ dimensions. We evaluated the energy distance over ten trials, which were divided into two distinct settings with five runs each. The first setting spanned from Day 2 to Day 4 (evaluated at Day 3), while the second setting considers duration from Day 3 to Day 7 (evaluated at Day 4)[2]. The quantitative results in Table 3 show that our VMSB method outperforms existing approaches by marginal amount, demonstrating its competitiveness as a practical SB solver for real-world problems.

### 6.3 Unpaired image-to-image transfer

**MNIST-EMNIST.** We applied VMSB to unpaired image translation tasks for MNIST and EMNIST datasets. In these tasks, LightSB methods struggled to generate raw pixels due to the limited scalability of the loss function. To solve this issue, we opted to find a viable alternative to LightSB the raw pixel space, and we discovered that the capabilities of GMM parameterization can be extended by incorporating the adversarial learning technique (Goodfellow et al., 2014; see Appendix D.5) was effective in providing rich learning signals for $\pi_\phi$. Therefore, we named the adversarial method and the VMSB adaptation **LightSB-adv** and **VMSB-adv**. Also, we pretrained encoder networks using the *Adversarial Latent AutoEncoder* (ALAE; Pidhorskyi et al., 2020) technique, and applied the LightSB and VMSB algorithms on the 128-dimensional latent space that represent the both of data. Fig. 8 shows that VMSB/VMSB-adv outperformed Light/LightSB-adv (with identical architecture) in the fidelity of samples and semantics of letters for latent and pixel spaces. In Table 4, the VMSB method on the ALAE embedding space was able to surpass deep SB models with a fewer number of parameters of $K = 256$. Even for raw pixels, our algorithm also achieved competitive FID and input/output MSD similarity scores for $K = 4096$. The consistent performance gains from the LightSB and LightSB-adv algorithms strongly supports our theoretical claims on online learning.

**FFHQ.** Following the latent SB setting of Korotin et al. (2024), we assessed our method by utilizing a pretrained ALAE model for generating $1024 \times 1024$ images of the FFHQ dataset (Karras et al., 2019). With the predefined 512-dimensional embedding space, we trained our SB models on the latent space to solve four distinct tasks: *Adult → Child*, *Child → Adult*, *Female → Male*, and *Male → Female*. Fig. 9 illustrates that our method delivered high-quality translation results. We also conducted a quantitative analysis using the ED on the ALAE embedding as a metric for evaluation, and the corresponding quantitative results are reported in Table 13. The result also verifies that our VMSB and VMSB-M algorithms consistently achieved lower ED scores than other baselines, demonstrating its applicability for the high dimensional embedding space. Consequently, the image-to-image transfer results showed that the generality of our online learning hypothesis and that the proposed algorithm is highly capable of interacting with neural networks of complex learning dynamics. Considering the significantly higher dimensionality of image domains relative to the batch sizes used in VOMD, the consistent and stable performance improvements demonstrated in our experiments strongly validate our theoretical claims regarding the robustness of our approach in online learning scenarios.

---

[2]In other words, we directly follows the original configuration of Tong et al. (2024a).

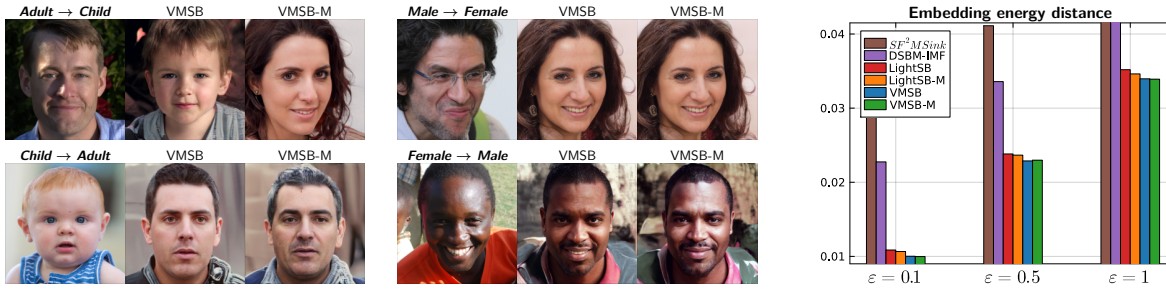

Figure 9: Image-to-Image translation on a latent space. Left: Generation results for the FFHQ dataset $(1024 \times 1024)$ using our two SB variants. Right: Quantitative results of ED metrics for ALAE embeddings.

# 7 Conclusion

In this paper, we introduced VMSB, a practical simulation-free Schrödinger bridge algorithm based on the online learning theory, designed for effectively solving SB problems encountered in real-world scenarios. We proposed a robust theoretical learning framework applicable to general SB solvers, leveraging a dual geometric interpretation of convex optimization to construct a robust OMD algorithm with rigorous guarantees on convergence and regret bounds. Furthermore, we proposed the computational algorithm for our OMD framework by employing the Wasserstein-Fisher-Rao geometry. Through extensive empirical evaluation, we validated the effectiveness of VMSB across diverse settings, including high-dimensional spaces, limited-sample regimes, and online learning environments. The experimental results consistently demonstrated stable and superior benchmark performance, highlighting the enhanced robustness of our approach. Consequently, we argue that the proposed VMSB algorithm and our theoretical arguments regarding VOMD provide a promising and robust methodology for probabilistic generative modeling within learning-theoretic contexts.

**Limitations.** In this work, we significantly reduced the computational complexity inherent in the MD framework by adopting the Wasserstein-Fisher-Rao geometry. GMM-based models, due to the lack of deep structural processing, tend to focus on *instance-level* associations of images in EOT couplings rather than the *subinstance-* or *feature-level* associations that are intrinsic to deep generative models. As a result, while VMSB produces statistically valid representations of optimal transportation within the given architectural constraints, these outcomes may be perceived as somewhat synthetic compared to large neural networks. Nevertheless, GMM-based models still hold an irreplaceable role in numerous problems such as latent diffusion and variational methods, due to their simplicity and distinctive properties Korotin et al. (2024). As we successfully demonstrated in two distinct ways of interacting with neural networks for solving unpaired image transfer, we hope our theoretical and empirical findings help novel neural architecture studies. While VMSB strictly outperforms existing SB solvers across standard numerical benchmarks, the performance gains are marginal in some scenarios. For instance, the single-cell dynamics experiment utilizes a PCA-based preprocessing pipeline (Tong et al., 2024a), and the transformation into this lower-dimensional space compromises the biological relevance of the SB objective, consequently narrowing the discernible performance gap.

**Future research.** One line of future studies in SB is a general understanding of learning in diffusion models with various regularizations. This includes diffusion models in various problem-specific constraints, and geometric constraints from manifolds. Another direction is the extension of the theoretical results into network architecture design. From § 4.2, a pair of Schrödinger potentials represent a dual representation of SB in a statistical manifold. In Gigli & Tamanini (2020), such potentials satisfy the Hamilton-Jacobi-Bellman (HJB) equations and, this can be trained with forward-backward SDE (SB-FBSDE) as presented by Liu et al. (2022). However, this requires many simulation samples from SDEs, and the requirements for applying VMSB contain a tractable way of estimating gradient flows, and a guarantee of measure concentration. Therefore, we expect there will be a new studies of energy-based neural architecture for efficiently representing SB, which will advance various subfields of machine learning. Lastly, a theoretical generalization of our work can be done by considering the Orlicz space for EOT studied by Lorenz & Mahler (2022). Since we essentially devised our theoretical framework to be compatible with arbitrary Bregman potentials, we believe controlling regularity of Young functionals can find more generalized learning algorithms for a wider range of OT problems.

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

**Appendices for**

# Variational Online Mirror Descent for Robust Learning in Schrödinger Bridge

## Abbreviation and Notation

| Abbreviation | Expansion |
|---|---|
| SB | Schrödinger Bridge |
| SBP | Schrödinger Bridge Problem |
| EOT | Entropy-regularized Optimal Transport |
| MD | Mirror Descent |
| OMD | Online Mirror Descent |
| KL | Kullback-Leibler |
| IPF | Iterative Proportional Fitting |
| BW | Bures–Wasserstein |
| WFR | Wasserstein-Fisher-Rao |
| SDE | Stochastic Differential Equation |
| PDE | Partial Differential Equation |
| FP | Fokker–Planck |
| GMM | Gaussian mixture model |

| Notation | Usage |
|---|---|
| $\mu, \nu$ | marginal distributions |
| $\varepsilon$ | volatility of reference measure |
| $c_\varepsilon$ | cost $c_\varepsilon(x, y) := \frac{1}{2\varepsilon}\|x - y\|^2$ |
| $\pi$ | a coupling of $\mu$ and $\nu$ |
| $\vec{\pi}, \overleftarrow{\pi}$ | conditional distributions |
| $\gamma_n$ | $n$-th marginal |
| $\varphi, \psi$ | log-Schrödinger potential |
| $\Omega, D_\Omega$ | Bregman potential/divergence |
| $d^+$ | directional derivative |
| $\delta_\mathcal{C}, \delta_\mathcal{D}$ | First variations |
| $\nabla_{\mathrm{w}}$ | Wasserstein-2 gradient operator |
| $\mathcal{T}$ | dynamic stochastic process in SB |
| $g$ | drift function |
| $i_\mathcal{C}$ | indicator function |

## A  Theoretical Details and Proofs

**Background on first variation operators.** In this paper, we utilize the notation of first variation operators $\delta_\mathcal{C}$ and $\delta_\mathcal{D}$ to identify the generalized primal and dual spaces in Schrödinger bridge. Since the problems are classified as an infinite-dimensional optimization (Aliprantis & Border, 2006), we introduce the essential background supporting the necessity of these operators. We introduce Gâteaux and Fréchet differentiablility (Aubin-Frankowski et al., 2022; Karimi et al., 2024).

**Definition 4** (Gâteaux & Fréchet differentiablility)**.** Let $\mathcal{M}$ be a topological vector space of measures on a space $\mathcal{X}$. Define the Gâteaux differentiablity of a functional $F : \mathcal{M} \to \mathbb{R}$, if there exists a gradient operator $\nabla_{\mathrm{Gât}}$ such that for an arbitrary direction $v \in \mathcal{M}$, defined as the limit

$$\nabla_{\mathrm{Gât}}F(x)[v] = \lim_{h \to 0} \frac{F(x + hv) - F(x)}{h}, \quad x \in \mathcal{M}$$

If the limit exists in the unit ball in $\mathcal{M}$, the function $F$ is called Fréchet differentiable with $\nabla_{\mathrm{Fré}}F(x)$.

The problem of the Gâteaux and Fréchet differentiability in the context of SB is that the limit must be given in *all* directions, implying that every neighboring point must be within the domain of the topological space $\mathcal{M}$. For the case of functionals such as the KL divergence functional $F(\cdot) = \mathrm{KL}(\cdot|\pi^*)$, the domain of $F$ and has an empty interior (Aubin-Frankowski et al., 2022). To resolve this issue, we use *directional derivatives* and *first variations*, defined in Definitions 1 and 2.

**First variations of KL.** Suppose that we have two distributions $\rho, \rho' \in \mathcal{P}_2(\mathcal{X}), \mathcal{X} \subseteq \mathbb{R}^d$. Let us consider the log likelihood of $\rho'$: $\ell'(x) := \log \rho'(x)$, and an element of a (topological) tangent space $v \in T_\rho \mathcal{P}_2(\mathcal{X})$ (Milnor, 1964). Then, we can achieve the followings:

$$\mathrm{KL}(\rho\|\rho') = \int_\mathcal{X} \log \rho(x) \, \mathrm{d}\rho(x) - \int_\mathcal{X} \ell'(x) \, \mathrm{d}\rho(x) \tag{22}$$

$$\int \ell'(x)\big[\rho(x) + hv(x)\big] \, \mathrm{d}x = \int \ell'(x)\rho(x) \, \mathrm{d}x + h \int \ell'(x)v(x) \, \mathrm{d}x \tag{23}$$

Given that $\log(z+\varepsilon)(z+\varepsilon) = \log(z)z + [\log(z)+1]\varepsilon + o(\varepsilon)$, and $\int_{\mathcal{X}} v(x)\,\mathrm{d}x = 0$, we achieve

$$
\begin{aligned}
\int_{\mathcal{X}} \log\big(\rho(x) + hv(x)\big)\big(\rho(x) + hv(x)\big)\,\mathrm{d}x &= \int_{\mathcal{X}} \log\rho(x)\rho(x) + [\log\rho(x)+1]hv(x) + o(h)\,\mathrm{d}x \\
&= \int_{\mathcal{X}} \log\rho(x)\rho(x)\,\mathrm{d}x + h\int_{\mathcal{S}} \log\rho(x)v(x)\,\mathrm{d}x + h\int v(x)\,\mathrm{d}x + o(h)
\end{aligned}
\tag{24}
$$

Combining Eqs. (22-24), we achieve

$$
F(\rho + hv) = F(\rho) + h\left\langle \log\left(\frac{\rho}{\rho'}\right), v \right\rangle + o(h).
\tag{25}
$$

By Eq. (25) and Definition 2, the first variation $\delta F_2 \in T^*\mathcal{P}(\mathcal{X})$ exists for infinitesimal $h > 0$. Therefore, the first variation of KL is derived as $\delta\mathrm{KL}(\rho\|\rho') = \log\frac{\rho}{\rho'}$. In machine learning, log likelihoods of probabilistic models are often given in a closed-form expression, incentivizing development of computational continuous EOT/SB methods. Generally, identical arguments generally apply to all KL functionals with respect to distributions ($\pi$, $\vec{\pi}$, and marginals) in our setup.

**Asymptotically log-concave distributions.** For convergence analysis, we assume each marginal distribution is in log-concave distribution, particularly satisfying the log Sobolev inequality of measures, motivated by relevant literature (Otto & Villani, 2000; Conforti, 2024). This assumption works a wider range of costs and marginals beyond popular choices with boundedness and compactness (Nutz & Wiesel, 2023; Conforti et al., 2023). Suppose that marginals admit densities of the form

$$
\mu(\mathrm{d}x) = \exp\big(-U_\mu(x)\big)\mathrm{d}x \qquad \text{and} \qquad \nu(\mathrm{d}y) = \exp\big(-U_\nu(y)\big)\mathrm{d}y.
\tag{26}
$$

We exploit the following definition from (Conforti et al., 2023) in order to describe asymptotically log-concaveness.

**Definition 5** (Asymptotically strongly log-concavity; Conforti et al., 2023)**.** Suppose that marginals $\mu$ and $\nu$ admit a positive density against the Lebesgue measure, which can be written in the form (26). In particular, consider a collection of functions $\mathcal{G} := \{g \in C^2((0,+\infty),\mathbb{R}_+)|r \mapsto r^{1/2}g(r^{1/2})$ is non-increasing and concave, $\lim_{r\to 0} rg(r) = 0\}$. Accordingly, define a set

$$
\tilde{\mathcal{G}} := \{g \in \mathcal{G} \text{ bounded and s.t. } \lim_{r\to 0^+} g(r) = 0,\ g' \geq 0 \text{ and } 2g'' + gg' \leq 0\} \subset \mathcal{G}.
$$

and *convexity profile* $\kappa_U : \mathbb{R}_+ \to \mathbb{R}$ of a differentiable function $U$ as the following

$$
\kappa_U(r) := \inf\left\{ \frac{\langle \nabla U(x) - \nabla U(y), x - y \rangle}{|x-y|^2} \ : \ |x - y| = r \right\}.
$$

We say a potential is asymptotically strongly convex if there exists $\alpha_U \in \mathbb{R}_+$ and $\tilde{g}_U \in \tilde{\mathcal{G}}$ such that

$$
\kappa_U(r) \geq \alpha_U - r^{-1}\tilde{g}_U(r)
\tag{27}
$$

holds for all $r > 0$. We consider the set of asymptotically strongly log-concave probability measures

$$
\mathcal{P}_{\mathrm{alc}}(\mathbb{R}^d) := \{\zeta(\mathrm{d}x) = \exp(-U(x))\mathrm{d}x : U \in C_2(\mathbb{R}^d),\ U \text{ is asymptotically strongly convex}\}.
$$

It is essential to note that a mixture of asymptotically log concave is also asymptotically log concave.

**Lemma 4.** *For positive weights $\boldsymbol{\beta} = \{\beta_k\}_{k=1}^K$ with $\sum_{k=1}^K \beta_k = 1$ and asymptotically log concave distributions $\{\rho_k\}_{k=1}^K$, $\pi = \sum_{k=1}^K \beta_k \rho_k$.*

*Proof.* Let us reformulate the mixture as $\log\pi(x) = \log\sum_k \beta_k \exp(-U_k(x))$ for asymptotically strongly convex functions $\mathbf{U} = \{U_k\}_{k=1}^K$. The gradient is

$$
\nabla\log\pi = \mathfrak{J}^T\mathbf{p}, \quad \mathfrak{J} = -\begin{bmatrix} \nabla U_1 \\ \vdots \\ \nabla U_K \end{bmatrix}, \quad \mathbf{p} = \mathrm{softmax}(\log\boldsymbol{\beta} - \mathbf{U})
$$

If each $U_k$ of mixture satisfy Eq. (27) with $\alpha_{U_k}$ and $\tilde{g}_{U_k}$, there exist $\alpha_U = \min_{1 \le k \le K} \alpha_{U_k}$ and $\tilde{g}_U(r) = -r \log \sum_{k=1}^{K} \exp(-r^{-1} \tilde{g}_{U_k})$ that satisfies the condition (27) for $U = -\log \pi$. By direct calculation, one can easily see that soft min-like property of $r^{-1} \tilde{g}_U$ from $\{r^{-1} \tilde{g}_{U_k}\}_{k=1}^{K}$ does not change the conditions of $\tilde{\mathcal{G}}$. □

**General assumptions and justifications.** We additionally need the following general assumptions for our OMD framework. ① (Existence) The sequence of MD from Eq. (13) exists $\{\pi_t\}_{t \in \mathbb{N}} \subset \mathcal{C}$, and are unique, ② (Relative smoothness/convexity) For some $l, L \ge 0$, the functional $F_t$ is $L$-smooth and $l$-strongly-convex relative to $\Omega$. ③ (Existence of first variations) For each $t \ge 0$, the first variation $\delta_c \Omega(\pi_t)$ exists. ④ (Boundedness of estimations) The asymptotic dual mean $\pi_{\mathcal{D}}^\circ$ is almost surely bounded $\Pr(D_\Omega(\pi_t \| \pi_{\mathcal{D}}^\circ) \le R) = 1$ for some $R > 0$. ⑤ (Ergodicity) The estimation process of $\{\pi_t^\circ\}_{t=1}^\infty$ is governed by a measure-preserving transformation on a measure space $(\mathcal{Y}, \Sigma, \varsigma)$ with $\varsigma(\mathcal{Y}) = 1$; for every event $E \in \Sigma$, $\varsigma(T^{-1}(E) \triangle E) = 0$ (that is, $E$ is invariant), either $\varsigma(E) = 0$ or $\varsigma(E) = 1$ (Cornfeld et al., 2012).[3] For ①, the temporal cost $F_t(\cdot) = \mathrm{KL}(\cdot | \pi_t^\circ)$ is well defined since KL is a strong Bregman divergence with lower semicontinuity, where the existence of a primal solution in guaranteed as discussed in Aubin-Frankowski et al. (2022). For ②-③, we can identify $l = L = 1$ and close-form expression of the first variation that is shown in Definition 6 and Proposition 3. For the assumptions ④-⑤, we postulate the existence of estimates produced from a Monte-Carlo method, using a fixed amount of updates on topological vector space. Hence, it is natural to consider that these estimates will be bounded in a probabilistic sense and yield Markovian transitions, which are aperiodic and irreducible.

### A.1 Proofs of Lemmas 2 and 3

The EOT in Eq. (2) can be reformulated as a divergence minimization problem with respective to a reference measure. If a Gibbs parameterization is enforced with the quadratic cost functional $c_\varepsilon(x,y) = \frac{1}{2\varepsilon} \|x - y\|^2$ for $\varepsilon > 0$, it is well-known that the problem has the equivalence with the entropy regularized optimal transport problem (Nutz, 2021)

$$\mathrm{OT}_\varepsilon(\mu, \nu) = \inf_{\pi \in \Pi(\mu,\nu)} \mathrm{KL}\big(\pi \| e^{-c_\varepsilon} \mu \otimes \nu\big). \tag{28}$$

Note that the above equation corresponds to the constrained minimization of $\mathrm{KL}(\mathcal{T} \| W^\varepsilon)$ in Eq. (6) by the disintegration theorem of Schrödinger bridge (Appendix A of Vargas et al., 2021). While the Bregman projection formulation of Sinkhorn Eq. (14) are described by the spaces $(\Pi_\mu^\perp, \Pi_\nu^\perp)$, it is (equally) natural to think that considering the problem as convex problem with the distributional constraint $\mathcal{C}$ (see the primal space in illustrated in Fig. 1). As a problem in the constraint $\mathcal{C}$, one can consider a temporal cost functional $\widetilde{F}_t(\pi) := a_t \mathrm{KL}(\gamma_1 \pi \| \mu) + (1 - a_t) \mathrm{KL}(\gamma_2 \pi \| \nu)$ with sequences $\{a_t\}_{t=1}^\infty = \{0, 1, 0, 1, \dots\}$ for $\gamma_1 \pi(x) := \int \pi(x,y) \mathrm{d}y$ and $\gamma_2 \pi(y) := \int \pi(x,y) \mathrm{d}x$. By construction, we have the following MD update:

$$\underset{\pi \in \mathcal{C}}{\mathrm{minimize}} \langle \delta_c \widetilde{F}_t(\pi_t), \pi - \pi_t \rangle + D_\Omega(\pi \| \pi_t). \tag{29}$$

The optimization problem (29) is equivalent to having the property for subsequent $\pi_{t+1}$:

$$d^+ \widetilde{F}_t(\pi_t; \pi - \pi_t) + D_\Omega(\pi \| \pi_t) \ge d^+ \widetilde{F}_t(\pi_t; \pi_{t+1} - \pi_t) + D_\Omega(\pi_{t+1} | \pi_t)$$
$$\iff \langle \delta_c \widetilde{F}_t(\pi_t) - \delta_c \Omega(\pi_t), \pi - \pi_{t+1} \rangle + \big(\Omega(\pi) - \Omega(\pi_{t+1})\big) \ge 0, \quad \forall \pi \in \mathcal{C}. \tag{30}$$

Setting the free parameter $\pi = \pi_{t+1} + h(\pi - \pi_{t+1})$ and taking the limit $h \to 0^+$ yields described the time evolution of the log-Schrödinger potentials for $\pi_t = e^{\varphi_t \oplus \psi_t - c_\varepsilon} \mathrm{d}(\mu \otimes \nu)$:

$$\dot{\varphi}_t = -\log \frac{\mathrm{d}(\gamma_1 \pi_t)}{\mathrm{d}\nu_*} = -\alpha \bigg( \varphi_t - \varphi^* + \log \int_{\mathbb{R}^d} e^{\psi_t - \psi^*} \nu(\mathrm{d}y) \bigg), \tag{31a}$$

$$\dot{\psi}_t = -\log \frac{\mathrm{d}(\gamma_2 \pi_t)}{\mathrm{d}\mu_*} = -\beta \bigg( \psi_t - \psi^* + \log \int_{\mathbb{R}^d} e^{\varphi_t - \varphi^*} \mu(\mathrm{d}x) \bigg), \tag{31b}$$

---

[3]Here, $\triangle$ denotes the symmetric difference, equivalent to the exclusive-or with respect to set membership.

for $\alpha = a_t$ and $\beta = 1 - a_t$.[4] Setting a discrete approximation of dynamics Eq. (31): $\varphi_{t+1} = \varphi_t + \dot{\varphi}_t$ and $\psi_{t+1} = \psi_t + \dot{\psi}_t$ yields the following alternating updates:

$$\psi_{2t+1}(y) = -\log \int_{\mathbb{R}^d} e^{\varphi_{2t}(x) - c_\varepsilon(x,y)} \mu(\mathrm{d}x), \quad \varphi_{2t+2}(x) = -\log \int_{\mathbb{R}^d} e^{\psi_{2t+1}(x) - c_\varepsilon(x,y)} \nu(\mathrm{d}y).$$

Therefore, the proof of Lemma 2 is complete.

From the dual iteration of KL stated in Eq. (44) (Aubin-Frankowski et al., 2022), the static, idealized MD cost $F(\cdot) = \mathrm{KL}(\cdot \| \pi^*)$ yield the following closed-form expression for the first variation:

$$\delta_c \Omega(\pi_t) - \delta_c \Omega(\pi_{t+1}) = \eta_t \big( \delta_c \Omega(\pi_t) - \delta_c \Omega(\pi^*) \big),$$

where the equation implies that setting $\eta_t \equiv 1$ for MD yields one-step optimality $\pi^*$ in this idealized condition. Utilizing the equivalence of first variation stated in Lemma 6 and the disintegration theorem for the Radon-Nikodym derivatives, we get the first variation of $F$ with respect to $\pi$ for all $x$ as

$$\delta F(\pi_t) = \log \frac{d\pi^*}{d\pi}. \tag{32}$$

And by the disintegration theorem (Léonard, 2014), we also achieve the first variation of $f$ with respect to $\vec{\pi}$ for all $x$ as

$$\delta f(\vec{\pi}_t^x) = \log \frac{d(\vec{\pi}_t^*)^x}{d\vec{\pi}^x}, \tag{33}$$

where $f(\vec{\pi}^x) = \mathrm{KL}(\vec{\pi}^x \| (\vec{\pi}^*)^x)$. Since this disintegration theorem always hold for every directional derivative, we can use expression for $\vec{\pi}^x$ and $\pi$ interchangeably. It is well-known that MD is a discretization of natural gradient descent (Gunasekar et al., 2020), and our setting for $\Omega$ generates the geometry governed by the (generalized) Fisher information. In this particular case, one can use Otto's formalization of Riemannian calculus (Otto, 2001; § 3.2), and the probability space equipped with the Wasserstein-2 metric $(\mathcal{P}_2(\mathbb{R}^d), W_2)$, is generally represented as a Wasserstein gradient flow

$$\partial_t \pi_t = -\nabla_{\mathbb{W}} F(\pi_t), \qquad \forall \pi_t \in \mathcal{C}, \tag{34}$$

where $\nabla_{\mathbb{W}}$ denotes the Wasserstein-2 gradient operator $\nabla_{\mathbb{W}} := \nabla \cdot \big( \rho \nabla \frac{\delta}{\delta \rho} \big)$. In particular, plugging Eq. (32) yields

$$\partial_t \pi_t = -\nabla \cdot (\pi \nabla \log \pi^*) + \Delta \pi, \tag{35}$$

where $\Delta$ denotes the Laplace operator. The foundational results concerning Wasserstein gradients were initially established by JKO (Jordan et al., 1998), who demonstrated that the formulation in Eq. (34) corresponds precisely to the Fokker–Planck equation (35). Consequently, it follows that Wasserstein gradients characterize the tangential direction of flows on a manifold constrained by distributional properties and endowed with the $W_2$ metric. □

## A.2 Proof of Theorem 1

We start with introducing basic properties of the Bregman divergence in Definition 3. First, the *idempotence* property states that a Bregman divergence associated with another Bregman divergence $D_\Omega(\cdot | y)$ remains as the identical divergence with the original. Note that the (global or universal) idempotence initially stated by Aubin-Frankowski et al. (2022), but we apply some changes to the statement and only work with localized version of idempotence for the purpose of this paper.

**Lemma 5** (Idempotence). *Suppose a convex potential $\Omega : \mathcal{M}(\mathcal{X}) \to \mathbb{R} \cup \{+\infty\}$, where $\mathcal{M}(\mathcal{X})$ denotes a topological vector space for $\mathcal{X}$. Assume that for all $z \in \mathrm{dom}(\Omega)$, $\delta_c \Omega(z)$ exists. Then, $\forall x, y \in \mathcal{C} \cap \mathrm{dom}(\Omega)$: $D_{D_\Omega(\cdot | y)}(x | y) = D_\Omega(x | y)$.*

---

[4]More precisely, one needs to apply Lemma 6 for KL, and the disintegration theorem to get Eq. (31).

*Proof of Lemma 5.* Both Bregman divergences and Bregman potentials are convex functionals. By definition, we have $D_{D_\Omega(\cdot|z)}(x|y) = D_\Omega(x\|z) - D_\Omega(y\|z) - \langle \delta_c\Omega(y) - \delta_c\Omega(z), \, x - y \rangle$ for arbitrary $z$, and setting $z = y$ completes the proof. Another (informal) point of view is considering the Bregman divergence as a first-order approximation of a Hessian structure, and $D_{D_\Omega(\cdot|z)}$ converges to $D_\Omega(\cdot|z)$ by taking a limit, knowing that $D_\Omega(y|y) = 0$. □

We then proceed to the equivalence property of the family of recursive Bregman divergences. The property is important for proving the theorem and representing the dual representation of MD. Moreover, it is also used in Theorem 2 as a key ingredient which constructs our VOMD framework.

**Lemma 6** (Equivalence of first variations). *Suppose $\Omega : \mathcal{M}(\mathcal{X}) \to \mathbb{R} \cup \{+\infty\}$ Assume that for all $z \in \mathrm{dom}(\Omega)$, the first variation $\delta_c\Omega(z)$ exists, then, for all $x, y, y_1, y_2 \in \mathrm{dom}(\Omega)$, the first variation taken for the first argument $x$ of the following Bregman divergences are equivalent: $\delta_c D_\Omega(x|y) = \delta_c D_{D_\Omega(\cdot|y_1)}(x|y) = \delta_c D_{D_\Omega(\cdot|y_2)}(x|y)$.*

*Proof of Lemma 6.* First, it can be analytically driven $\delta_c D_\Omega(x|y) = \delta_c\Omega(x) - \delta_c\Omega(y)$. Next, by definition, taking the first variation of $D_{D_\Omega(\cdot|z)}(x|y)$ with respect to $x$ for arbitrary $z \in \mathrm{dom}(\Omega)$ yields $\delta_c D_\Omega(x\|z) - \delta_c\langle\Omega(y) - \Omega(z), x - y\rangle$. Knowing that the second term $\delta_c\langle\Omega(y) - \Omega(z), x - y\rangle$ is linear, we achieve $\delta D_{D_\Omega(\cdot|z)}(x|y) = \delta_c\Omega(x) - \delta_c\Omega(z) - (\delta_c\Omega(y) - \delta_c\Omega(z)) = \delta_c\Omega(x) - \delta_c\Omega(y)$, which completes the proof. □

By an inductive reasoning, we arrive at the basic property of family of Bregman divergences, that all divergence recursively defined by the Bregman potential $\Omega$, has the (local) idempotence and the (global) equivalence of first variation. To address characteristics for particular Bregman potential $\Omega$, we apply the notions of relative smoothness and convexity with respect to $\Omega$, which was first introduced by Birnbaum et al. (2011).

**Definition 6** (Relative smoothness and convexity). Let $G : \mathcal{M}(\mathcal{X}) \to \mathbb{R} \cup \{+\infty\}$ be a proper convex functional. Given scalar $l, L \geq 0$, we define that $G$ is $L$-smooth and $l$-strongly-convex relative to $\Omega$ over $\mathcal{C}$ if for every $x, y \in \mathrm{dom}(G) \cap \mathrm{dom}(\Omega) \cap \mathcal{C}$, we have

$$D_G(x\|y) \leq L D_\Omega(x\|y), \qquad D_G(x\|y) \geq l D_\Omega(x\|y),$$

respectively, where $D_G$ and $D_G$ are Bregman divergences associated with $G$ defined in Definition 3.

Applying the idempotence lemma Lemma 5, we immediately recognize that the Bregman divergence $D_\Omega$ is relatively 1-smooth and 1-strongly-convex for $\Omega$. To start our analysis, we reintroduce the well-known three-point identity for a Bregman divergence.

**Lemma 7** (Three-point identity). *For all $\pi_a, \pi_b, \pi_c \in \mathcal{C} \cap \mathrm{dom}(\Omega)$, we have the following identity*

$$\langle \delta_c\Omega(\pi_a) - \delta_c\Omega(\pi_b), \pi_c - \pi_b \rangle = D_\Omega(\pi_c\|\pi_b) - D_\Omega(\pi_c\|\pi_a) + D_\Omega(\pi_b\|\pi_a)$$

*when $D_\Omega$ is the Bregman divergence defined in Definition 3.*

*Proof of Lemma 7.* By the definition of Bregman divergence, we have

$$\begin{aligned}
D_\Omega(\pi_c\|\pi_b) - D_\Omega(\pi_c\|\pi_a) + D_\Omega(\pi_b\|\pi_a) &= \Omega(\pi_c) - \Omega(\pi_b) - \langle \delta_c\Omega(\pi_b), \pi_c - \pi_b \rangle \\
&\quad - \Omega(\pi_c) + \Omega(\pi_a) + \langle \delta_c\Omega(\pi_a), \pi_c - \pi_a \rangle \\
&\quad + \Omega(\pi_b) - \Omega(\pi_a) - \langle \delta_c\Omega(\pi_a), \pi_b - \pi_a \rangle \\
&= \langle \delta_c\Omega(\pi_a) - \delta_c\Omega(\pi_b), \pi_c - \pi_b \rangle.
\end{aligned}$$

Therefore, the proof is complete. □

Utilizing the three-point identity, we present the following useful lemmas for dealing inequalities regarding improvements by Han et al. (2022), which we call left and right Bregman differences.

**Lemma 8** (Left Bregman difference). *For all $\pi_a, \pi_b, \pi_c \in \mathcal{C} \cap \text{dom}(\Omega)$, the following identity holds.*

$$D_\Omega(\pi_b \| \pi_a) - D_\Omega(\pi_c \| \pi_a) = -\langle \delta_c \Omega(\pi_c) - \delta_c \Omega(\pi_a), \pi_c - \pi_b \rangle + D_\Omega(\pi_b \| \pi_c). \tag{36}$$

*Proof of Lemma 8.* Using Lemma 7, we have

$$D_\Omega(\pi_b \| \pi_a) - D_\Omega(\pi_c \| \pi_a) = -D_\Omega(\pi_c \| \pi_b) + \langle \delta_c \Omega(\pi_a) - \delta_c \Omega(\pi_b), \pi_c - \pi_b \rangle.$$

Utilizing an identity of two Bregman divergences for arbitrary $(\rho, \bar{\rho})$:

$$D_\Omega(\rho \| \bar{\rho}) + D_\Omega(\bar{\rho} \| \rho) = \langle \delta_c \Omega(\rho) - \delta_c \Omega(\bar{\rho}), \rho - \bar{\rho} \rangle. \tag{37}$$

We separate $\delta_c \Omega(\pi_a) - \delta_c \Omega(\pi_b)$ into $\delta_c \Omega(\pi_a) - \delta_c \Omega(\pi_c)$ and $\delta_c \Omega(\pi_c) - \delta_c \Omega(\pi_b)$ and write the rest of the derivation as follows.

$$D_\Omega(\pi_b \| \pi_a) - D_\Omega(\pi_c \| \pi_a)$$
$$= \underbrace{-D_\Omega(\pi_c \| \pi_b) + \langle \delta_c \Omega(\pi_c) - \delta_c \Omega(\pi_b), \pi_c - \pi_b \rangle}_{\text{Eq. (37)}} + \langle \delta_c \Omega(\pi_a) - \delta_c \Omega(\pi_c), \pi_c - \pi_b \rangle$$
$$= D_\Omega(\pi_b \| \pi_c) + \langle \delta_c \Omega(\pi_a) - \delta_c \Omega(\pi_c), \pi_c - \pi_b \rangle$$

Therefore, we achieve the desired identity. □

**Lemma 9** (Right Bregman difference). *For all $\pi_a, \pi_b, \pi_c$, the following identity holds.*

$$D_\Omega(\pi_c \| \pi_b) - D_\Omega(\pi_c \| \pi_a) = D_\Omega(\pi_a \| \pi_b) + \langle \delta_c \Omega(\pi_a) - \delta_c \Omega(\pi_b), \pi_c - \pi_a \rangle \tag{38}$$

*Proof of Lemma 9.* By Lemma 7, we have

$$D_\Omega(\pi_c \| \pi_b) - D_\Omega(\pi_c \| \pi_a) = -D_\Omega(\pi_b \| \pi_a) + \langle \delta_c \Omega(\pi_a) - \delta_c \Omega(\pi_b), \pi_c - \pi_b \rangle.$$

We separate $\pi_c - \pi_b$ into $\pi_c - \pi_a$ and $\pi_a - \pi_b$ and write the rest of the derivation as follows.

$$D_\Omega(\pi_c \| \pi_b) - D_\Omega(\pi_c \| \pi_a)$$
$$= \underbrace{-D_\Omega(\pi_b \| \pi_a) + \langle \delta_c \Omega(\pi_a) - \delta_c \Omega(\pi_b), \pi_a - \pi_b \rangle}_{\text{Eq. (37)}} + \langle \delta_c \Omega(\pi_a) - \delta_c \Omega(\pi_b), \pi_c - \pi_a \rangle$$
$$= D_\Omega(\pi_a \| \pi_b) + \langle \delta_c \Omega(\pi_a) - \delta_c \Omega(\pi_b), \pi_c - \pi_a \rangle$$

Therefore, we achieve the desired identity. □

Additionally, we introduce the three-point inequality (Chen & Teboulle, 1993), which has been a key statement for proving MD convergence for a static cost functional (Aubin-Frankowski et al., 2022), and OMD improvement for temporal costs. The proof mostly follows Aubin-Frankowski et al. (2022) with a slight change of notation.

**Lemma 10** (Three-point inequality). *Given $\pi \in \mathcal{M}(\mathcal{X})$ and some proper convex functional $\Psi : \mathcal{M}(\mathcal{X}) \to \mathbb{R} \cup \{+\infty\}$, if $\delta_c \Omega$ exists, as well as $\bar{\rho} = \arg\min_{\rho \in \mathcal{C}} \{\Psi(\rho) + D_\Omega(\rho \| \pi)\}$, then for all $\rho \in \mathcal{C} \cap \text{dom}(\Omega) \cap \text{dom}(\Psi)$: $\Psi(\rho) + D_\Omega(\rho \| \pi) \geq \Psi(\bar{\rho}) + D_\Omega(\bar{\rho} \| \pi) + D_\Omega(\rho \| \bar{\rho})$.*

*Proof of Lemma 10.* The existence of $\delta_c \Omega$ implies $\mathcal{C} \cap \text{dom}(D_\Omega(\cdot | y)) = \mathcal{C} \cap \text{dom}(\Omega) \cap \text{dom}(\Psi)$. Set $G(\cdot) = \Psi(\cdot) + D_\Omega(\cdot \| y)$. By linearity and idempotence, we have for any $\rho \in \mathcal{C} \cap \text{dom}(\Omega) \cap \text{dom}(\Psi)$

$$D_G(\rho \| \bar{\rho}) = D_\Psi(\rho \| \bar{\rho}) + D_\Omega(\rho \| \bar{\rho}) \geq D_\Omega(\rho \| \bar{\rho}). \tag{39}$$

By $\bar{\rho}$ being the optimality for $G$, for all $x \in \mathcal{C}$,

$$d^+ G(\bar{\rho}; \rho - \bar{\rho}) = \lim_{h \to 0^+} \frac{G((1-h)\bar{\rho} + h\rho) - G(\bar{\rho})}{h} \geq 0,$$

which suggests $G(\rho) \geq G(\bar{\rho}) + D_G(\rho \| \bar{\rho})$. Applying (39) to this inequality complete the proof. □

The following argument is from the convergence rate of mirror descent for relatively smooth and convex pairs of functionals, and extend to infinite dimensional convergence results of Lu et al. (2018) and Aubin-Frankowski et al. (2022). We aim to reformulate the statements in online learning, addressing one-step improvement of OMD.

**Lemma 11** (OMD improvement). *Suppose a temporal cost $F_t : \mathcal{M}(\mathcal{X}) \to \mathbb{R}$ which is $L$-smooth and $l$-strongly-convex relative to $\Omega$ and $\eta_t \leq \frac{1}{L}$. Then, OMD improves for current cost $F_t(\pi_{t+1}) \leq F_t(\pi_t)$.*

*Proof of Lemma 11.* Since $F$ is $L$ relatively smooth, we initially have the inequality

$$F_t(\pi_{t+1}) \leq F_t(\pi_t) + d^+F(\pi_t; \pi_{t+1} - \pi_t) + LD_\Omega(\pi_{t+1}|\pi_t) \tag{40}$$

Applying the three-point inequality (Lemma 10) to Eq. (40), and setting a linear functional $\Psi(\rho) = \eta_t d^+ F_t(\pi_t; \pi - \pi_t)$, $\rho = \pi_t$ and $\bar{\rho} = \pi_{t+1}$ yields

$$d^+F_t(\pi_t; \pi_{t+1} - \pi_t) + \frac{1}{\eta_t}D_\Omega(\pi_{t+1}|\pi_t) \leq d^+F_t(\pi_t; \rho - \pi_t) + \frac{1}{\eta_t}D_\Omega(\rho|\pi_t) - \frac{1}{\eta_t}D_\Omega(\rho\|\pi_{t+1}).$$

Since $F_t$ is assumed to be $l$-strongly convex relative to $\Omega$, we also have

$$d^+F(\pi_t; \rho - \pi_t) \leq F_t(\rho) - F_t(\pi_t) - lD_\Omega(\rho|\pi_t), \tag{41}$$

Then, by using (41), Eq. (40) becomes

$$F_t(\pi_{t+1}) \leq F_t(\rho) + (\frac{1}{\eta_t} - l)D_\Omega(\rho|\pi_t) - \frac{1}{\eta_t}D_\Omega(\rho|\pi_{t+1}) + (L - \frac{1}{\eta_t})D_\Omega(\pi_{t+1}\|\pi_t). \tag{42}$$

By substituting $\rho = \pi_t$, since $D_\Omega(\rho|\pi_{t+1}) \geq 0$ and $L - \frac{1}{\eta_t} \leq 0$, this shows $F_t(\pi_{t+1}) \leq F_t(\pi_t)$, *i.e.*, $F_t$ is decreasing at each iteration. This completes the proof. $\square$

A fundamental property with the dual space $\mathcal{D}$ induced by the first variation $\delta_c$ holds in our OMD setting. The existence of such learning sequence–particularly in Sinkhorn–is well discussed by Nutz (2021) and Aubin-Frankowski et al. (2022). Focusing on the dual geometry, we explicitly call this relationship with arbitrary step size $\eta_t$ as "dual iteration."

**Lemma 12** (Dual iteration). *Suppose that first variations $\delta_c F_t(\pi_t)$ and $\delta_c \Omega(\pi_t)$ exists for $t \geq 0$. Then, online mirror descent updates Eq. (13) is equivalent to $\delta_c \Omega(\pi_{t+1}) - \delta_c \Omega(\pi_t) = -\eta_t \delta_c F_t(\pi_t)$, for all $\pi_t \in \mathcal{C}, t \in \mathbb{N}$.*

*Proof of Lemma 12.* The optimization (13) is equivalent to having the property for subsequent $\pi_{t+1}$:

$$d^+F_t(\pi_t; \pi - \pi_t) + \frac{1}{\eta_t}D_\Omega(\pi\|\pi_t) \geq d^+F_t(\pi_t; \pi_{t+1} - \pi_t) + \frac{1}{\eta_t}D_\Omega(\pi_{t+1}|\pi_t)$$
$$\Longleftrightarrow \langle \delta_c F_t(\pi_t) - \frac{1}{\eta_t}\delta_c \Omega(\pi_t), \pi - \pi_{t+1} \rangle + \frac{1}{\eta_t}\big(\Omega(\pi) - \Omega(\pi_{t+1})\big) \geq 0, \quad \forall \pi \in \mathcal{C}. \tag{43}$$

Setting the free parameter $\pi = \pi_{t+1} + h(\pi - \pi_{t+1})$ and taking the limit $h \to 0^+$ yields the result. $\square$

**Remark 3.** With applications of Lemma 12 and Lemma 6, we can achieve a concise form of iteration in the dual using our temporal cost as:

$$\delta_c \Omega(\pi_t) - \delta_c \Omega(\pi_{t+1}) = \eta_t\big(\delta_c(-H)(\pi_t) - \delta_c(-H)(\pi_t^\circ)\big)$$
$$= \eta_t\big(\delta_c \Omega(\pi_t) - \delta_c \Omega(\pi_t^\circ)\big), \tag{44}$$

where $H$ denotes the entropy, *i.e.*, the minus KL divergence with the Lebesgue measure.

Leveraging the aforementioned lemmas, we have systematically introduced and rigorously formalized the essential concepts necessary to progress with our analysis within the OMD framework. Finally, we are ready to describe a suitable step size scheduling by the following arguments.

**Lemma 13** (Step size I). *Suppose that $F_t = \mathrm{KL}(\pi\|\pi_t^\circ)$ and $\Omega = \mathrm{KL}(\pi\|e^{-c_\varepsilon}\mu \otimes \nu)$. If ① $\lim_{t\to\infty} \eta_t = 0^+$ and ② $\sum_{t=1}^\infty \eta_t = +\infty$ ③ $\eta \leq \frac{1}{L}$, the OMD algorithm converges to a certain $\pi_\mathcal{D}^\circ$*

*Proof of Lemma 13.* From Lemma 11, we have

$$\eta_t(F_t(\pi_{t+1}) - F_t(\pi_t)) \leq -D_\Omega(\pi_t\|\pi_{t+1}) + (\eta_t L - 1)D_\Omega(\pi_{t+1}\|\pi_t). \tag{45}$$

Taking $\lim_{t\to\infty} \eta_t = 0$ ensures improvements; this means for any $\varepsilon > 0$ there exists some $0 < \delta \leq 1$ such that $D_\Omega(\pi_t\|\pi_{t+1}) + D_\Omega(\pi_{t+1}\|\pi_t) < \varepsilon$ whenever $\eta_t < \delta$. Since convexity and the lower semicontinuity of the Bregman divergence $D_\Omega$ induced by KL, we conclude that OMD to a certain point upon the assumed step size scheduling. $\qquad\square$

**Lemma 14** (Step size II). *Assume that $\inf_{\pi \in \mathcal{C}} \mathbb{E}_t[D_\Omega(\pi_t, \pi_t^\circ)] > 0$ for all $t \in [1, \infty)$. Suppose that $\eta_t \to 0$ and $\lim_{T\to\infty} \mathbb{E}[\frac{1}{T}\sum_{t=1}^T D_\Omega(\pi_t\|\pi_t^\circ)] = 0$ if and only if $\sum_{t=1}^\infty \eta_t = +\infty$.*

*Proof of Lemma 14.* We note that due to dual iteration equation Eq. (44), improvements on KL in Lemma 11 are also improvements in the Bregman divergence, *i.e.* $D_\Omega(\pi_{t+1}\|\pi_t^\circ) \leq D_\Omega(\pi_t\|\pi_t^\circ)$, and if $\eta_t \to 0$, then the process $\{\pi_t\}_{t=1}^\infty$ is convergent. By the dominated convergence theorem, assuming ergodicity of non-stationary $\{\pi_t^\circ\}_{t=1}^\infty$ (Cornfeld et al., 2012), there is a constant $\varepsilon$ that satisfies $\mathbb{E}_{1:t+1}[D_\Omega(\pi_{t+1}\|\pi_{t+1}^\circ)] \geq \mathbb{E}_{1:t+1}[D_\Omega(\pi_{t+1}\|\pi_t^\circ)] + \varepsilon$ for $t > n$ for some $n$ as $\eta_t \to 0$, where an expectation subscripted by "$1:t$" indicates the notation of time-averaging from 1 to $t$. Consequently, we achieve the following inequality

$$
\begin{aligned}
\mathbb{E}_{1:t+1}&[D_\Omega(\pi_{t+1}\|\pi_{t+1}^\circ)] \\
&\geq \mathbb{E}_{1:t+1}[D_\Omega(\pi_{t+1}\|\pi_t^\circ)] + \varepsilon \\
&\geq \mathbb{E}_{1:t}\big[D_\Omega(\pi_t\|\pi_t^\circ) - \langle \delta_c\Omega(\pi_{t+1}) - \delta_c\Omega(\pi_t), \pi_t^\circ - \pi_t\rangle\big] + \mathbb{E}_{1:t+1}\big[D_\Omega(\pi_{t+1}\|\pi_t)\big] + \varepsilon &&\text{Lemma 8} \\
&= \mathbb{E}_{1:t}\big[D_\Omega(\pi_t\|\pi_t^\circ) - \eta_t D_\Omega(\pi_t\|\pi_t^\circ) + \eta_t D_\Omega(\pi_t^\circ\|\pi_t)\big] + \mathbb{E}_{1:t+1}\big[D_\Omega(\pi_{t+1}\|\pi_t)\big] + \varepsilon &&\text{Eq. (44)} \\
&= (1-\eta_t)\mathbb{E}_{1:t}\big[D_\Omega(\pi_t\|\pi_t^\circ)\big] + \mathbb{E}_{1:t+1}\big[D_\Omega(\pi_{t+1}\|\pi_t) + \eta_t D_\Omega(\pi_t^\circ\|\pi_t)\big] + \varepsilon \\
&\geq (1-\eta_t)\mathbb{E}_{1:t}\big[D_\Omega(\pi_t\|\pi_t^\circ)\big] + \varepsilon'
\end{aligned}
\tag{46}
$$

for some $t$ and $0 < \varepsilon < \varepsilon'$, where Lemma 8 and Eq. (44) are used.

*Necessity.* For big enough $t \geq n$ where $n \in \mathbb{N}$, we can achieve the inequality in Eq. (46) as

$$\mathbb{E}_{1:t+1}\big[D_\Omega(\pi_{t+1}\|\pi_{t+1}^\circ)\big] \geq (1-\eta_t)\mathbb{E}_{1:t}\big[D_\Omega(\pi_t\|\pi_t^\circ)\big], \tag{47}$$

Since we have assumed that $\eta_t$ converges to 0, consider a step size sequence $0 < \eta_t \leq \frac{2}{2+k}$ for $k > 0$. Denote a constant $a = \frac{2+k}{2}\log\frac{2+k}{k}$ and apply the elementary inequality (Lei & Zhou, 2020)

$$1 - x \geq \exp(-ax), \quad \text{such that} \quad 0 < x \leq \frac{2}{2+k}.$$

From Eq. (47), we achieve

$$\mathbb{E}_{1:t+1}\big[D_\Omega(\pi_{t+1}\|\pi_{t+1}^\circ)\big] \geq \exp(-a\eta_t)\mathbb{E}_{1:t}\big[D_\Omega(\pi_t\|\pi_t^\circ)\big].$$

for all $t \geq n$. Iteratively applying this inequality iterative for $t = n, n+1, \cdots, T-1$ gives

$$
\begin{aligned}
\mathbb{E}_{1:T}[D_\Omega(\pi_T\|\pi_T^\circ)] &\geq \mathbb{E}_{1:n}[D_\Omega(\pi_n\|\pi_n^\circ)]\prod_{t=n}^{T-1}\exp(-a\eta_t) \\
&= \exp\left\{-a\sum_{t=n}^{T-1}\eta_t\right\}\mathbb{E}_{1:n}[D_\Omega(\pi_n\|\pi_n^\circ)].
\end{aligned}
\tag{48}
$$

From the assumption $\pi^* \neq \pi_n$, $D_\Omega(\pi_n\|\pi_n^\circ) > 0$ by the property of divergence. Therefore, by Eq. (48), the convergence $\lim_{t\to\infty}\mathbb{E}_{1:t}[D_\Omega(\pi_t\|\pi_t^\circ)] = 0$ implies the series $\sum_{t=1}^\infty \eta_t$ diverges to $+\infty$ so that $\exp(-a\sum_{t=n}^{T-1}\eta_t)$ converges to 0.

*Sufficiency.* Consider a static Schrödinger bridge problem with couplings $\pi \in \Pi(\mu, \nu)$, which is in a constraint set

$$\mathcal{C} = \big\{\pi|(\mu,\nu) \in \mathcal{P}_2(\mathbb{R}^d) \cap \mathcal{P}_{\text{alc}}(\mathbb{R}^d), (\varphi, \psi) \in L^1(\mu) \times L^1(\nu), \text{ and } \varphi, \psi \in C^2(\mathbb{R}^d) \cap \text{Lip}(\mathcal{K})\big\}.$$

For $\rho, \bar{\rho} \in \mathcal{C}$, we can see

$$D_\Omega(\bar{\rho}\|\rho) = \Omega(\bar{\rho}) - \Omega(\rho) - \langle \delta_c \Omega(\rho), \bar{\rho} - \rho \rangle \geq 0 \iff -\langle \delta_c \Omega(\rho), \bar{\rho} - \rho \rangle \geq \Omega(\rho) - \Omega(\bar{\rho}).$$

By adding $\langle \delta_c \Omega(\bar{\rho}), \bar{\rho} - \rho \rangle$, we achieve a property:

$$\langle \delta_c \Omega(\rho) - \delta_c \Omega(\bar{\rho}), \rho - \bar{\rho} \rangle \geq D_\Omega(\rho\|\bar{\rho}). \tag{49}$$

Then, suppose that we have the asymptotic dual mean $\pi_\mathcal{D}^\circ$. Using the right Bregman difference Lemma 9, the one-step progress from the perspective of dual mean writes as

$$
\begin{aligned}
D_\Omega(\pi_\mathcal{D}^\circ\|\pi_{t+1}) - D_\Omega(\pi_\mathcal{D}^\circ\|\pi_t) &= \langle \delta_c \Omega(\pi_t) - \delta_c \Omega(\pi_{t+1}), \pi_\mathcal{D}^\circ - \pi_t \rangle + D_\Omega(\pi_t\|\pi_{t+1}). \\
&= \eta_t \langle \delta_c \Omega(\pi_t) - \delta_c \Omega(\pi_t^\circ), \pi_\mathcal{D}^\circ - \pi_t \rangle + D_\Omega(\pi_t\|\pi_{t+1}) \\
&= \eta_t \langle \delta_c \Omega(\pi_t) - \delta_c \Omega(\pi_\mathcal{D}^\circ), \pi_\mathcal{D}^\circ - \pi_t \rangle + \eta_t \langle \delta_c \Omega(\pi_\mathcal{D}^\circ) - \delta_c \Omega(\pi_t^\circ), \pi_t^\circ - \pi_t \rangle + D_\Omega(\pi_t\|\pi_{t+1}) \\
&\leq -\eta_t D(\pi_\mathcal{D}^\circ\|\pi_t) + \eta_t \langle \delta_c \Omega(\pi_\mathcal{D}^\circ) - \delta_c \Omega(\pi_t^\circ), \pi_t^\circ - \pi_t \rangle + D_\Omega(\pi_t\|\pi_{t+1})
\end{aligned}
$$

where the inequality is from Eq. (49). By applying the definition of $\pi_\mathcal{D}^\circ$ and ergodicity of $\{\pi_t^\circ\}_{t=1}^\infty$, we can bound the expectation by finding some $t > n$ such that

$$
\begin{aligned}
\mathbb{E}_{1:t+1}\big[D_\Omega(\pi_\mathcal{D}^\circ\|\pi_{t+1})\big] &\leq \mathbb{E}_{1:t}\big[(1-\eta_t)D_\Omega(\pi_\mathcal{D}^\circ\|\pi_t)\big] + \mathbb{E}_{1:t+1}\big[D_\Omega(\pi_t\|\pi_{t+1})\big] \\
&\leq \mathbb{E}_{1:t}[(1-\eta_t)D_\Omega(\pi_\mathcal{D}^\circ\|\pi_t)] + \frac{1}{2\omega}\mathbb{E}_{1:t+1}\big[\big\|\nabla(\delta_c \Omega(\pi_t) - \delta_c \Omega(\pi_{t+1}))\big\|^2_{L^2(\pi_t)}\big] \\
&\leq \mathbb{E}_{1:t}[(1-\eta_t)D_\Omega(\pi_\mathcal{D}^\circ\|\pi_t)] + \frac{\eta_t^2}{2\omega}\mathbb{E}_{1:t}\big[\big\|\nabla(\delta_c \Omega(\pi_t) - \delta_c \Omega(\pi_t^\circ))\big\|_{L^2(\pi_t)}\big] \\
&\leq \mathbb{E}_{1:t}[(1-\eta_t)D_\Omega(\pi_\mathcal{D}^\circ\|\pi_t)] + 2\eta_t^2 \omega^{-1}\mathcal{K}, \tag{50}
\end{aligned}
$$

where $\mathcal{K}$ is the Lipschitz constant for each log Schrödinger potential in $\mathcal{C}$. For the first inequality, we use Assumption 2, and we use the log Sobolev inequality $\mathrm{LSI}(\omega)$ from Assumption 1 in the second inequality. Let $\{A_t\}_{t=1}^\infty$, denote a sequence of $A_t = \mathbb{E}_{1:t}[D_\Omega(\pi_\mathcal{D}^\circ\|\pi_t)]$. As a result, we have

$$A_{t+1} \leq (1-\eta_t)A_t + z\eta_t^2, \quad \forall t > n, \tag{51}$$

where $z := 2\omega^{-1}\mathcal{K}$. For a constant $h > 0$, we argue that $A_{t_1} < h$ for some $t_1 > n'$. Suppose that this statement is *not* true; we find some $t \geq t_1$ such that $A_t > h$, $\forall t \geq t_2$. Since $\lim_{t\to\infty} \eta_t = 0$, there are some $t > t_3 > t_2$ that $\eta_t \leq \frac{h}{4}$. However, Eq. (51) tells us that for $t \geq t_3$, for $t \geq t_3$,

$$A_{t+1} \leq (1-\eta_t)A_t + z\eta_t^2 \leq A_{t_3} - \frac{h}{4}\sum_{k=t_3}^T \eta_k \to -\infty \quad (\text{as } t \to \infty).$$

This results to a contradiction, which verifies $A_t < h$ for $t > n'$. Since $\lim_{t\to\infty} \eta_t = 0$, we can find some $\eta_t$ which makes $A_t$ monotonically decreasing. Therefore, we conclude the nonnegative sequence $\{A_t\}_{t=1}^\infty$ finds convergence by iteratively applying the upper bound in Eq. (51). $\qquad\square$

We now prove Theorem 1 under consideration of the particular case of $\eta_t = \frac{2}{t+1}$. Then, Eq. (51) becomes

$$A_{t+1} \leq \left(1 - \frac{2}{t+1}\right)A_t + \frac{4z}{(t+1)^2}, \quad \forall t \geq n.$$

It follows that recursive relation writes as

$$t(t+1)A_{t+1} \leq (t-1)tA_t + 4z, \quad \forall t \geq n.$$

Iterative applying the relation, we achieve the following inequality:

$$(T-1)TA_T \leq (n-1)nA_n + 4z(T-n), \quad \forall T \geq n.$$

Therefore, we finally achieve inequality as follows:

$$\mathbb{E}_{1:T}[D_\Omega(\pi_\mathcal{D}^\circ\|\pi_T)] \leq \frac{(n-1)n\mathbb{E}_{1:n}[D_\Omega(\pi_\mathcal{D}^\circ\|\pi_n)]}{(T-1)T} + \frac{4z}{T}, \quad \forall T \geq n. \tag{52}$$

Since we assumed $\pi^* = \pi_\mathcal{D}^\circ$, $\mathbb{E}_{1:T}[D_\Omega(\pi^*\|\pi_T)] = \mathcal{O}(1/T)$. Combining this with Lemmas 13 and 14, the proof of Theorem 1 is complete.

## A.3 Proof of Proposition 1

The proof is based on the Doob's forward convergence theorem.

**Theorem 3** (Doob's forward convergence theorem). *Let $\{X_t\}_{t\in\mathbb{N}}$ be a sequence of nonnegative random variables and let $\{\mathcal{F}_t\}_t$ be a random variable and let $\{\mathcal{F}_t\}_{t\in\mathbb{N}}$ be a filtration with $\mathcal{F}_t \subset \mathcal{F}_{t+1}$ for every $t \in \mathbb{N}$. Assume that $\mathbb{E}[X_{t+1}|\mathcal{F}_t] \leq X_t$ almost surely for every $t \in \mathbb{N}$. Then, the sequence $\{X_t\}$ converges to a nonnegative random variable $X_\infty$ almost surely.*

We follow the derivation of Eq. (50): there exists $n \in \mathbb{N}$ which satisfies

$$\mathbb{E}_t[D_\Omega(\pi_\mathcal{D}^\circ\|\pi_{t+1})] \leq D_\Omega(\pi_\mathcal{D}^\circ\|\pi_t) + 2\eta_t^2\omega^{-1}\mathcal{K}, \quad \forall t \geq n$$

and since the step size is scheduled as $\lim_{t\to\infty}\eta_t = 0$, the condition $\sum_{t=1}^\infty \eta_t^2 < \infty$ enables us to define a stochastic process $\{X_t\}_{t\in\mathbb{N}}$:

$$X_t = D_\Omega(\pi_\mathcal{D}^\circ\|\pi_t) + 2\omega^{-1}\mathcal{K}\sum_{i=t}^\infty \eta_i^2. \tag{53}$$

It is straightforward that the defined random variable satisfies $\mathbb{E}_t[X_{t+1}] \leq X_t$ for $t \geq n$. Since $X_t \geq 0$, the process is a sub martingale. By Theorem 3, the sequence $\{X_t\}_{t\in\mathbb{N}}$ converges to a nonnegative random variable $X_\infty$ almost surely. Therefore $D_\Omega(\pi_\mathcal{D}^\circ\|\pi_t)$ converges to 0 almost surely. □

## A.4 Proof of Proposition 2

To achieve a meaningful regret bound for our problem setup, we first demonstrate the following.

**Lemma 15.** *For all $w = \arg\min_y\{\langle\hat{g}, y\rangle + \frac{1}{\eta}D_\Omega(y\|z)\}$ with $\eta > 0$, the following equation.*

$$\forall u.\langle\eta\hat{g}, w - u\rangle \leq D_\Omega(u\|z) - D_\Omega(u\|w) - D_\Omega(w\|z) \tag{54}$$

*Proof of Lemma 15.* By the first order optimality of $\{\langle g, y\rangle + D_\Omega(y\|z)\}$ as a function of $w$, we have

$$\langle\hat{g} + \frac{1}{\eta}\delta_c D_\Omega(w\|z), u - w\rangle \geq 0$$
$$\implies \langle\hat{g}, w - u\rangle \leq \frac{1}{\eta}\langle-\delta_c D_\Omega(w\|z), w - u\rangle = \frac{1}{\eta}(D_\Omega(u\|z) - D_\Omega(u\|w) - D_\Omega(w\|z)).$$

where used Lemma 8 in the derivation. This completes the proof. □

Next, we derive the one-step relationship for OMD. The result entails that the regret at each step is related to a quadratic expression of $\eta_t$, which is a key aspect of sublinear total regret. From a technical standpoint, we can see that the assumption for log Sobolev inequality generally works as a premise for Lipschitz continuity of gradient, *i.e.*, $\nabla\Omega$ in classical MD analyses.

**Lemma 16** (Single step regret). *Suppose a static Schrödinger bridge problem with the aforementioned constraint $\mathcal{C}$. Let $D_\Omega$ be the Bregman divergence wrt $\Omega : \mathcal{P}(\mathcal{X}) \to \mathbb{R} + \{+\infty\}$. Then,*

$$\eta_t(F_t(\pi_t) - F_t(u)) \leq D_\Omega(u\|\pi_t) - D_\Omega(u\|\pi_{t+1}) + \frac{\eta_t^2}{2\omega}\|\hat{g}_t\|_{L^2(\pi_t)}^2, \quad \forall u \in \mathcal{C} \tag{55}$$

*holds, where $\hat{g}_t := \delta_c F_t(\pi_t) = \frac{1}{\eta_t}(\delta_c\Omega(\pi_t) - \delta_c\Omega(\pi_{t+1}))$ in an MD iteration for the dual space for a step size $\eta_t$, and $\omega > 0$ is drawn from a type of log Sobolev inequality in Assumption 1.*

*Proof of Lemma 16.* Consider single step regrets by the adversary plays of a linearization for $\hat{g}_t$:

$$F_t(\pi_t) - F_t(u) \leq \langle\hat{g}_t, \pi_t - u\rangle.$$

Therefore, we derive a inequality for $\langle\hat{g}_t, \pi_t - u\rangle$ as follows.

$$\begin{aligned}
\langle\eta_t\hat{g}_t, \pi_t - u\rangle &= \langle\eta_t\hat{g}_t, \pi_{t+1} - u\rangle + \langle\eta_t\hat{g}_t, \pi_t - \pi_{t+1}\rangle \\
&\leq D_\Omega(u\|\pi_t) - D_\Omega(u\|\pi_{t+1}) - D_\Omega(\pi_{t+1}\|\pi_t) + \langle\eta_t\hat{g}_t, \pi_t - \pi_{t+1}\rangle \\
&= D_\Omega(u\|\pi_t) - D_\Omega(u\|\pi_{t+1}) - D_\Omega(\pi_{t+1}\|\pi_t) + \langle\delta_c\Omega(\pi_{t+1}) - \delta_c\Omega(\pi), \pi_t - \pi_{t+1}\rangle \\
&= D_\Omega(u\|\pi_t) - D_\Omega(u\|\pi_{t+1}) + D_\Omega(\pi_t\|\pi_{t+1}).
\end{aligned}$$

Since we assumed that $\hat{g}_t = \frac{1}{\eta_t}(\delta_c\Omega(\pi_t) - \delta_c\Omega(\pi_{t+1}))$ by the dual iteration and that Assumption 1 holds, we can achieve the upperbound $D_\Omega(\pi_t\|\pi_{t+1}) \leq \frac{\eta_t^2}{2\omega}\|\hat{g}_t\|_{L^2(\pi_t)}^2$ by direct calculation. $\qquad\square$

We now show our upper bound of total regret by utilizing Lemma 16.

**Lemma 17.** *Assume $\eta_{t+1} \leq \eta_t$. Then, $u \in \mathcal{C}$, the following regret bounds for fixed $u \in \mathcal{C}$ hold*

$$\sum_{t=1}^{T} F_t(\pi_t) - F_t(u) \leq \max_{1 \leq t \leq T} \frac{D_\Omega(u\|\pi_t)}{\eta_T} + \frac{1}{2\omega}\sum_{t=1}^{T}\eta_t\|\tilde{g}_t\|_{L^2(\pi_t)}^2 \tag{56}$$

*where $\hat{g}_t = \frac{1}{\eta_t}(\delta_c\Omega(\pi_t) - \delta_c\Omega(\pi_{t+1}))$.*

*Proof of Lemma 17.* Define $D^2 = \max_{1 \leq t \leq T} D_\Omega(u\|\pi_t)$. We get

$$\text{Regret}(u) = \sum_{t=1}^{T}(F_t(\pi_t) - F_t(u))$$

$$\leq \sum_{t=1}^{T}\left(\frac{1}{\eta_t}D_\Omega(u\|\pi_t) - \frac{1}{\eta_t}D_\Omega(u\|\pi_{t+1})\right) + \sum_{t=1}^{T}\frac{\eta_t}{2\omega}\|\hat{g}_t\|_{L^2(\pi_t)}^2$$

$$= \frac{1}{\eta_1}D_\Omega(u\|\pi_1) - \frac{1}{\eta_T}D_\Omega(u\|\pi_{T+1}) + \sum_{t=1}^{T-1}\left(\frac{1}{\eta_{t+1}} - \frac{1}{\eta_t}\right)D_\Omega(u\|\pi_{t+1}) + \sum_{t=1}^{T}\frac{\eta_t}{2\omega}\|\hat{g}_t\|_{L^2(\pi_t)}^2$$

$$\leq \frac{1}{\eta_1}D^2 + D^2\sum_{t=1}^{T-1}\left(\frac{1}{\eta_{t+1}} - \frac{1}{\eta_t}\right) + \sum_{t=1}^{T}\frac{\eta_t}{2\omega}\|\hat{g}_t\|_{L^2(\pi_t)}^2 = \frac{D^2}{\eta_T} + \sum_{t=1}^{T}\frac{\eta_t}{2\omega}\|\hat{g}_t\|_{L^2(\pi_t)}^2.$$

Therefore, the proof is complete. $\qquad\square$

Following Lemma 17 and Assumption 1, we can have the inequality

$$\sum_{t=1}^{T} F_t(\pi_t) - F_t(u) \leq \frac{D^2}{\eta_T} + \sum_{t=1}^{T}\frac{\eta_t}{2\omega}\|\hat{g}_t\|_{L^2(\pi_t)}^2 \leq \frac{D^2}{\eta_T} + 2\eta_1\omega^{-1}\mathcal{K}T.$$

where $D^2 = \max_{1 \leq t \leq T} D_\Omega(u\|\pi_t)$. Setting a constant step size $\eta_t \equiv \frac{D\sqrt{\omega}}{\sqrt{2\mathcal{K}T}}$ yields an upper bound of $2D\sqrt{2\omega^{-1}\mathcal{K}T}$ which proves the regret bound of $\mathcal{O}(\sqrt{T})$. Also, recall that the following lemma.

**Lemma 18** (Lemma 3.5 of Auer et al., 2002). *Let a sequence $a_1, a_2, \ldots, a_T$ be non-negative real numbers. If $a_1 > 0$, then*

$$\sum_{t=1}^{T}\frac{a_t}{\sqrt{\sum_{i=1}^{t}a_i}} \leq 2\sqrt{\sum_{t=1}^{T}a_t}. \tag{57}$$

Setting a adaptive scheduling $\eta_t = \frac{D\sqrt{\omega}}{\sqrt{2\sum_{i=1}^{t}\|\hat{g}_i\|^2}}$ yields $2D\sqrt{2\omega^{-1}\sum_{t=1}^{T}\|\hat{g}_t\|^2}$ which has a possibility to be lower than $\mathcal{O}(\sqrt{T})$ depending on $\{\pi_t^\circ\}_{t=1}^{T}$. Therefore, we have formally expanded the convergence results of OMD (Lei & Zhou, 2020; Orabona & Pál, 2018; Srebro et al., 2011) to SBPs. $\qquad\square$

### A.5 Proof of Theorem 2

Since $D_\Omega(\cdot\|\cdot) := D_{\text{KL}(\cdot\|\mathcal{R})}(\cdot\|\cdot)$ for a reference measure $\mathcal{R} \in \mathcal{C}$, we can apply Lemma 6 and achieve Eq. (17). We write the following equivalent convex problems, using the equivalence of first variation for recursively

defined Bregman divergences.

$$\langle \delta_c F_t(\pi_t), \pi - \pi_t \rangle + \tfrac{1}{\eta_t} D_\Omega(\pi \| \pi_t) = \langle \delta_c D_\Omega(\pi_t \| \pi_t^\circ), \pi - \pi_t \rangle + \tfrac{1}{\eta_t} D_\Omega(\pi \| \pi_t)$$
$$= \langle \delta_c \Omega(\pi_t) - \delta_c \Omega(\pi_t^\circ), \pi - \pi_t \rangle + \tfrac{1}{\eta_t} D_\Omega(\pi \| \pi_t)$$
$$= D_\Omega(\pi \| \pi_t^\circ) - D_\Omega(\pi \| \pi_t) + \tfrac{1}{\eta_t} D_\Omega(\pi \| \pi_t)$$
$$= \left( \frac{1}{\eta_t} \right) D_\Omega(\pi \| \pi_t^\circ) + \left( \frac{1 - \eta_t}{\eta_t} \right) D_\Omega(\pi \| \pi_t)$$

We refer to Appendix B for the stability of Wasserstein gradient flows according to the LaSalle's invariance principle. We can now interpret $\delta_c \mathcal{E}_t$ as a dynamics that reaches an equilibrium solution

$$\underset{\pi \in \mathcal{C}}{\text{minimize}} \ \langle \delta_c F_t(\pi_t), \pi - \pi_t \rangle + \tfrac{1}{\eta_t} D_\Omega(\pi \| \pi_t) \ \Leftrightarrow \ \underset{\pi \in \mathcal{C}}{\text{minimize}} \ \eta_t \underbrace{D_\Omega(\pi \| \pi_t^\circ)}_{\text{empirical estimates}} + (1 - \eta_t) \underbrace{D_\Omega(\pi \| \pi_t)}_{\text{proximity}},$$

At a glance, the above equation appears analogous to the interpolation search between two points, where the influence of $\pi_t^\circ$ is controlled by $\eta_t$. $\qquad\square$

## A.6 Proof of Proposition 3

The proof is closely related to the work of Lambert et al. (2022) where the difference lies in we correct the Wasserstein gradient term $\dot\alpha_{k,\tau}$ for suitable for generally unbalanced weight. Suppose take parameterization $\theta \in (\mathcal{P}_2(\texttt{BW}(\mathbb{R}^d)), \texttt{WFR})$, the space of Gaussian mixtures equipped with the Wasserstein-Fisher-Rao metric, over the measure space of Gaussian particles. Following the arguments from Appendix B.2 and the studies for this particular GMM problem (Lu et al., 2019; Lambert et al., 2022) of the Wasserstein-Fisher-Rao of the KL functional is derived as

$$\nabla_{\texttt{WFR}} \text{KL}(\rho_\theta \| \rho^*) = \left( \nabla_{\texttt{BW}} \delta \text{KL}(\rho \| \rho^*), \frac{1}{2} \left( \delta \text{KL}(\rho_\theta \| \rho^*) - \int \delta \text{KL}(\rho \| \rho^*) \mathrm{d}\rho \right) \right), \qquad (58)$$

where we can consider the WFR gradient is taken with respect to $\theta$ of its first argument. By Eq. (58), we separately consider Wasserstein gradient in the Bures–Wasserstein space and the space of lighting that controls the amount of each Gaussian particle.

Given a functional $F : \mathcal{P}_2(\mathcal{X}) \to \mathbb{R} \cup \{+\infty\}$, the Wasserstein gradient $\nabla_{\texttt{w}} F \cap T_\rho \mathcal{P}_2(\mathcal{X})$ such that all $\{\rho_t\}_{t \in \mathbb{R}^+}$ satisfy the continuity eqatuion starting from $\rho_0$ (Jordan et al., 1998; Villani, 2021). If the functional is the KL divergence $\text{KL}(\rho \| \pi)$ we can compute the Bures–Wasserstein gradient for the Gaussian distribution with respect to $(m, \Sigma)$ using Eq. (76)

$$\nabla_{\texttt{BW}} F(m, \Sigma) = (\nabla_m F(m, \Sigma), 2 \nabla_\Sigma F(m, \Sigma))$$
$$= \left( \int \nabla_m \rho_{m,\Sigma} \log \frac{\rho_{m,\Sigma}}{\pi}, 2 \int \nabla_\Sigma \rho_{m,\Sigma} \log \frac{\rho_{m,\Sigma}}{\pi} \right),$$

with some abuse of notation for $\rho$. Using the following closed-form identities for the Gaussian distributions

$$\forall x. \quad \nabla_m \rho_{m,\Sigma}(x) = -\nabla_x \rho_{m,\Sigma}(x) \quad \text{and} \quad \nabla_\Sigma \rho_{m,\Sigma}(x) = \frac{1}{2} \nabla_x^2 \rho_{m,\Sigma}(x).$$

and the equivalence between the Hessian and Fisher information, we achieve the following form:

$$\nabla_{\texttt{BW}} F(m, \Sigma) = \left( \mathbb{E}_\rho \left[ \nabla \frac{\rho}{\pi} \right], \mathbb{E}_\rho \left[ \nabla^2 \log \frac{\rho}{\pi} \right] \right).$$

Define $r_{k,\tau} = \sqrt{\alpha_{k,\tau}}$. Since $r_t$ follows the Fisher–Rao metric in Definition 7, by the Proposition A.1 from Lu et al. (2019) and specialization of Lambert et al. (2022), we can think of dynamics of $K$ Gaussian particles $\{\alpha_{k,\tau}, m_{k,\tau}, \Sigma_{k,\tau}\}_{k=1}^{K}$ such that

$$\dot r_{k,\tau} = -\frac{1}{2} \left( \mathbb{E} \left[ \log \frac{\rho_{\theta_\tau}}{\rho^*}(y_{k,\tau}) \right] - \frac{1}{z_\tau} \sum_{\ell=1}^{K} \alpha_\ell \mathbb{E} \left[ \log \frac{\rho_{\theta_\tau}}{\rho^*}(y_{\ell,\tau}) \right] \right) r_{k,\tau},$$

$$\dot m_{k,\tau} = -\mathbb{E} \left[ \nabla \log \frac{\rho_{\theta_\tau}}{\rho^*}(y_{k,\tau}) \right], \quad \dot\Sigma_{k,\tau} = -\mathbb{E} \left[ \nabla^2 \log \frac{\rho_{\theta_\tau}}{\rho^*}(y_{k,\tau}) \right] \Sigma_{k,\tau} - \Sigma_{k,\tau} \mathbb{E} \left[ \nabla^2 \log \frac{\rho_{\theta_\tau}}{\rho^*}(y_{k,\tau}) \right],$$

Since $\alpha_{k,\tau} = \sqrt{r_{k,\tau}}$ by previous definition, it is straightforward that

$$\dot{\alpha}_{k,\tau} = -\left(\mathbb{E}\left[\log\frac{\rho_{\theta_\tau}}{\rho^*}(y_{k,\tau})\right] - \frac{1}{z_\tau}\sum_{\ell=1}^{K}\alpha_\ell\mathbb{E}\left[\log\frac{\rho_{\theta_\tau}}{\rho^*}(y_{\ell,\tau})\right]\right)\alpha_{k,\tau}.$$

For $\alpha_k > 0$. This completes the proof. $\qquad\square$

## B  A Riemannian Perspective on Wasserstein Geometries

### B.1  An introduction to Otto calculus and the LaSalle invariance principle

In this appendix, we introduce a basic notion of Wasserstein gradient flows in the space of continuous probability measures. We focus on describing the particular example, the KL cost, initially studied by JKO (Jordan et al., 1998) and formally generalized by Otto (2001) in the context of Riemannian geometry. For more details and mathematical rigor, we refer the reader to (Ambrosio et al., 2005b; Carrillo et al., 2023). For $\mathcal{X} \subset \mathbb{R}^d$, and functions $U : \mathbb{R}_{\geq 0} \to \mathbb{R}$; $V, W : \mathcal{X} \to \mathbb{R}$. We first consider an energy function $\mathcal{E} : \mathcal{P}_2(\mathcal{X}) \to \mathbb{R}$:

$$\mathcal{E}(\rho) = \underbrace{\int_{\mathcal{X}} U(\rho(x))\,\mathrm{d}x}_{\text{internal potential } \mathcal{U}} + \underbrace{\int_{\mathcal{X}} V(x)\,\mathrm{d}\rho(x)}_{\text{external potential } \mathcal{E}_V} + \underbrace{\frac{1}{2}\int_{\mathcal{X}}(W*\rho)(x)\,\mathrm{d}\rho(x)}_{\text{interaction energy } \mathcal{W}}, \quad \rho \in \mathcal{P}_2(\mathcal{X}). \tag{59}$$

For this function, we refer to the solution of the following PDE:

$$\partial_t\rho_t = \nabla\cdot\left[\rho\,\nabla(U' + V + W*\rho)\right], \qquad t \geq 0 \tag{60}$$

as the Wasserstein gradient flow of $\mathcal{E}$. Following Otto's formalization of Riemannian calculus on the continuous probability space equipped with the Wasserstein metric $(\mathcal{P}_2(\mathcal{X}), W_2)$, the PDE (60) can be interpreted close to an ODE of Riemannian gradient flow:

$$\partial_t\rho_t = -\nabla_{\mathrm{w}}\mathcal{E}(\rho), \tag{61}$$

where $\nabla_{\mathrm{w}}$ denotes the Wasserstein-2 gradient operator $\nabla_{\mathrm{w}} := \nabla\cdot\left(\rho\,\nabla\frac{\delta}{\delta\rho}\right)$. Considering the Otto's Wasserstein-2 Riemannian metric $\mathfrak{g}$ (Otto, 2001; Lott, 2008), under the absolute continuity, we see that

$$\frac{\partial}{\partial t}\mathcal{E}(\rho_t) = -\mathfrak{g}_\rho\left(\frac{\partial\rho}{\partial t}, \frac{\partial\rho}{\partial t}\right) = -\int_{\mathcal{X}}\left|\nabla(U' + V + W*\rho)\right|^2\mathrm{d}\rho(x) \leq 0, \tag{62}$$

which is closely related to the strict Lyapunov condition. As a result, dynamical systems following the PDE are guaranteed to reach an equilibrium solution, under the LaSalle invariance principle for probability measures (Carrillo et al., 2023).

For a representative example, we identify Eq. (59) for the relative entropy (the KL functional) for a target density $\rho^* \in \mathcal{P}_2(\mathcal{X})$ writes

$$\mathcal{E}(\rho) = \mathrm{KL}(\rho\|\rho^*) = \underbrace{\int_{\mathcal{X}} U(\rho(x))\,\mathrm{d}x}_{\mathcal{U}} + \underbrace{\int_{\mathcal{X}} V(x)\,\mathrm{d}\rho(x)}_{\mathcal{E}_V} - C,$$

where $U(s) = s\log s$, $V(x) = -\log\rho^*(x)$, and $C = \mathcal{U}(\rho^*) + \mathcal{E}_V(\rho^*)$. Recall that $\delta\mathcal{E}(\rho) = \log\frac{\rho(x)}{\rho^*}$, then we have

$$\nabla_{\mathrm{w}}\mathcal{E}(\rho) = \mathfrak{G}_\rho^{-1}\delta E(\rho) = -\nabla\cdot[\rho\nabla\delta E(\rho)] = \nabla\cdot\left[\rho\nabla\log\frac{\rho}{\rho_*}\right] \tag{63}$$

where $\mathfrak{G}$ denotes the metric tensor in matrix form. We can derive the the Fokker–Planck equation

$$\partial_t\rho_t = -\nabla\cdot(\rho\nabla\log\rho^*) + \Delta\rho_t,$$

describing the time evolution of the probability density. Combining the convexity of KL and the LaSalle invariance principle Wasserstein gradient flows, the PDE reaches a unique stationary solution of $\frac{e^{-V(x)}}{\int_{\mathcal{X}} e^{-V(y)}\mathrm{d}y}$.

## B.2 Background on Wasserstein-Fisher-Rao and other related geometries

The Wasserstein-Fisher-Rao geometry is also known as *Hellinger–Kantorovich* in some of papers (Liero et al., 2016; 2018). In this section, we provide an overview of the geometry tailored to meet our technical needs. Along the way, we also briefly describe various metrics and geometries related to the Wasserstein space.

**The Wasserstein space.** Let $\mu, \nu \in \mathcal{P}_2(\mathbb{R}^d)$ be marginal probability densities with respect to the Lebesgue measure. We define the squared Wasserstein distance by a problem of couplings (Villani, 2009)

$$\tfrac{1}{2}W_2^2(\mu,\nu) = \inf_{\pi \in \Pi(\mu,\nu)} \int_{\mathbb{R}^d \times \mathbb{R}^d} \frac{1}{2}\|x-y\|^2 d\pi(x,y). \tag{64}$$

The Wasserstein distance offers a principled metric for quantifying the discrepancy between the probability distributions of random variables $X$ and $Y$. Moreover, the Wasserstein space admits a fluid-dynamical formulation, where optimal transport is represented by space-time velocity fields that satisfy the continuity equation. The Brenier theorem (Villani, 2021) states that there exists an optimal mapping function that pushes forward $\mu$ to $\nu$, *i.e.* $\nu = \nabla\zeta_{\#}\mu$, where $\zeta : \mathbb{R}^d \to \mathbb{R}^d \cup \{+\infty\}$ is a convex and lower semicontinuous function. The property is formally referred to as an instance of the Monge–Ampère equation (Villani, 2009)

$$g\big(\nabla\zeta(x)\big)\det\big(\nabla^2\zeta(x)\big) = f(x) \quad x \in \mathbb{R}^d, \tag{65}$$

after identifying the source and target densities with $\mu(x) = f(x)dx$ and $\nu(y) = g(y)dy$, respectively. In terms of fluid dynamics, the Brenier map $T = \nabla\zeta$ internally yields a constant-speed geodesic $\{\rho_t\}_{t \in [0,1]}$, time-dependent density evolving from $\rho_0$ to $\rho_1$, described by the following differential equation

$$\rho_t = (\nabla\zeta_t)_{\#}\rho_0, \qquad \nabla\zeta_t := (1-t)\mathrm{id} + t\nabla\zeta, \tag{66}$$

where $\rho_0 = \mu$ and $\rho_1 = \nu$. Assuming the existence of such geodesic, we can also understand finding the optimality of $\{\rho_t\}_{t \in [0,1]}$ with the Benamou–Brenier formulation (Benamou & Brenier, 2000), which involves a velocity field $v_t$ for minimizing the total $L^2$ cost of transportation

$$W_2^2(\mu,\nu) = \min_{\rho,v}\left\{ \int_0^1 \int_{\mathbb{R}^d} \frac{1}{2}\|v_t(x)\|^2 \mathrm{d}\rho_t(x)\,dt \;\Big|\; \rho_0 = \mu,\; \rho_1 = \nu,\; \partial_t\rho_t = -\nabla \cdot (v_t\rho_t) \right\}. \tag{67}$$

The equation dictates *how* the fluid should be transported (which shall be controlled by speed $v_t$) while satisfying the continuity equation of path measure on the right hand side. In the Otto calculus (Otto, 2001), we can understand the Benamou–Brenier formula (67) as a Riemannian geometry with respect to the $W_2$ metric. In this geometric interpretation, the tangent space at $\rho \in \mathcal{P}_2(\mathcal{X})$ are measures of the form $\delta\rho = -\nabla \cdot (v\rho)$ with a velocity field $v \in L^2(\rho, \mathbb{R}^d)$ and the metric is given by

$$\|\rho\|_\rho^2 = \inf_{v \in L^2(\rho,\mathbb{R}^d)}\left\{ \int \|v\|^2\,d\rho \;\Big|\; \delta\rho = -\nabla \cdot (v\rho) \right\}. \tag{68}$$

The Benamou–Brenier formula exhibits dynamics in the Wasserstein space of probability densities where the metric generally governed by dynamical mass transportation cost with the continuity equation, implying the mass of probability is preserved.

**Fisher–Rao metric.** The Fisher–Rao metric is a metric on the space of positive measures $\mathcal{P}_+$ with possibly different total masses. We are interested in the simple case where such measure are represented with a fininte number of parameters such as exponential families. We use the following definition throughout the paper.

**Definition 7** (Fisher–Rao metric)**.** The Fisher–Rao distance between measures $\rho_0, \rho_1 \in \mathcal{M}_+$ is given by

$$d_{\mathrm{FR}}^2(\rho_0,\rho_1) := \inf_{\rho,v \in \mathcal{A}[\rho_0,\rho_1]} \int_0^1 \int_{\mathbb{R}^d} \frac{1}{2}\omega_t^2(x)\,d\rho_t(x)dt = 2\int_{\mathbb{R}^d}\left|\sqrt{\frac{\mathrm{d}\rho_0}{\mathrm{d}\lambda}} - \sqrt{\frac{\mathrm{d}\rho_1}{\mathrm{d}\lambda}}\right|^2 d\lambda$$

where $\mathcal{A}$ is an admissible set for a scalar field on positive measures; $\lambda$ is any reference measure such that $\rho$ and $\rho'$ are both absolutely continuous with respect to $\lambda$, with Radon-Nikodym derivatives $\frac{\mathrm{d}\rho_i}{\mathrm{d}\lambda}$.

The equivalence between the square Fisher–Rao distance and squared Hellinger distance (Liero et al., 2016; 2018) quantifies the similarity between two probability distributions ranging from 0 to 1. The total variation bounds the squared form and is well-studied in the information geometry (Amari, 2016). The partial differential equations of the form $\partial_t \rho_t = \alpha_t \rho_t$ are called reaction equations of $\alpha_t$, which describes dynamics regarding concentration.

**Wasserstein-Fisher-Rao.** The Wasserstein-Fisher-Rao geometry, or equivalently, spherical Hellinger–Kantorovich distance, considers liftings of positive, complete, and separable measures while preserving the total mass. This can be expresses as combining the Fisher–Rao and Wasserstein geometries characterized by PDE such as (Liero et al., 2016):

$$\partial_t \rho_t + \nabla \cdot (v_t \rho_t) = \frac{\omega_t}{2} \rho_t. \tag{69}$$

One problem, is that the PDE (69) In order to stay the dynamics on the space of probability measures, which is our interest, we adopt the definition from (Lu et al., 2019; Lambert et al., 2022) the equation becomes

$$\partial_t \rho_t + \nabla \cdot (\rho_t v_t) = \frac{1}{2}\left(\beta_t - \int_{\mathbb{R}^d} \beta_t \, d\rho_t\right)\rho_t, \tag{70}$$

which satisfies mass conservation. For the geometry, the norm on tangent space is given by

$$\|(\beta_t, \rho)\|_\rho^2 := \int \left\{\left(\omega - \int_{\mathbb{R}^d} \beta_t \, \mathrm{d}\rho\right)^2 + \|v\|^2\right\}\mathrm{d}\rho. \tag{71}$$

and we define the WFR distance as

$$d_{\mathtt{WFR}}^2(\rho_0, \rho_1) := \inf_{\rho, \beta_t, v}\left\{\int_0^1 \|(\beta_t, v_t)\|_{\rho_t}^2 \mathrm{d}t \,\bigg|\, \{\rho_t, \beta_t, v_t\}_{t\in[0,1]} \text{ satisfies (70)}\right\}. \tag{72}$$

Lu et al. (2019) demonstrated that Wasserstein-Fisher-Rao gradient dynamics over the Bures–Wasserstein space can be analytically derived with closed form expressions. In this work, we were able to design a computational method for OMD iterates in the WFR geometry. Using Proposition 3, this geometry allowed the VMSB algorithm to perform tractable gradient computation within Wasserstein space.

### B.3 The Bures–Wasserstein space and a mixture of Gaussians

The space of Gaussian distribution in the Wasserstein space is known as Bures–Wasserstein space, denoted as $\mathtt{BW}(\mathbb{R}^d)$. Given $\theta_0, \theta_1 \in \mathtt{BW}(\mathbb{R}^d)$, we can identify the space with the manifold $\mathbb{R}^d \times \mathbf{S}_{++}^d$, where $\mathbf{S}_{++}^d$ denotes the space of symmetric positive definite matrices. For $\theta_0 = (m_0, \Sigma_0)$ and $\theta_1 = (m_1, \Sigma_1)$ an affine map from $p_{\theta_0}$ to $p_{\theta_1}$ is given as a closed-form expression:

$$\nabla\zeta(x) = m_1 + \Sigma_0^{-1/2}\left(\Sigma_0^{1/2}\Sigma_1\Sigma_0^{1/2}\right)^{1/2}\Sigma^{-1/2}(x - m_0).$$

Note that the constant-speed geodesic also lies in $\mathtt{BW}(\mathbb{R}^d)$, as pushforward of a Gaussian with an affine map is also a Gaussian. Therefore, it can be said that $\mathtt{BW}(\mathbb{R}^d)$ is a geodesically convex subset of $\mathcal{P}_2(\mathbb{R}^d)$. For the Brenier map, a constant-speed geodesic in $\mathtt{BW}(\mathbb{R}^d)$, for the tangent vector to the geodesic $(r, S)$

$$p_{\theta_t} = \exp_{p_{\theta_0}}\!\big(t \cdot (r, S)\big) = \mathcal{N}\big(m_0 + tr, (tS + I_d)\Sigma_0(tS + I_d)\big), \tag{73}$$

and the dynamics at its current position at time $t = 0$ is represented as

$$\dot{m}_0 = r, \tag{74}$$

$$\dot{\Sigma}_0 = S\Sigma_0 + \Sigma_0 S. \tag{75}$$

Generalizing this geodesic dynamics, the Bures–Wasserstein gradient $\nabla_{\mathtt{BW}} f$ of a function $f : \mathbb{R}^d \times \mathbf{S}_{++}^d \to \mathbb{R}$ for a tangent vector $(r, S)$ at time 0 Altschuler et al. (2021)

$$\big\langle \nabla_{\mathtt{BW}} f(m_0, \Sigma_0), (r, S)\big\rangle_{\mathtt{BW}} = \partial_t f(m_t, \Sigma_t)\Big|_{t=0}$$

Identifying each component, we achieve the following result of Wasserstein gradient flow in Bures–Wasserstein space as

$$\nabla_{\text{BW}} f = (\nabla_m f, 2\nabla_\Sigma f), \tag{76}$$

where $\nabla_m$ and $\nabla_\Sigma$ denote Euclidean gradient. We refer readers to Appendix A of Altschuler et al. (2021) and Appendix B of Lambert et al. (2022) for further useful geometric properties of Wasserstein spaces and dedicated discussions for the Bures–Wasserstein space.

## C  Background on the Dynamic Schrödinger Bridge Problem

The equivalence between static and dynamic SBPs (Pavon & Wakolbinger, 1991; Léonard, 2012) has been studied, allowing us to consider the both problems interchangeably. This appendix introduces a general control dynamic formulation for describing SB and SB-FBSDE theory (Gigli & Tamanini, 2020; Chen et al., 2022). These formulations establish fundamental links to optimal control theory and diffusion models.

### C.1  Time-Symmetric Formulations of Optimal Control

For variable drift and diffusion coefficient, a stochastic process, and its time reversal respectively follow the forward and backward Kolmogorov (or Fokker–Planck) equations (Fa, 2011; Gigli & Tamanini, 2020):

$$-\frac{\partial \rho}{\partial t} + \nabla \cdot \left[(f_t + \nabla\varphi_t)\rho_t\right] = \frac{1}{2}\nabla^2 \cdot \left(g_t g_t^\mathsf{T} \rho_t\right), \tag{77a}$$

$$\frac{\partial \rho_t}{\partial t} + \nabla \cdot \left[(-f_t + \nabla\psi_t)\rho_t\right] = \frac{1}{2}\nabla^2 \cdot \left(g_t g_t^\mathsf{T} \rho_t\right), \tag{77b}$$

where $f_t$ and $g_t$ are time-varying base drift and diffusion coefficients which together determine a reference measure $R$;[5] $\nabla\cdot$ denotes divergence; $\nabla^2\cdot$ denotes squared divergence. By subtracting Eq. (77b) from Eq. (77a), the continuity equation

$$\frac{\partial \rho}{\partial t} + \nabla \cdot (v\rho) = 0 \tag{78}$$

is achieved by $v = f + \frac{1}{2}[\nabla\varphi - \nabla\psi]$. Adding and scaling of Eq. (77), also derives another identity

$$(\nabla\varphi + \nabla\psi)\rho = \nabla \cdot \left(gg^\mathsf{T}\rho\right), \tag{79}$$

where we can derive an explicit form of the score function

$$\nabla\varphi + \nabla\psi = gg^\mathsf{T}\nabla\log\rho + \underbrace{\nabla \cdot gg^\mathsf{T}}_{\text{heat flux}}. \tag{80}$$

The rightmost term of (80) is a heat flux, indicating the amount of diffusion per unit time.

The time-symmetry relation can be considered as a multivariate derivation of the the stochastic mechanics. Generalizing the Nelson's notation (Nelson, 2001), let us define the SB drifts and *current* drifts for continuous-time path measures $P$ and $Q$ receptively satisfying Eq. (77) with the shared diffusion coefficient:

$$f_{P,t}^+ := f_{P,t} + \nabla\varphi_{P,t}, \quad f_{P,t}^- := -f_{P,t} + \nabla\psi_{P,t}, \quad f_{Q,t}^+ := f_{Q,t} + \nabla\varphi_{Q,t}, \quad f_{Q,t}^- := -f_{Q,t} + \nabla\psi_{Q,t} \tag{81}$$

$$v_P = \frac{f_P^+ - f_P^-}{2}, \qquad v_Q = \frac{f_Q^+ - f_Q^-}{2}. \tag{82}$$

Then, we consider the Girsanov theorem (Øksendal, 2003) for the drifts. The theorem formulate Radon-Nikodym derivatives between path measures of stochastic processes.

**Lemma 19** (Girsanov theorem; Theorem 8.6.3 of Øksendal, 2003). *For adapted processes with a given time interval $[0, T]$, let $\widehat{B}_s$ be an Itô process solving SDE*

$$\mathrm{d}\widehat{B}_s = -\alpha(\omega, s) + \mathrm{d}B_s' \tag{83}$$

---

[5]In (7) of § 3, base drift is zero and diffusion is $\sqrt{\varepsilon}$.

*for $\omega \in \mathcal{X}$, $0 \le s \le T$ and $\widehat{B}_0 = 0$, where $\alpha$ satisfies the Novikov's condition. Then, $\widehat{B}_s$ is a Brownian motion with respect to the path measure $Q$, satisfying the Radon-Nikodym derivative*

$$\frac{\mathrm{d}P}{\mathrm{d}Q}(\omega) \coloneqq \exp\left(\int_0^T \alpha(\omega, s)\mathrm{d}B_s' - \int_0^T \frac{1}{2}\|\alpha(\omega, s)\|^2 \mathrm{d}s\right) \tag{84}$$

Next, we present the disintegration theorem in the context of probability measures (Léonard, 2014; Vargas et al., 2021), which extends the product rule to measures that do not admit the traditional product rule. Similar to the product rule, these theorems are essential for decomposing and manipulating path measures for dynamic SBPs, eventually connecting various formulation of Schrödinger bridge problems.

**Lemma 20** (Disintegration for continuous probability measures)**.** *For a probability space $(\mathcal{Z}, \mathcal{F}, \mathcal{P})$ where $\mathcal{Z}$ is a product space: $\mathcal{Z} = \mathcal{X} \times \mathcal{Y}$ and ① $\mathcal{X}, \mathcal{Y} \in \mathbb{R}^d$ and $\phi_i : \mathcal{Z} \to \mathcal{Z}_i$ is a measurable function known as the canonical projection operator (i.e., $\phi_1(x, \cdot) = x$ and $\phi_1^{-1}(x) = \{(x, y)|\phi_1(x, y) = x\}$), There exists a measure $P_{y|x}(\cdot|x)$, such that*

$$\iint_{\mathcal{X} \times \mathcal{Y}} f(x, y)\mathrm{d}P(y) = \iint_{\mathcal{X} \times \mathcal{Y}} f(x, y)\mathrm{d}P_{y|x}(y|x)\mathrm{d}P(\phi_1^{-1}(x)) \tag{85}$$

*where $P_x(\cdot) = P(\phi_1^{-1}(\cdot))$ is a probability measure, referred to as a push-forward measure, and corresponds to the marginal distribution.*

The theorem suggests that one way to achieve KL projection between path measures is by matching drifts with the time reversal drifts of $(\gamma^+, \gamma^-)$. Under the Girsanov theorem, we observe that

$$\begin{aligned}
\mathrm{KL}(P\|Q) &= \mathrm{KL}(P_0\|Q_0) + \mathbb{E}_P\left[\int_0^T \frac{1}{2}\|f_{P,t}^+ - f_{Q,t}^+\|^2 \mathrm{d}t\right] \\
&= \mathrm{KL}(P_T\|Q_T) + \mathbb{E}_P\left[\int_0^T \frac{1}{2}\|f_{P,t}^- - f_{Q,t}^-\|^2 \mathrm{d}t\right]
\end{aligned} \tag{86}$$

Let us consider a reference measure $Q = \mathcal{R}$ with based drift and diffusion *i.e.* $f_Q^+ = f$. Knowing the boundary conditions $P_0 = \mathcal{R}_0 = \mu$ and $\mathcal{R}_T = Q_T = \nu$, Eq. (86) can be reduce to the following problems

$$\mathrm{KL}(P\|R) = \inf_{(v^+, \rho^+)} \int_0^T \iint_{\mathbb{R}^d} \frac{1}{2}|v_t^+(x)|^2 \rho_t^+(x) \,\mathrm{d}x\mathrm{d}t, \tag{87a}$$

$$= \inf_{(v^-, \rho^-)} \int_0^T \iint_{\mathbb{R}^d} \frac{1}{2}|v_t^-(x)|^2 \rho_t^-(x) \,\mathrm{d}x\mathrm{d}t, \tag{87b}$$

and Eqs. (6) and (7) in the main text represent a simplified case when $R = W^\varepsilon$. Furthermore, by combining Eq. (86), we get a symmetrized version of relative entropy (KL functional)

$$\mathrm{KL}(P\|Q) = \frac{1}{2}\mathrm{KL}(P_0\|Q_0) + \frac{1}{2}\mathrm{KL}(P_T\|Q_T) + \mathbb{E}_P\left[\int_0^T \frac{1}{4}\|f_P^+ - f_Q^+\|^2 + \frac{1}{4}\|f_P^- - f_Q^-\|^2\mathrm{d}t\right] \tag{88}$$

Finally, using Eq. (79) we get the constrained problem:

$$\inf_{(\tilde{\rho}, \tilde{v})} \int_0^T \iint_{\mathbb{R}^d} \left(\frac{1}{2}\|\tilde{v}(t, x)\|^2 + \frac{1}{8}\|\nabla \log \tilde{\rho}(t, x)\|_{gg^\mathsf{T}}^2 + \frac{1}{8}\|\nabla \cdot gg^\mathsf{T}\|^2\right)\tilde{\rho}(x, t) \,\mathrm{d}t \,\mathrm{d}x,$$

$$\text{such that} \quad \frac{\partial \tilde{\rho}}{\partial t} + \nabla \cdot [(f + \tilde{v})\tilde{\rho}] = 0, \ \ \tilde{\rho}(0, \cdot) \equiv \mu, \ \ \tilde{\rho}(T, \cdot) \equiv \nu.$$

To solve this problem, we convert the problem to the Lagrangian function:

$$\begin{aligned}
\mathcal{L}(\rho, v) = \int_0^T \iint_{\mathbb{R}^d} &\frac{1}{2}\|\tilde{v}(t, x)\|^2 \tilde{\rho}(t, x) + \frac{1}{8}\|\nabla \log \tilde{\rho}(t, x)\|_{gg^\mathsf{T}}^2 \tilde{\rho}(t, x) + \frac{1}{8}\|\nabla \cdot gg^\mathsf{T}(t, x)\|^2 \tilde{\rho}(t, x) \\
&+ \lambda(t, x)\left(\frac{\partial \tilde{\rho}}{\partial t} + \nabla \cdot ((f + \tilde{v})\tilde{\rho})\right)\mathrm{d}x \,\mathrm{d}t
\end{aligned}$$

where $\lambda$ is $C^{1,2}$-Lagrangian multiplier. After integration by part, assuming that limits for $x \to \infty$ are zero, and observing that the boundary values are constant over $\Pi(\mu, \nu)$, we resort to the following problem:

$$\inf_{(\tilde{\rho}, \tilde{v}) \in \mathcal{P} \times \mathcal{V}} \int_{\mathbb{R}^d} \int_0^T \left[ \frac{1}{2} \|\tilde{v}(t, x)\|^2 + \frac{1}{8} \|\nabla \log \tilde{\rho}(t, x)\|_{gg^\mathsf{T}}^2 + \frac{1}{8} \|\nabla \cdot gg^\mathsf{T}(t, x)\|^2 + \left( -\frac{\partial \lambda}{\partial t} - \nabla \lambda \cdot (f + \tilde{v}) \right) \right] \tilde{\rho}(t, x) \mathrm{d}t \mathrm{d}x. \tag{89}$$

Pointwise minimization with respect to $\tilde{v}$ for each fixed flow of probability densities $\tilde{\rho}$ gives $v^*(x, t) = \nabla \lambda(x, t)$. Plugging this form of the optimal control into Eq. (89), we get the functional of $\tilde{\rho} \in \mathcal{P}$:

$$\mathcal{J}(\tilde{\rho}) = -\int_{\mathbb{R}^d} \int_0^T \left[ \frac{\partial \lambda}{\partial t} + (f + v) \cdot \nabla \lambda + \frac{1}{2} \|\nabla \lambda\|^2 + \frac{1}{8} \|\nabla \log \tilde{\rho}(t, x)\|_{gg^\mathsf{T}}^2 + \frac{1}{8} \|\nabla \cdot gg^\mathsf{T}(t, x)\|^2 \right] \tilde{\rho}(t, x) \mathrm{d}t \mathrm{d}x$$

Utilizing the existence and uniqueness of SDE solutions (Øksendal, 2003), the optimality of the problem is uniquely identified by $\nabla \lambda$ being the mean current of SB-FBSDE which is explained in the following section.

## C.2 The Schrödinger bridge forward-backward stochastic differential equation theory

We introduce a complete, multivariate derivation of SB-FBSDE (Chen et al., 2022; Liu et al., 2022), which is an explicit representation of a dynamic Schrödinger bridge model using the forward-backward stochastic differential equation theory. First, we present the Itô's lemma for stochastic differential equations.

**Lemma 21** (Itô's lemma (Itô, 1951)). *Let $X_t$ be the solution to the Itô SDE:*

$$\mathrm{d}X_t = f(t, X_t)\, \mathrm{d}t + g(t, X_t)\mathrm{d}W_t$$

*Then, the stochastic process $u(t, X_t)$, where $u \in C^{1,2}([0, T], \mathbb{R}^d)$, is also an Itô process satisfying*

$$\mathrm{d}u(t, X_t) = \frac{\partial u(t, X_t)}{\partial t}\mathrm{d}t + \left[ \nabla u(t, X_t)^\mathsf{T} f(t, X_t) + \frac{1}{2} \mathrm{Tr}[gg^\mathsf{T}(t, X_t)\nabla^2 u(t, X_t)]\mathrm{d}W_t. \right]\mathrm{d}t \tag{90}$$
$$+ [\nabla u(t, X_t)^\mathsf{T} g(t, X_t)]\mathrm{d}W_t$$

Next, we introduce the nonlinear Feynman–Kac lemma, which predicts partial differential equation involving potentials, or value functions for describing SDE.

**Lemma 22** (Nonlinear Feynman–Kac (Exarchos & Theodorou, 2018; Yong & Zhou, 1999)). *Let $u \equiv u(x, t)$ be a function that is twice continuously differentiable in $x \in \mathbb{R}^d$ and once differentiable in $t \in [0, T]$, i.e., $u \in C^{1,2}([0, T], \mathbb{R}^d)$. Consider the following second-order parabolic PDE,*

$$\frac{\partial u}{\partial t} + \frac{1}{2} \mathrm{Tr}(gg^\mathsf{T}\nabla^2 u) + \nabla u^\mathsf{T} f(t, x) + h(t, x, u, g^\mathsf{T}\nabla u) = 0, u(T, \cdot) \equiv \tau(\cdot), \tag{91}$$

*where the functions $f$, $g$, $h$, and $\tau$ satisfy proper regularity conditions. Specifically, ① $f$, $g$, $h$, and $\tau$ are continuous, ② $f(t, x)$ and $g(t, x)$ are uniformly Lipschitz in $x$, and ③ $h(t, x, y, z)$ satisfies quadratic growth condition in $z$. Then, Eq. (91) exists a unique solution $v = u$ such that the following stochastic representation (known as the nonlinear Feynman–Kac transformation) holds:*

$$Y_t = u(t, X_t), \qquad Z_t = g^\mathsf{T}(t, X_t)\nabla u(t, X_t) \tag{92}$$

*where $(X_t, Y_t, Z_t)$ are the unique adapted solutions to the following FBSDEs:*

$$\mathrm{d}X_t = f(X_t, t)\mathrm{d}t + g(X_t, t)\mathrm{d}W_t, \qquad X_0 = x_0, \tag{93a}$$
$$\mathrm{d}Y_t = -h(X_t, Y_t, Z_t, t)\mathrm{d}t + Z_t^\mathsf{T}\mathrm{d}W_t, \qquad Y_T = \tau(X_T). \tag{93b}$$

*The original deterministic PDE solution $v(x, t)$ can be recovered by taking conditional expectations:*

$$\mathbb{E}[Y_t | X_t = x] = u(t, x), \qquad \mathbb{E}[Z_t | X_t = x] = g(t, x)^\mathsf{T}\nabla u(t, x).$$

**SB-FBSDE.** SB-FBSDE is a class of probabilistic models that, inspired by optimal control and neural differential equations (Chen et al., 2022; Kirk, 1970; Chen et al., 2018a) to generalize the score-based diffusion models. We state a more general expression of SB-FBSDE than previous literature

$$
\begin{cases}
\mathrm{d}X_t = f_t^+ \mathrm{d}t + g^\mathsf{T}\mathrm{d}W_t = (f_t + gZ_t)\mathrm{d}t + g^\mathsf{T}\mathrm{d}W_t & \text{(94a)} \\[2mm]
\mathrm{d}Y_t = \frac{1}{2}\|Z_t\|^2\mathrm{d}t + Z_t^\mathsf{T}\,\mathrm{d}W_t & \text{(94b)} \\[2mm]
\mathrm{d}\widehat{Y}_t = \left(\frac{1}{2}\|\widehat{Z}_t\|^2 + \widehat{Z}_t^\mathsf{T}Z_t + \nabla\cdot f_t^- + \mathbf{G}_t\right)\mathrm{d}t + \widehat{Z}_t^\mathsf{T}\,\mathrm{d}W_t & \text{(94c)}
\end{cases}
$$

where we define $f_t^- := -f_t + g_t\widehat{Z}_t$ and $\mathbf{G}_t := \frac{1}{2}\nabla^2\cdot(gg_t^\mathsf{T})$. The Feynman–Kac transform (Lemma 22) reads

$$
Y_t = \log\Psi(t,X_t), \quad Z_t = g(t,X_t)^\mathsf{T}\nabla\log\Psi(t,X_t), \quad \widehat{Y}_t = \log\widehat{\Psi}(t,X_t), \quad \widehat{Z}_t = g(t,X_t)^\mathsf{T}\nabla\log\widehat{\Psi}(t,X_t),
$$

which immediately suggests that $\mathbb{E}[Y_t|X_t = x] = \log\Psi(t,x)$ and $\mathbb{E}[\widehat{Y}_t|X_t = x] = \log\widehat{\Psi}(t,x)$. The above arguments also holds for the backward timeline of $s$ with equivalent $\widehat{\Psi}$ and $\Psi$ interchange their roles.

### C.3 Derivations for Multi-Variate SB-FBSDE on General Diffusion

### C.3.1 A Hopf–Cole Transformation

For an Hamiltonian $\mathcal{H}$, we consider a dynamic formulation with the following PDE that generalizes Eq. (7)

$$
\begin{cases}
-\partial_t u + \mathcal{H}(x,\nabla u) - \frac{1}{2}\mathrm{Tr}(gg^\mathsf{T}\nabla^2 u) = 0, \\
\partial_t\rho - \nabla\cdot(\nabla_p\mathcal{H}(x,\nabla u)\,\rho) - \frac{1}{2}\nabla^2\cdot(gg^\mathsf{T}\rho) = 0.
\end{cases}
\tag{95}
$$

This can be viewed as blending of the Hamilton-Jacobi-Bellman equation and the Fokker–Planck equation (Buckdahn et al., 2017). For detailed derivation, we use the Hopf–Cole transform (Hopf, 1950; Cole, 1951)

$$
\Psi(t,x(t)) := \exp(-u(t,x(t))), \qquad \widehat{\Psi}(\xi(t),t) := \rho(t,x(t))\exp\big(u(t,x(t))\big),
$$

where $\rho(t,x)$ is an arbitrary density function. Note that the mulvariate calculus yields

$$
\nabla\Psi = -\exp(-u)\nabla u, \qquad \nabla^2\Psi = \exp(-u)[\nabla u\nabla u^\mathsf{T} - \nabla^2 u],
$$
$$
\nabla\widehat{\Psi} = \exp(u)(\rho\nabla u + \nabla\rho), \nabla^2\widehat{\Psi} = \exp(u)\Big[\rho\nabla u\nabla u^\mathsf{T} + \nabla\rho\nabla u^\mathsf{T} + \nabla u\nabla\rho^\mathsf{T} + \nabla^2\rho + \rho\nabla^2 u\Big].
$$

Hence, we can draw the following derivations regarding the control-affine Hamiltonian, *i.e.*, $\mathcal{H}(x,\nabla u) = \frac{1}{2}\|g^\mathsf{T}\nabla u\|^2 - \nabla u^\mathsf{T}f$:

$$
\begin{aligned}
\frac{\partial\Psi}{\partial t} &= \exp(-u)\left(-\frac{\partial u}{\partial t}\right) = \exp(-u)\left(-\frac{1}{2}\|g^\mathsf{T}\nabla u\|^2 + \nabla u^\mathsf{T}f + \frac{1}{2}\mathrm{Tr}(gg^\mathsf{T}\nabla^2 u)\right) \\
&= \exp(u)\left(\left(\nabla\cdot(\rho(gg^\mathsf{T}\nabla u - f)) + \frac{1}{2}\nabla^2\cdot(gg^\mathsf{T}\rho)\right) + \rho\left(\frac{1}{2}\|g^\mathsf{T}\nabla u\|^2 - \nabla u^\mathsf{T}f - \frac{1}{2}\mathrm{Tr}(gg^\mathsf{T}\nabla^2 u)\right)\right) \\
&= \frac{1}{2}\mathrm{Tr}(gg^\mathsf{T}\nabla^2\widehat{\Psi}) + \nabla\cdot(gg^\mathsf{T})^\mathsf{T}\nabla\widehat{\Psi} + \frac{1}{2}\nabla^2\cdot(gg^\mathsf{T})\widehat{\Psi} - \nabla\widehat{\Psi}^\mathsf{T}f - \widehat{\Psi}\nabla\cdot f \\
&= \frac{1}{2}\nabla^2\cdot(gg^\mathsf{T}\widehat{\Psi}) - \nabla\widehat{\Psi}^\mathsf{T}f - \widehat{\Psi}\nabla\cdot f = -\nabla\cdot(\widehat{\Psi}f)
\end{aligned}
$$

and we have derivations for the control PDE (95), which fully generalizes Eq. (7).

### C.3.2 Nonlinear Multi-variate Feynman–Kac Derivations

Let us apply the Itô's lemma to the $u := \log\Psi(t,X_t)$ where $X_t$ follows the forward equation:

$$
\mathrm{d}\log\Psi = \frac{\partial\log\Psi}{\partial t} + \left[\nabla\log\Psi^\mathsf{T}\big(f + gg^\mathsf{T}\nabla\log\Psi\big) + \frac{1}{2}\mathrm{Tr}\big(gg^\mathsf{T}\nabla^2\log\Psi\big)\right]\mathrm{d}t + g\nabla\log\Psi^\mathsf{T}\mathrm{d}W_t.
$$

Notice that the PDE of $\frac{\partial \log \Psi}{\partial t}$ is achieved by applying the Hopf–Cole transform

$$\frac{\partial \log \Psi}{\partial t} = \frac{1}{\Psi}\left(-\nabla\Psi^\mathsf{T} f - \frac{1}{2}\mathrm{Tr}(gg^\mathsf{T}\nabla^2\Psi)\right) = -\nabla\log\Psi^\mathsf{T} f - \frac{1}{2}\mathrm{Tr}(gg^\mathsf{T}\nabla^2\log\Psi) - \frac{1}{2}\|g^\mathsf{T}\nabla\log\Psi\|^2.$$

Therefore, combining above differential terms yields

$$\mathrm{d}\log\Psi = \frac{1}{2}\|g\nabla\log\Psi\|^2\,\mathrm{d}t + g\nabla\log\Psi^\mathsf{T}\,\mathrm{d}W_t. \tag{96}$$

Also, apply the Itô lemma by instead substituting $u := \log\widehat{\Psi}(t, X_t)$

$$\mathrm{d}\log\widehat{\Psi} = \frac{\partial\log\widehat{\Psi}}{\partial t}\mathrm{d}t + \left[\nabla\log\widehat{\Psi}^\mathsf{T}(f + gg^\mathsf{T}\nabla\log\Psi) + \frac{1}{2}\mathrm{Tr}(gg^\mathsf{T}\nabla^2\log\widehat{\Psi})\right]\mathrm{d}t + g\nabla\log\widehat{\Psi}^\mathsf{T}\mathrm{d}W_t.$$

Notice that the PDE of $\frac{\partial\log\widehat{\Psi}}{\partial t}$ obeys

$$\frac{\partial\log\widehat{\Psi}}{\partial t} = \frac{1}{\widehat{\Psi}}\left(-\nabla\cdot(\widehat{\Psi}f) + \frac{1}{2}\nabla^2\cdot(gg^\mathsf{T}\widehat{\Psi})\right) = -\nabla\log\Psi^\mathsf{T} f - \nabla\cdot f + \frac{1}{2}\|g^\mathsf{T}\nabla\log\widehat{\Psi}\|^2 + \frac{1}{2}\mathrm{Tr}(gg^\mathsf{T}\nabla^2\log\widehat{\Psi}) + \frac{1}{2}\nabla^2\cdot(gg^\mathsf{T})$$

where $\frac{1}{2}\nabla^2\cdot(gg^\mathsf{T})$ is the adjustment term for non-constant diffusion $g$. This yields

$$\mathrm{d}\log\widehat{\Psi} = \left[\frac{1}{2}\|g^\mathsf{T}\nabla\log\widehat{\Psi}\|^2 + (g^\mathsf{T}\nabla\log\widehat{\Psi})^\mathsf{T}(g^\mathsf{T}\nabla\log\Psi) + \nabla\cdot(gg^\mathsf{T}\nabla\log\widehat{\Psi} - f) + \frac{1}{2}\nabla^2\cdot(gg^\mathsf{T})\right]\mathrm{d}t + g\nabla\log\widehat{\Psi}^\mathsf{T}\mathrm{d}W_t,$$

Therefore, with the nonlinear FK transformation (Pereira et al., 2020), we can write the SB-FBSDE of the system system

$$\mathrm{d}X_t = (f_t + gZ_t)\,\mathrm{d}t + g\,\mathrm{d}W_t, \mathrm{d}Y_t = \frac{1}{2}\|Z_t\|^2\,\mathrm{d}t + Z_t^\mathsf{T}\mathrm{d}W_t$$

$$\mathrm{d}\widehat{Y}_t = \left[\frac{1}{2}\|\widehat{Z}_t\|^2 + \widehat{Z}_t^\mathsf{T} Z_t + \nabla\cdot\left(g_t\widehat{Z}_t - f_t\right) + \frac{1}{2}\nabla^2\cdot(gg_t^\mathsf{T})\right]\mathrm{d}t + \widehat{Z}_t^\mathsf{T}\mathrm{d}W_t$$

where we find forward control and reverse drift has the relationship with FK transformation as

$$f_t^+ = f_t + g_t Z_t \quad \text{and} \quad f_t^- = g_t\widehat{Z}_t - f_t$$

Derivation of the second FBSDEs system in time-reversal follows the identical flow, except that we need to rebase the PDE to the "reversed" time coordinate $s := T - t$. This can also be derived by reformulating (95) under the $s$ coordinate, then applying the following Hopf–Cole transform.

Finally, we present the SB-FBSDE for $X(t)$

$$\text{SB-FBSDE} \atop \text{(Forward)} : \begin{cases} \mathrm{d}X_t = f_t^+\mathrm{d}t + g_t\,\mathrm{d}W_t, & \text{(97a)} \\[2mm] \mathrm{d}Y_t = \frac{1}{2}\|Z_t\|^2\mathrm{d}t + Z_t^\mathsf{T}\,\mathrm{d}W_t, & \text{(97b)} \\[2mm] \mathrm{d}\widehat{Y}_t = \left(\frac{1}{2}\|\widehat{Z}_t\|^2 + \widehat{Z}_t^\mathsf{T} Z_t + \nabla\cdot f_t^- + \mathbf{G}_t\right)\mathrm{d}t + \widehat{Z}_t^\mathsf{T}\,\mathrm{d}W_t, & \text{(97c)} \end{cases}$$

where $\mathbf{G}_t := \frac{1}{2}\nabla^2\cdot(gg_t^\mathsf{T})$. The SB-FBSDE comprises a dynamic system of evolution toward a unique solution with respect to the cost of $\frac{1}{2}\|Z\|^2$ and $\frac{1}{2}\|\widehat{Z}\|^2$. Since $(Z, \widehat{Z})$ models the pure commitment of control, learning through Eq. (97b) corresponds to satisfying Hamilton–Jacobi solution of the minimum control. The dynamic Schrödinger bridge problem stems from a time-symmetric optimal control formulation, and its associated stochastic flow naturally implements the minimization of a relative entropy cost functional via the Girsanov theorem.

Table 5: Hyperparameters.

|  | **2D** | **EOT** | **MSCI** | **MNIST (Pixel)** | **MNIST (Latent)** | **FFHQ** |
|---|---|---|---|---|---|---|
| Dimension $d$ | 2 | $\{2, 16, 64, 128\}$ | $\{50, 100, 1000\}$ | 784 | 128 | 512 |
| Modality $K$ | $\{8, 20, 50\}$ | $[5, 100]$ | 50 | $\{256, 1024, 4096\}$ | $\{256, 1024\}$ | 10 |
| Volatility $\varepsilon$ | 0.1 | $\{0.1, 1, 10\}$ | 0.1 | $10^{-4}$ | $10^{-3}$ | $\{0.1, 0.5, 1.0, 10.0\}$ |
| Total steps $(\tau)$ | 20,000 | 30,000 | 10,000 | 100,000 | 30,000 | 20,000 |
| OMD steps $(t)$ | 400 | 600 | 200 | 1000 | 375 | 400 |

# D  Experimental Details

## D.1  Rationales of the GMM parameterization for VMSB

Our parameterization choice follows LightSB (Korotin et al., 2024) because of the following two key reasons. First, GMMs ensure that the model space satisfies certain measure concentration, which is suitable for analyzing theoretical properties of SB models (Conforti et al., 2023). Firstly, we analyzed the regret under the log Sobolev inequality in Proposition 2. Enforcing the LightSB parameterization will automatically satisfy Assumption 1. Secondly, VMSB requires tractable gradient computation of Wasserstein gradient flow in § 4.3. As shown in Proposition 3, we can perform VMSB using the variational inference in the WFR geometry of the GMM parameterization.

## D.2  Hyperparameters.

The hyperparameters are displayed in Table 5. For step size scheduling, we followed the theoretical result in Theorem 1 and Proposition 1, and chose $\eta_1 = 1$ and $\eta_T \in \{0.05, 0.1\}$ with harmonic sequences, as illustrated in Fig. 5. For high dimensional tasks in MSCI (1000d), MNIST-EMNIST (784d), and latent FFHQ Image-to-Image transfer tasks (512d), the initial *warm up* steps for 10% of the total learning helped starting a training sequence from a reasonable starting point as this set $\eta_t = 1$ as verified in Fig. 6 (c).

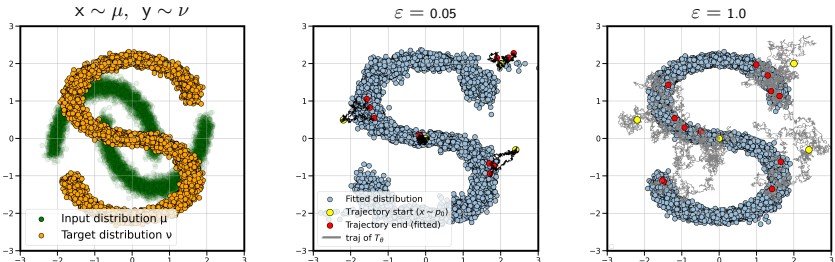

Figure 10: SB in 2D synthetic datasets. SB processes $\mathcal{T}_\theta$ with different volatility $\varepsilon$.

## D.3  2D Synthetic datasets and the online learning setup

Fig. 10 demonstrates that our method achieved the SB model for the various volatility $\varepsilon$. For various configurations, most of baseline SB algorithms are capable of learning in the 2D space (10). In order to align our theoretical arguments for online learning, we selectively offered with a rotating filter that only 12.5% of the samples to the SB solvers based on the angles measured from the origin. For instance, we provided data for angle of $[0, \pi/4]$ for first $t \in [0, 25)$ steps, and so on. This partial observability is periodically rotated through the data stream, thereby testing the algorithm's ability to learn robustly under sparse and shifting information. Since this requires 200 batches for the full rotation of the filter, the problem became substantially more challenging, and LightSB and LightSB-M algorithms oftentimes failed on this online learning setting.

## D.4 Entropic optimal transport benchmark

Our hyperparameter for the EOT benchmarks choices mostly follow the official repositories of the LightSB[6] and LightSB-M[7]. Since it is known that initial distribution $\mu$ is the standard Gaussian distribution (Gushchin et al., 2024b), we only trained $v_\theta$ using the variational MD algorithm. Due to the huge number of configurations, some hyperparameter settings were not clearly reported. Thus, we conducted our own examination on these cases; we replicated better performance than the reported numbers by carefully dealing each benchmark configuration.

## D.5 SB learning with adversarial networks

Suppose a discriminator network, denoted as $D$, is equipped with useful architectural properties for discriminating images. The discriminator outputs a binary classification regarding authenticity through sigmoidal outputs, i.e., $D(x) \in [0,1] \ \forall x \in \mathbb{R}^{28 \times 28 \times 1}$. For image samples $\mathbf{x} = \{x^1, \dots, x^N\} \sim \mu$, we trained the discriminator $D$ with the logistic regression:

$$\underset{D}{\text{maximize}} \ \frac{1}{N} \sum_{n=1}^{N} \log D(y^n) + \frac{1}{B} \sum_{m=1}^{M} \log(1 - D(\hat{y}_\phi^m)), \tag{98}$$

where $\hat{y}_\phi^m$ in the right-hand side denotes a sample from an SB model parameterized by $\phi$, generated using an input $x^m$. From our experiment setting, we use the SB distribution $\rho_\phi$ which is generated by $\vec{\pi}_\phi$ from samples of the marginal $\mu$. This makes the objective of adversarial learning of training the law of SB process at time $t = 1$. For a completely separable metric space, it is well known that the discriminator converges at $D(x) = \frac{\nu(x)}{\nu(x) + \rho_\phi(x)}$ (Goodfellow et al., 2014).

In the adversarial learning technique, retaining a fully differentiable computation path from the input pixels to the discriminator outputs is essential. Therefore, we implemented a differentiable inference function using the categorical reparameterization trick with Gumbel-softmax (Jang et al., 2016), as well as the Gaussian reparameterization trick. These reparameterization tricks enabled learning with samples generated through LightSB-adv-$K$, directly by maximizing

Table 6: A simple discriminator $D$.

| Layer Type | Shape |
|---|---|
| Input Layer | (-1, 28, 28, 1) |
| Conv Layer 1 | (-1, 14, 14, 64) |
| Conv Layer 2 | (-1, 7, 7, 128) |
| Batch Norm | (-1, 7, 7, 128) |
| Flatten | (-1, 6272) |
| Dense | (-1, 1024) |
| Dense | (-1, 1) |

$$\tilde{\mathcal{J}}(\phi) = \frac{1}{M} \sum_{m=1}^{M} \log D(y_\phi^m) - \log(1 - D(y_\phi^m)),$$

where the term essentially represents the *logit* function $\text{logit}(D(y)) = \log \frac{D(y)}{1-D(y)}$. When $D$ approaches the equilibrium, we can approximate the following KL learning

$$\tilde{\mathcal{J}}(\phi) \approx \int \log \frac{\nu(y)}{\rho_\phi(y)} \rho_\phi(y) \mathrm{d}y = \text{KL}(\rho_\phi \| \nu),$$

where the KL functional directly corresponds to the divergence minimization of the SB problems (6) and (28), under the disintegration theorem of Schrödinger bridge (Léonard, 2014).

In the MNIST-EMNIST image transfer tasks, we set one of the baseline as the aforementioned adversarial learning as the baseline for training the SB model for the pixel space. Among our attempts, while the LightSB-adv method successfully generated learning signals to train GMM-based models, the losses proposed by LightSB (Korotin et al., 2024) and LightSB-M (Gushchin et al., 2024a) failed to generate relevant images with high fidelity. For the discriminator, we used the DCGAN (Radford, 2015) architecture shown in Table 6, and this can be replaced with more complex architecture for more realistic images with high fidelity. We fixed the covariance after warm-ups in 10,000 steps, and we used the entropy coefficient $\varepsilon = 10^{-4}$ based on our hyperparameter search.

---

[6]https://github.com/ngushchin/LightSB
[7]https://github.com/SKholkin/LightSB-Matching

Table 7: Training time for the 100-dimension single-cell data problem.

| Sinkhorn (IPF) | LightSB | VMSB |
|---|---|---|
| 8m (GPU) | 66s (CPU) | 32s (GPU) / 22m (CPU) |

Table 8: Generation time for the 784-dimension MNIST pixel data.

| | $K = 64$ | $K = 256$ | $K = 1024$ | $K = 4096$ | NN (SDE) |
|---|---|---|---|---|---|
| GPU | $721\mu s$ | $726\mu s$ | $739\mu s$ | $740\mu s$ | 1.372s |
| CPU | 60.140ms | 133.333ms | 428.433ms | 1.527s | – |

## D.6 Latent diffusion experiments

For the latent space, we pretrained ALAE (Pidhorskyi et al., 2020) model using the both MNIST and EMNIST (first ten letters) datasets. The ALAE is a high-fidelity autoencoder internally use an adversarial learning to generate high-fidelity images. For the encoder network, as well as decoder network, we mostly adopt the DCGAN architecture. Therefore, the encoder is mostly identical to Table 6 except the point the final layer is 128 dimension instead of 1, and the decoder is a convolutional neural network with four convolutional layers.

Following the latent SB setting (Korotin et al., 2024), we assessed our method by utilizing the ALAE model (Pidhorskyi et al., 2020) for generating $1024 \times 1024$ images of the FFHQ dataset (Karras et al., 2019). The base generative model has a latent embedding layer which represent 512-dimensional embedding space. The goal is to transport a point latent space to another, performing unpaired image-to-image translation tasks for four distinct cases: *Adult→ Child*, *Child→ Adult*, *Female→ Male*, and *Male→ Female*. We conducted a quantitative analysis using the ED on the predefined ALAE embedding as a metric for evaluation.

## E Discussion on Implementation of VMSB

**Limitations.** GMM-based SB models, due to the lack of deep structural processing, tend to focus on *instance-level* associations of images in EOT couplings rather than the *subinstance-* or *feature-level* associations that are intrinsic to deep generative models. As a result, while VMSB produces statistically valid representations of optimal transportation within the given architectural constraints, these outcomes may be perceived as somewhat "synthetic." Nevertheless, GMM-based models still hold an irreplaceable role in numerous problems such as latent diffusion and variational methods, due to their simplicity and distinctive properties (Korotin et al., 2024). As we successfully demonstrated in two distinct ways of interacting with neural networks for solving unpaired image transfer, we hope our theoretical and empirical findings help novel neural architecture studies.

**Computation.** For fast computation, we utilized the JAX automatic differentiation library (Bradbury et al., 2018) for computing gradients and Hessians in Proposition 3. For each input, the computational of VMSB requires quadratic time for computing the Wasserstein gradient flow (asymptotically $\mathcal{O}(K^2 n_y)$) and the memory footprint for estimating with internal Gaussian particles is linear (asymptotically $\mathcal{O}(Kn_y)$). There are inherent trade-offs between accuracy and computational efficiency when choosing between LightSB and VMSB; nevertheless, VMSB remains significantly more manageable and computationally tractable compared to deep learning methods for moderate settings. For instance, we have presented performance regarding efficiency and scalability up to 1,000 dimensions in the experiments. Driven by parallel nature of Gaussian particles, we observed that the computation of Proposition 3 favors vectorized instructions, and the expected speed enhancement from using GPUs is much more evident in neural network cases. In Table 7, we report the wall-clock time for a 100-dimensional single-cell data problem Vargas et al. (2021); Korotin et al. (2024), where the performance is reported in Table 3. Additionally, training time in the MNIST-EMNIST translation is reported in Table 11 in the ablation study. This property also holds for generation, allowing practitioners to deploy the model much faster on GPUs. In Table 8, we also report that generating 100

MNIST samples from 4096 Gaussian particles, equipped with competitive performance, can be done 1,854 times faster under the same hardware. Since VMSB a simulation-free, the GMM generation process does not suffer from discretization errors of SDE.

**Reproducibility statement.** Comprehensive justification and theoretical background are presented in Appendices A and B. Since the primary contributions of this paper pertain to the learning methodology, we ensured that all architectures and hyperparameters remained consistent across the LightSB variants. All datasets utilized in this study are available for download alongside the training scripts. Please refer to Appendix D for more information on the experimental setups.

Table 9: EOT Benchmark scores of $\mathbb{BW}_2^2$-UVP $\downarrow$ (%).

| Type | Solver | $\varepsilon = 0.1$ | | | | $\varepsilon = 1$ | | | | $\varepsilon = 10$ | | | |
|---|---|---|---|---|---|---|---|---|---|---|---|---|---|
| | | $d = 2$ | $d = 16$ | $d = 64$ | $d = 128$ | $d = 2$ | $d = 16$ | $d = 64$ | $d = 128$ | $d = 2$ | $d = 16$ | $d = 64$ | $d = 128$ |
| | Classical solvers (best)[†] | 0.016 | 0.05 | 0.25 | 0.22 | 0.005 | 0.09 | 0.56 | 0.12 | 0.01 | 0.02 | 0.15 | 0.23 |
| Bridge-M | DSBM (Shi et al.)[‡] | 0.03 | 0.18 | 0.7 | 2.26 | 0.04 | 0.09 | 1.9 | 7.3 | 0.26 | 102 | 3563 | 15000 |
| Bridge-M | SF$^2$M-Sink (Tong et al.)[‡] | 0.04 | 0.18 | 0.39 | 1.1 | 0.07 | 0.3 | 4.5 | 17.7 | 0.17 | 4.7 | 316 | 812 |
| rev. KL | LightSB (Korotin et al.) | $0.004 \pm 0.004$ | $0.009 \pm 0.004$ | $0.023 \pm 0.003$ | $0.036 \pm 0.003$ | $0.004 \pm 0.005$ | $0.009 \pm 0.003$ | $0.016 \pm 0.002$ | $0.035 \pm 0.003$ | $0.009 \pm 0.004$ | $0.013 \pm 0.007$ | $0.034 \pm 0.004$ | $0.066 \pm 0.008$ |
| Bridge-M | LightSB-M (Gushchin et al.) | $0.005 \pm 0.003$ | $0.012 \pm 0.004$ | $0.034 \pm 0.003$ | $0.063 \pm 0.002$ | $0.005 \pm 0.001$ | $0.027 \pm 0.007$ | $0.057 \pm 0.010$ | $0.108 \pm 0.004$ | $0.004 \pm 0.002$ | $0.017 \pm 0.007$ | $0.133 \pm 0.010$ | $0.409 \pm 0.042$ |
| EMA | LightSB-EMA | $0.004 \pm 0.002$ | $0.014 \pm 0.003$ | $0.021 \pm 0.003$ | $0.044 \pm 0.001$ | $0.004 \pm 0.003$ | $0.009 \pm 0.004$ | $0.013 \pm 0.001$ | $0.032 \pm 0.004$ | $0.004 \pm 0.001$ | $0.008 \pm 0.003$ | $0.023 \pm 0.013$ | $0.010 \pm 0.002$ |
| Var-MD | VMSB (ours) | $\mathbf{0.003 \pm 0.001}$ | $\mathbf{0.007 \pm 0.003}$ | $\mathbf{0.018 \pm 0.002}$ | $\mathbf{0.039 \pm 0.001}$ | $\mathbf{0.002 \pm 0.002}$ | $\mathbf{0.004 \pm 0.001}$ | $\mathbf{0.009 \pm 0.001}$ | $\mathbf{0.023 \pm 0.003}$ | $\mathbf{0.005 \pm 0.007}$ | $\mathbf{0.006 \pm 0.004}$ | $\mathbf{0.011 \pm 0.010}$ | $\mathbf{0.011 \pm 0.004}$ |
| Var-MD | VMSB-M (ours) | $\mathbf{0.002 \pm 0.001}$ | $\mathbf{0.010 \pm 0.067}$ | $\mathbf{0.031 \pm 0.004}$ | $\mathbf{0.056 \pm 0.005}$ | $\mathbf{0.003 \pm 0.004}$ | $\mathbf{0.005 \pm 0.002}$ | $\mathbf{0.032 \pm 0.006}$ | $\mathbf{0.077 \pm 0.018}$ | $\mathbf{0.003 \pm 0.003}$ | $\mathbf{0.011 \pm 0.004}$ | $\mathbf{0.117 \pm 0.012}$ | $\mathbf{0.429 \pm 0.748}$ |

Table 10: EOT scores of $c\mathbb{BW}_2^2$-UVP, the fully extended version of Table 2.

| Type | Solver | $\varepsilon = 0.1$ | | | | $\varepsilon = 1$ | | | | $\varepsilon = 10$ | | | |
|---|---|---|---|---|---|---|---|---|---|---|---|---|---|
| | | $d = 2$ | $d = 16$ | $d = 64$ | $d = 128$ | $d = 2$ | $d = 16$ | $d = 64$ | $d = 128$ | $d = 2$ | $d = 16$ | $d = 64$ | $d = 128$ |
| | Classical solvers (best)[†] | 1.94 | 13.67 | 11.74 | 11.4 | 1.04 | 9.08 | 18.05 | 15.23 | 1.40 | 1.27 | 2.36 | 1.31 |
| Bridge-M | DSBM (Shi et al.)[‡] | 5.2 | 10.8 | 37.3 | 35 | 0.3 | 1.1 | 9.7 | 31 | 3.7 | 105 | 3557 | 15000 |
| Bridge-M | SF$^2$M-Sink (Tong et al.)[‡] | 0.54 | 3.7 | 9.5 | 10.9 | 0.2 | 1.1 | 9 | 23 | 0.31 | 4.9 | 319 | 819 |
| rev. KL | LightSB (Korotin et al.) | $0.007 \pm 0.005$ | $0.040 \pm 0.023$ | $0.100 \pm 0.013$ | $0.140 \pm 0.003$ | $0.014 \pm 0.003$ | $0.026 \pm 0.002$ | $0.060 \pm 0.004$ | $0.140 \pm 0.003$ | $0.019 \pm 0.005$ | $0.027 \pm 0.005$ | $0.052 \pm 0.002$ | $0.092 \pm 0.001$ |
| Bridge-M | LightSB-M (Gushchin et al.) | $0.017 \pm 0.004$ | $0.088 \pm 0.014$ | $0.204 \pm 0.036$ | $0.346 \pm 0.036$ | $0.020 \pm 0.007$ | $0.069 \pm 0.016$ | $0.134 \pm 0.014$ | $0.294 \pm 0.017$ | $0.014 \pm 0.001$ | $0.029 \pm 0.004$ | $0.207 \pm 0.005$ | $0.747 \pm 0.028$ |
| EMA | LightSB-EMA | $0.005 \pm 0.002$ | $0.040 \pm 0.014$ | $0.078 \pm 0.007$ | $0.149 \pm 0.006$ | $0.012 \pm 0.002$ | $0.022 \pm 0.003$ | $0.051 \pm 0.001$ | $0.127 \pm 0.002$ | $0.017 \pm 0.003$ | $0.021 \pm 0.003$ | $0.025 \pm 0.002$ | $0.042 \pm 0.002$ |
| Var-MD | VMSB (ours) | $\mathbf{0.004 \pm 0.001}$ | $\mathbf{0.012 \pm 0.002}$ | $\mathbf{0.038 \pm 0.002}$ | $\mathbf{0.101 \pm 0.002}$ | $\mathbf{0.010 \pm 0.001}$ | $\mathbf{0.018 \pm 0.001}$ | $\mathbf{0.044 \pm 0.001}$ | $\mathbf{0.114 \pm 0.001}$ | $\mathbf{0.013 \pm 0.001}$ | $\mathbf{0.019 \pm 0.001}$ | $\mathbf{0.021 \pm 0.008}$ | $\mathbf{0.040 \pm 0.001}$ |
| Var-MD | VMSB-M (ours) | $\mathbf{0.015 \pm 0.016}$ | $\mathbf{0.067 \pm 0.036}$ | $\mathbf{0.108 \pm 0.020}$ | $\mathbf{0.253 \pm 0.107}$ | $\mathbf{0.010 \pm 0.001}$ | $\mathbf{0.019 \pm 0.001}$ | $\mathbf{0.094 \pm 0.010}$ | $\mathbf{0.222 \pm 0.033}$ | $\mathbf{0.013 \pm 0.001}$ | $\mathbf{0.029 \pm 0.003}$ | $\mathbf{0.193 \pm 0.015}$ | $\mathbf{0.748 \pm 0.036}$ |

## F   Additional Experimental Results

### F.1   Additional results on the EOT benchmark

We present the full results of EOT benchmark experiments. Tables 9 and 10 show comprehensive statistics on the EOT benchmark with more SB solvers. As mentioned in § 6.2, the VMSB and VMSB-M solvers consistently brought better performance with low standard deviations of scores for $c\mathbb{BW}_2^2$-UVP and $\mathbb{BW}_2^2$-UVP measures. We note that the experiment was conducted in a highly controlled setting with identical model configurations; with all other aspects controlled and outcomes differing only by learning methods, the consistent performance gains of our work were a well-anticipated result from our theoretical analysis.

### F.2   Additional image generation results

In the unpaired EMNIST-to-MNIST translation task for the raw 784 pixel, we measured FID scores for various $K$ for the SB parameterization. We considered $K \in \{64, 256, 1024, 4096\}$ with $\varepsilon = 10^{-4}$ for our VMSB algorithm. Our observations, both qualitative and quantitative, indicate that higher modalities yield higher-quality samples. In every case of $K$, VMSB-adv outperformed its counterpart. For instance, Fig. 11 demonstrates that VMSB generates more diverse samples with high fidelity. Notably, we achieved the competitive FID score of 15.471 using a standard neural network discriminator with relatively low MSD similarity scores. As the latent VMSB model for 128-dimensional embeddings also achieved the considerably low FID score of 9.558 (Table 4), we concluded that VMSB showed promising quality improvements for the both case, and this supports the generality of our theory.

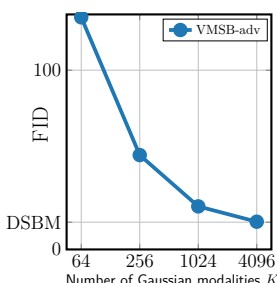

Figure 11: FID vs. modality.

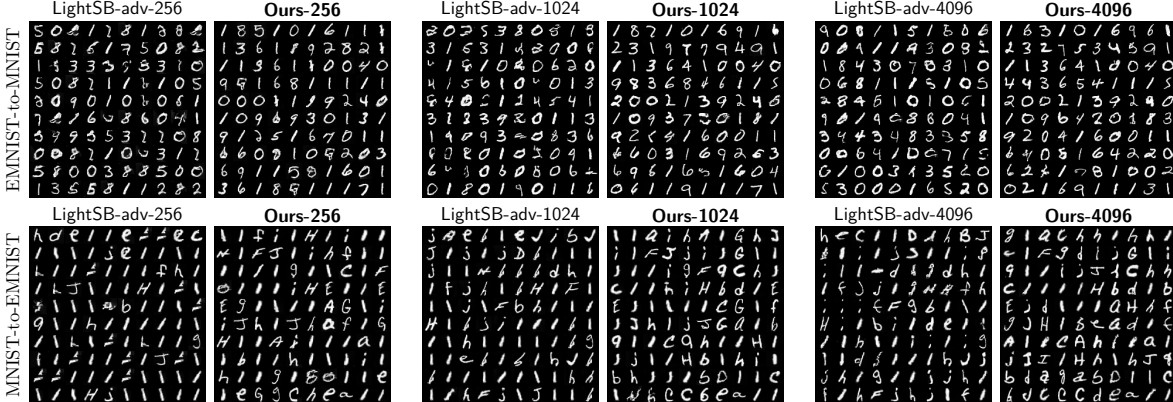

Figure 12: Generation results of unpaired image-to-image translation in the raw pixel space. We considered image data from MNIST and EMNIST (containing the first ten letters), sized as 28×28 pixels. For comparison, we trained GMM-based models with adversarial learning using a simple logistic discriminator (Table 6). This was used as both a benchmark and a tractable target SB model (LightSB-adv-$K$). VMSB in the raw pixel domain demonstrate qualitative improvements in terms of diversity and clarity of image samples.

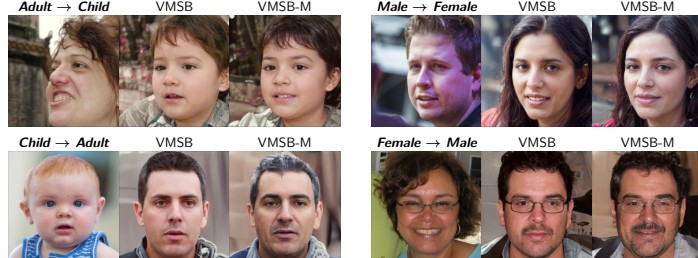

Figure 13: Image-to-Image translation on a latent space for the VMSB and VMSB-M algorithms.

Fig. 12 demonstrates that VMSB generated more diverse samples with high fidelity. Note that the proposed method suffers less from mode collapse than LightSB method (especially on the transfer MNIST-to-EMNIST), with the same Gaussian mixture setting. This result is especially a good point where the difference only lies in the learning methodology, which aligns with our theory. Tables 11 and 12 effectively show the statistics and FID scores on both the train and the test datasets. The quantitative results highlight that the VMSB solver is more performant with less overfitting than its counterpart. Consequently, our claim regarding the stability of SB solution acquisition is verified by additional experiments involving pixel spaces.

We present Embedding-ED scores (Jayasumana et al., 2024) in Table 13, and some qualitative generation results which is visualized in Fig. 9. For quantitative results, we calculated statistics from ED scores on embeddings of the ALAE model (Pidhorskyi et al., 2020), for the four different unpaired image-to-image translation tasks. The results show that VMSB is capable of translating an arbitrary representation, which

Table 11: MNIST transfer statistics.

|  | FID | Time | Parameters |
|---|---|---|---|
| LightSB-256 | 61.257 | 30m | 0.4M |
| LightSB-1024 | 26.487 | 53m | 1.6M |
| LightSB-4096 | 20.017 | 135m | 6.4M |
| VMSB-256 | 52.634 | 76m | 0.4M |
| VMSB-1024 | 24.022 | 203m | 1.6M |
| VMSB-4096 | 15.471 | 44h | 6.4M |
| DSBM-IMF | 11.429 | 42h | 6.6M |

Table 12: FID scores and differences for generated MNIST.

|  | FID (Train) | FID (Test) | Diff. (test − train). |
|---|---|---|---|
| LightSB-adv-256 | 60.746 | 61.604 | 0.858 |
| LightSB-adv-1024 | 25.934 | 26.569 | 0.635 |
| LightSB-adv-4096 | 19.960 | 20.196 | 0.237 |
| VMSB-adv-256 | 51.684 | 52.283 | 0.599 |
| VMSB-adv-1024 | 23.853 | 24.053 | 0.200 |
| VMSB-adv-4096 | 15.508 | 15.496 | −0.012 |

is closer to target domain than baselines. In Fig. 13, as well as Fig. 9, we can see that VMSB and VMSB-M algorithms generate FFHQ data with a given translation task. To qualitatively verify these generation results, we generated images using LightSB and VMSB in Figures 14 and 15. Since these improvements are purely based on information geometry and learning theory, we anticipate that following works on the variational principle application across various fields such as image processing, natural language processing, and control systems (Caron et al., 2020; Liu et al., 2023; Alvarez-Melis & Jaakkola, 2018; Chen et al., 2022).

Table 13: ALAE Embedding-ED scores. To evaluate the performance, we computed averages and standard deviations of the ED scores across four different transfer tasks.

| | $\varepsilon = 0.1$ | $\varepsilon = 0.5$ | $\varepsilon = 1.0$ | $\varepsilon = 10.0$ |
|---|---|---|---|---|
| SF$^2$M-Sink | $0.02916 \pm 0.00145$ | $0.04112 \pm 0.00191$ | $0.05670 \pm 0.00249$ | $0.06641 \pm 0.00441$ |
| DSBM-IMF | $0.02275 \pm 0.00101$ | $0.03358 \pm 0.00142$ | $0.04866 \pm 0.00168$ | $0.06474 \pm 0.00381$ |
| LightSB | $0.01086 \pm 0.00045$ | $0.02382 \pm 0.00093$ | $0.03462 \pm 0.00148$ | $0.05376 \pm 0.00273$ |
| LightSB-M | $0.01066 \pm 0.00055$ | $0.02366 \pm 0.00107$ | $0.03519 \pm 0.00153$ | $0.05975 \pm 0.00298$ |
| VMSB | $\mathbf{0.01002 \pm 0.00055}$ | $\mathbf{0.02288 \pm 0.00101}$ | $\mathbf{0.03396 \pm 0.00174}$ | $\mathbf{0.05315 \pm 0.00307}$ |
| VMSB-M | $\mathbf{0.00997 \pm 0.00054}$ | $\mathbf{0.02298 \pm 0.00106}$ | $\mathbf{0.03391 \pm 0.00140}$ | $\mathbf{0.05351 \pm 0.00241}$ |

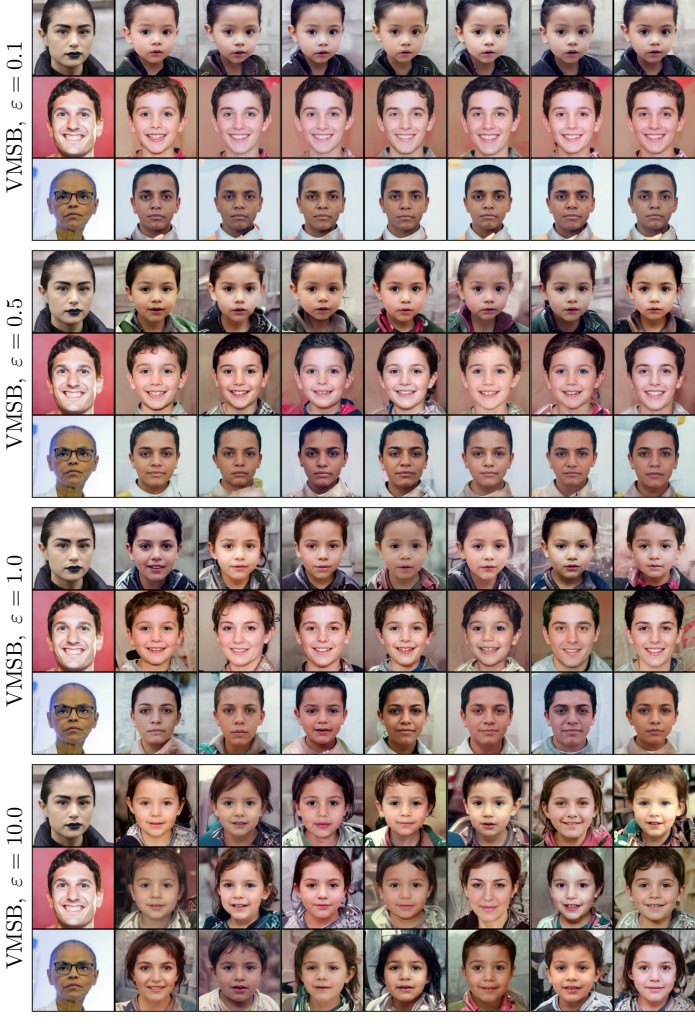

Figure 14: Generation results of VMSB (*Adult → Child*) with different volatility settings

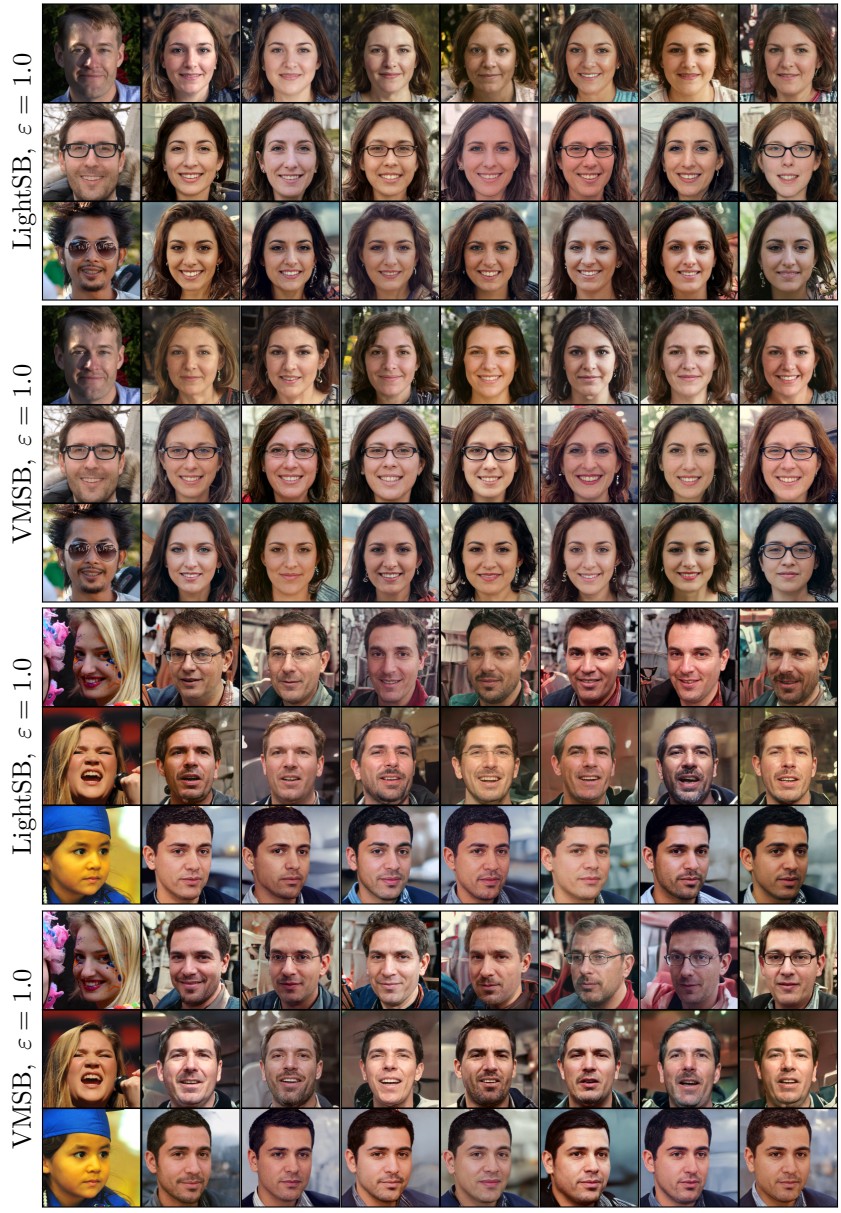

Figure 15: Qualitative comparison between LightSB and VMSB for relatively high volatility, $\varepsilon = 1.0$. Top (*Male → Female*): We find that VSBM has preserved more facial details, such as wearing glasses, than LightSB. Bottom (*Adult → Child*): VSBM was stable at retaining facial position even with high $\varepsilon$.

