# OpenReview forum: "Variational Online Mirror Descent for Robust Learning in Schrödinger Bridge"
_TMLR — Rejected by TMLR_

### Review · Reviewer_KFL8 · 2025-05-22

**Summary Of Contributions:**

The paper introduces a new algorithm, Variational Mirrored Schrödinger Bridge (VMSB), aimed at improving robustness and efficiency in solving Schrödinger Bridge problems. It uses a technique called variational Online Mirror Descent and presents theoretical analysis to support its design. The authors claim that their method performs better than existing approaches and provide experimental results to support this.

**Audience:**

Yes

**Claims And Evidence:**

Yes

**Requested Changes:**

1. Add Background: Include a clear, brief explanation of Schrödinger Bridge problems and why they are interesting or important.
2. Clarify Motivation: Explain more clearly what limitations the proposed method addresses and what kind of problems it could help solve.
3. Increase Accessibility: Consider adding intuitive overviews, visual aids, or concrete examples to make the content more approachable.
4. Improve Terminology Use: Reduce the number of acronyms or reintroduce them more frequently for clarity.

# Typos etc.
* p2. "novel learning theoretical algorithm"
* p4. "for an transportation", "from the from the", "ensure the Gateaux differentiability"

**Strengths And Weaknesses:**

# Strengths
* The paper appears to address a non-trivial challenge in probabilistic modeling.
* The algorithm is supported by theoretical results, including convergence guarantees.
* Experimental results seem to indicate strong performance relative to existing methods.

# Weaknesses
* Difficult to Understand: As someone not already familiar with Schrödinger Bridge problems, I found the paper very hard to follow. Key ideas, motivations, and assumptions are not explained in an accessible way.
* Too Many Acronyms: The paper introduces a large number of acronyms (e.g., OMD, SB, WFR) with little repetition or reminders, which makes the reading experience harder, especially for newcomers.
* Limited Background Explanation: There is not enough explanation of what Schrödinger Bridge problems are or why solving them matters.
* Heavy Technical Density: The paper could benefit from more intuitive explanation, diagrams, or examples to help non-expert readers engage with the content.

---

> ### Author Response · Authors · 2025-06-13
> **Response to Reviewer KFL8**
>
> We would like to thank **Reviewer KFL8** for the thoughtful feedback and for highlighting our theoretical and empirical contributions in the **Strengths** section. We are pleased to report that most of your suggestions have been incorporated into the revision. Below, we respond to each point in detail.
>
> ## **Weaknesses**
>
> - **Difficulty.** We acknowledge that our contributions, which come from the intersection of convex analysis, optimal transport, and variational methods, can make the paper challenging to follow. To ease comprehension, **Sections 3 and 4** have been reorganized and expanded with additional explanations and guiding remarks. We have also improved the writing style regarding theoretical claims, making our assumptions and central claims explicit.
> - **Acronyms.** While a comprehensive acronym list appears in the appendix, we agree that early sections would benefit from full terminology and, where helpful, their reintroduction in each section. We therefore replaced or delayed several acronyms in **Sections 1–3** to improve first-read accessibility and fully replaced the term WGF to Wasserstein gradient flow. However,  we retain moderate acronym use in **Sections 4 and 5**, where brevity is necessary for technical development.
> - **Background.** We understand that the original background section might appear a bit technical and overwhelming. The revised **Section 2** now opens with a concise overview of the Schrödinger bridge problem, its significance to machine-learning research, and how our approach differs from prior work, providing a smoother entry point for readers less familiar with the topic.
> - **Technical Density.** We believe **Figures 1–6** are indeed well-curated visual resources for presenting our research; we think that these were not sufficiently supported by engaging text and clear explanations in the previous version. To resolve technical density in some parts, we have inserted intuitive commentary and expanded introductory paragraphs to balance technical depth with readability.
>
> ## **Requested Changes**
>
> - **Add Background.** As suggested, **Section 2** now contains an extended background that motivates Schrödinger bridges and delineates our contributions.
> - **Clarify Motivation.** **Section 1** explicitly explains why online mirror descent is well-suited to learning Schrödinger bridges and why we favor a particular variational formulation.
> - **Increase Accessibility.** The manuscript now features more goal-oriented sentences, reduced acronym usage, and with detailed explanation to each step to be accessible to a broader audience.
> - **Improve Terminology Use.** Terminology has been improved through additional explanation, with careful adoption of acronyms only where they genuinely streamline our discussion.
>
> ## **Minor Comments**
>
> All typographical errors identified by you and the other reviewers have been corrected.
>
> Please let us know if further clarification is needed during the discussion phase.
>
> Sincerely,
>
> Authors of Submission #4592

---

### Review · Reviewer_X26F · 2025-05-22

**Summary Of Contributions:**

This work builds upon previous research connecting the relaxed Optimal Transport problem (and its dynamic counterpart, known as the Schrödinger bridge) with Mirror Descent. The authors extend this line of work to the online setting, providing a mathematical analysis of the proposed framework. They introduce a corresponding algorithm and illustrate its performance through numerical experiments.

**Audience:**

Yes

**Broader Impact Concerns:**

I do not foresee any negative broader impact concerns arising from this work. The contributions are primarily theoretical and algorithmic in nature, and the methods proposed are general tools that are not targeted at specific sensitive applications.

**Claims And Evidence:**

Yes

**Requested Changes:**

1/ The background section (Section 3) should be significantly expanded. Improving this part would clearly enhance the paper’s readability and broaden its accessibility. It is somewhat surprising that the conference version of [Karimi, Hsieh, Krause, 2024], despite strict space constraints, contains more background on key concepts than the present submission. You should take full advantage of TMLR’s lack of page limits.

In particular, several useful reminders currently placed in Appendix B.2 could be integrated into the main text. For example, the Benamou-Brenier formulation of optimal transport is clearly recalled in this appendix, but there is no in-text reference to it. As a result, I had to consult external references unnecessarily.

Although I am familiar with Optimal Transport, I was less familiar with its relaxed version and with the Sinkhorn method. That said, their core ideas are not difficult to grasp. I would therefore recommend the following improvements to make the exposition more self-contained:

  -  Briefly recall the classical Optimal Transport problem, or at least mention that it is recovered when $\varepsilon = 0$ in Equation (2).

   - Include a formulation of the dual problem for classical OT.

  -  Provide a more detailed presentation of the Sinkhorn algorithm. Once the dual is recalled, this becomes clearer. You could also reference results that connect the Sinkhorn algorithm to the primal formulation, such as Theorem 2 of [F. Léger, 2021].

 -   Clarify the mathematical meaning of solving the dynamic Schrödinger Bridge (SB) problem. Equation (4) is not sufficiently explained. For instance, solving for a path in the space of probability measures is a complex task — a brief explanation of how this is usually handled (e.g., by solving a dual formulation) would be helpful.

-    Within this mathematical framework, clearly indicate the specific contribution of the present paper.

2/ Regarding Theorem 1, its proof seems to appear only implicitly at the end of the proof of Lemma 13. This is hard to follow and gives the impression that only the second part of the theorem is addressed. I suggest writing a dedicated {proof} entitled “Proof of Theorem 1” that gives the full proof, including the main steps (e.g., "For the first part, we apply Lemma X and then Lemma Y... because these assumption are satisfied...").

3/ Consider making some comments regarding Weakness 2/.

$\textbf{Minor Comments:}$

1/   What is the space $\mathbb P(\mathcal S, [0,1])$ ?  Perhaps you meant, after Equation (4),  $\mathbb P([0,1], \mathcal S)$, with $\mathcal T_0 = \mu$ and $\mathcal T_1 = \nu$?

 2/ It would be better to have only notation for the set of probability measures: you use $\mathcal P$ at the beginning of Sec. 3 and $\mathbb P$ when it comes to stochastic processes. Moreover the space you write $\mathbb P([0,1],\mathcal S)$ does not seem clearly defined to me: why is it $a$ space of path measures? Is there any results regarding the regularity of the solutions of dynamic SB ? If it continuous for instance, you could write $C([0,1], \mathcal P(\mathcal S))$ which is clearly defined.

3/    Please recall the definition of KL divergence, especially in Equation (4), where it is applied to paths. Clarify how KL is defined in this case.

  4/  Is it necessary to formulate the theory in full generality using abstract topological spaces? If not, it might be clearer to define first variations only for probability distributions, as in Definition 7.12 of [Santambrogio, Optimal Transport for Applied Mathematicians, 2015].

 5/   At the beginning of Appendix B.2, you write “the equation is generally referred to as the Monge–Ampère equation,” but no equation is actually stated. Please clarify that you refer to the PDE satisfied by the optimal transport map and write it explicitly.

**Strengths And Weaknesses:**

As a non-specialist in this specific subfield, it is difficult for me to assess how significant the contributions are. I will defer to other reviewers for an evaluation of the novelty and importance of the results.

My main focus is therefore on clarity and accessibility: to what extent a reader unfamiliar with all the references can understand the core topic and contextualize the authors’ contributions within the existing literature.

$\textbf{Strengths:}$

 1/   The literature review appears to be thorough and well-structured.

 2/    The paper is quite well written, and the figures are of high quality and effectively support the text.

$\textbf{Weaknesses:}$

1/    I found it necessary to consult many of the cited references in order to grasp the main ideas. In particular, the paper provides little to no background on Mirror Descent, which is central to the contribution. I elaborate further in the “Requested Changes” section.

 2/   I did not focus extensively on the technical core of the paper, as I mainly spent time trying to understand the foundations. However, it seems that the theoretical analysis is conducted under idealized assumptions, and that the proposed algorithm is an approximation of this theoretical framework. A brief explanation in plain terms—e.g., sentences such as “Algorithm 1 is an approximation of (...), where [...] is approximated by [...]”—particularly at the beginning of Section 5, would greatly help non-expert readers and improve the accessibility and impact of the work.

---

> ### Author Response · Authors · 2025-06-13
> **Response to Reviewer X26F (1/2)**
>
> We appreciate **Reviewer X26F** for the thoughtful and detailed feedback. Your comments, especially those in the **Requested Changes section**, have greatly improved the clarity and readability of our manuscript. We address each point below.
>
> ## **Weaknesses**
>
> - **Background on Mirror Descent.**  We acknowledge that our original description of the MD algorithm [1] was too brief for readers. Our aim had been to distinguish the classical algorithm from the recent extension to probabilistic models by
>  Aubin-Frankowski et al. [2] (**Section 3** and **Appendix A**), which is directly applicable to Schrödinger-bridge problems. Following your suggestion, we have expanded the introductory MD overview in **Sections 1 and 2**. We also added remarks explaining why our setting requires additional assumptions compared with the classical OMD framework.
> - **Idealized assumptions and its explanation.** As the reviewer pointed out, our analysis is also based on multiple assumptions such as asymptotical log-concave distributions (**Assumption 1**) and an asymptotic condition on the dully stationary process (**Assumption 2**). We found that these conditions play critical roles analogous to strong convexity and smoothness in convergence proofs of the classical theory. While these assumptions may appear idealized, we respectfully point out that we have empirically validated them through intuitive and numerical experiments (**Figs. 4 & 5; Section 6**). Following your fruitful suggestion, we have added more justification in **Section 5**.
>
> ## **Requested changes**
>
> We again thank the reviewer for helpful recommendation. Below we list the concrete revisions made in response to your detailed guidance.
>
> > **The background section (Section 3) should be expanded**
>
> As this point is raised my multiple reviewers, we enhanced clarity of **Section 3** and increased its volume.
>
> - **Appendix B.2 could be integrated into the main text.** We generally agree with your opinion. Considering overall theme of Section 3 and the dynamic Benamou-Brenier formulation of optimal transport,  we rewrote the paragraph of dynamic Schrödinger bridge part and referenced **Appendix B.2**.
> - **Recall the classical optimal transport problem.** We acknowledge that previously this fact was only partially mentioned in either the related work section or the appendix. As suggested, we clearly indicated that the classical OT problem is recovered when ε = 0 in **Section 2**.
> - **Include a formulation of the dual problem for OT & Provide a more detailed presentation of the Sinkhorn algorithm.** Thank you for great suggestion. The log-Schrödinger potentials, which are the dual parameters naturally arise from the duality of static SB problem. Hence, we included dual problem of entropic optimal transport (Theorem 3.2 of Nutz, 2021 [3]) in Section 3. We also strengthen explaining the Sinkhorn algorithm by referencing results that connect the Sinkhorn algorithm to the primal formulation.
> - **Clarify the mathematical meaning of solving the dynamic SB problem.** We agree. However, the dynamic SB problem could appear to be an overly complex subject to general readers. Therefore, we made a new dedicated appendix (**Appendix C**) for explaining the full story of the dynamic SB problem for ones that are interested.
> - Following your suggestion, we also clearly indicated the contributions of the paper in **Sections 1-3**.
>
> > **Regarding Theorem 1, its proof seems to appear only implicitly … I suggest writing a dedicated proof.**
>
> - Thank you for the suggestion. We enhanced the proof of theorem by making its conclusion part more explicit.
>
> > **Consider making some comments regarding Weakness 2.**
>
> - As suggested, we clarified the statement and provide relevanet context for each assumption.

---

> > ### Author Response · Authors · 2025-06-13
> > **Response to Reviewer X26F (2/2)**
> >
> > ## **Minor Comments**
> >
> > - **Notation** $\mathbb{P}(\mathcal{S}, [0,1])$**.** We clarify that path measure space for $\mathcal{T}_0=\mu$ and $\mathcal{T}_1 = \nu$ as $\mathbb{P}([0,1], \mathcal{S})$.
> > - **It would be better to have only notation for the set of probability measures.** We sincerely appreciatethis insightful suggestion. While we have discussed adopting the notation $\mathcal{C}([0,1], \mathcal{P}(\mathcal{S}))$, we require additional time to assess its full implications, and will decide on its inclusion before the camera-ready submission.
> > - **Clarify how KL is defined in this case.** As suggested, we added the definition of KL divergence in **Section 3**.
> > - **Is it necessary to formulate the theory in full generality using abstract topological spaces?** We thank the reviewer for bringing the alternative definition in [5] to our attention. We are aware of this reference but choose the definition stated for the notational preference and the ease of technical usage in proof. We incorporated a remark Definition 7.12 from [5] to **Section 3** in **The Bregman divergence** paragraph.
> > - **Please clarify that you refer to the PDE satisfied by the optimal transport map and write it explicitly.** The Monge–Ampère equation is now written explicitly at the start of **Appendix B.2**, together with a short commentary on regularity. We also provides the new appendix (**Appendix C**) that describes background of the dynamic SB problem.
> >
> > Please let us know if there are further concerns or suggestions during the discussion phase.
> >
> > Sincerely,
> >
> > Authors of Submission #4592
> >
> > [1] Arkadiǐ Semenovich Nemirovsky and David Borisovich Yudin. Problem complexity and method efficiency in optimization. A Wiley-Interscience publication. Wiley, 1983.
> >
> > [2] Pierre-Cyril Aubin-Frankowski, Anna Korba, and Flavien Léger. Mirror descent with relative smoothness in measure spaces, with application to Sinkhorn and EM. In NeurIPS. 2022.
> >
> > [3] Marcel Nutz. Introduction to entropic optimal transport. Columbia University, 2021.
> >
> > [4] Flavien Léger. A gradient descent perspective on Sinkhorn. Applied Mathematics & Optimization, 84(2):1843–1855, 2021.
> >
> > [5] Filippo Santambrogio. Optimal transport for applied mathematicians. Birkäuser, NY, 55(58-63):94, 2015.

---

### Review · Reviewer_uevG · 2025-05-30

**Summary Of Contributions:**

This work introduces two new algorithms, VOMD and VMSB, for solving Schrödinger Bridge (SB) problems.

[1]. The authors developed a Schrödinger Bridge (SB) learning algorithm based on an Online Mirror Descent (OMD) framework with theoretical guarantees of convergence under mild conditions.

[2]. They introduced VMSB, a simulation-free SB method that uses Wasserstein-Fisher-Rao geometry to guarantee stability. This approach allows for a straightforward and efficient implementation.

[3]. The proposed algorithms were tested in various scenarios, including online learning, Entropic Optimal Transport (EOT) benchmarks, and image-to-image translation.

**Audience:**

Yes

**Broader Impact Concerns:**

N/A.

**Claims And Evidence:**

Yes

**Requested Changes:**

[1]. The paper's introduction would be strengthened by a dedicated discussion of the specific limitations and bottlenecks of existing methods. Clearly articulating these challenges first would better motivate the proposed approach and highlight how it is designed to overcome them.

[2]. The confidence intervals in the experiments are noted to be based on 10 trials. To provide a more robust validation of the proposed algorithm's advantages, it would be beneficial to increase the number of trials. This would likely narrow the confidence intervals, potentially demonstrating a clearer and more statistically significant performance gap between the proposed method and others, especially in cases where the intervals currently overlap.

[3]. A minor point regarding the "SB on single-cell dynamics" experiment: while PCA is a valid technique for dimensionality reduction, it can create features that are difficult to interpret biologically. In biomedical contexts where feature interpretability is crucial, employing a feature selection method (e.g., based on t-tests or other importance metrics) instead of PCA might better emulate real-world applications and yield more clinically relevant insights.

**Strengths And Weaknesses:**

Strengths:

[1]. The theoretical contributions are a key strength. The paper establishes a convergence proof and a regret bound for the proposed algorithms. The paper provides a mathematical guarantee of the algorithm's learning efficiency.

[2]. The theories and methodologies are validated on both synthetic and real-world experiments. The algorithm demonstrates superior robustness over existing methods across a range of tasks, including synthetic data and challenging real-world high-dimensional datasets.

[3]. The paper provides a comprehensive review of related work and introduces the necessary prerequisites.

Weaknesses:

[1]. As a reader who is not an expert in this specific area, I found it difficult to grasp the key bottlenecks, the limitations of current methods, and the overall motivation for this work.

[2]. The practical significance of the results is not compelling. For example, in Figure 3, the authors claim that "VMSB achieved the best results." However, this performance gain appears to be statistically insignificant, as the confidence intervals for VMSB seem to overlap with those of other methods.

---

> ### Author Response · Authors · 2025-06-13
> **Response to Reviewer uevG**
>
> We thank **Reviewer uevG** for the constructive feedback and for highlighting our contributions. We have carefully incorporated your suggestions in the revised manuscript. We address each of your comments in detail below.
>
> ## **Weaknesses**
>
> - **On the motivation and limitations of current methods.** We appreciate you highlighting the need for a clearer articulation of our work's motivation. The central goal of our paper is to address the critical, yet often overlooked, role of uncertainty in the training of probabilistic generative models. We have substantially revised the introduction (**§ 1**) to better contextualize our work. The new text now explicitly details the key bottlenecks and limitations of existing SB methods, thereby establishing a stronger foundation for our proposed approach, VMSB. We are confident that these revisions will help readers better appreciate our paper's contribution to the TMLR community—namely, providing a promising theoretical link between diffusion models and learning theory.
> - **On the significance of the results (Table 3).** Thank you for the detailed comment regarding our experimental results. We believe this comment refers to **Table 3**, not Figure 3. The experiments on single-cell dynamics presented therein directly adhere to the protocol established by Tong et al. [1], which is now a standard benchmark for evaluating simulation-free SB solvers. This ensures a direct and fair comparison with prior art. Your comment made us realize that our description in **§ 6.2** could be improved for clarity. We have expanded on this in the **Requested Changes** section below and have revised the manuscript accordingly to make the significance of these results more apparent.
>
> ## **Requested Changes**
>
> - **Clarity on the limitations of existing methods.** We agree that our previous manuscript's description of the limitations of prior work (e.g., "...methods of SB remain somewhat *atypical*...") was too nuanced. Taking your suggestion, we have expanded **§ 1** to provide a more direct and thorough discussion of the specific bottlenecks in current methods. Furthermore, we have added a clearer, more accessible motivation for employing mirror descent, particularly for readers who may not be deeply familiar with the theory of Wasserstein gradient flows.
> - **Clarification of experimental details:** We appreciate the call for greater clarity. We respectfully clarify that all key results in our experiments were averaged over multiple trials. Specifically for the single-cell dynamics in **Table 3**, the statistics are derived from **ten distinct runs**, as noted in the revised caption. These ten runs comprise two separate experimental settings, each performed with five different random seeds: (1) Start = Day 2, End = Day 4 (Evaluation at Day 3) - 5 runs, and (2) Start = Day 3, End = Day 7 (Evaluation at Day 4) - 5 runs.  This setup mirrors the original configuration from Tong et al. [1]. While we acknowledge the overlapping 95% confidence intervals, we maintain that VMSB achieves competitive performance on this challenging benchmark. It is also worth noting that this experiment uses real-world biological data that has undergone PCA preprocessing. Consequently, the SB's qudratic cost may not perfectly capture the intrinsic manifold of cell dynamics. We have added these crucial details to the caption of **Table 3** and the main text in **§ 6.2** to ensure our evaluation protocol is transparent.
> - **Regarding the single-cell dynamics experiment:** We agree that PCA, while a standard preprocessing step, may limit the direct biological interpretability of the learned dynamics. Our primary objective for including this experiment was the "Quantitative Evaluation" of VMSB's performance against an established benchmark for simulation-free SB solvers, with biological interpretation being a secondary goal. To transparently address your valid concern, we have now added a discussion of this limitation to the conclusion of the revised manuscript.
>
> We hope these revisions and clarifications have fully addressed your concerns. We are grateful for your guidance, which has helped us improve the quality of our paper, and we welcome any further suggestions during the discussion phase.
>
> Sincerely,
>
> Authors of Submission #4592
>
> [1] Alexander Tong, Kilian Fatras, Nikolay Malkin, Guillaume Huguet, Yanlei Zhang, Jarrid Rector-Brooks, Guy Wolf, and Yoshua Bengio. Improving and generalizing flow-based generative models with minibatch optimal transport. TMLR, 2024.

---

### Review · Reviewer_QGaD · 2025-06-02

**Summary Of Contributions:**

This paper introduces Variational Mirrored Schrödinger Bridge, a simulation-free algorithm to solve Schrödinger bridge problems by combining variational inference with online mirror descent in the Wasserstein-Fisher-Rao geometry. The authors propose a novel theoretical formulation based on these ideas, derive theoretical convergence and regret bounds for their approach, and present a practical instantiation of their idea using a Gaussian mixture variational approximation. Experimental results are provided across synthetic datasets, entropic optimal transport benchmarks, and high-dimensional applications like image translation and single-cell datasets.

**Audience:**

Yes

**Broader Impact Concerns:**

No broader impact concerns.

**Claims And Evidence:**

No

**Requested Changes:**

- Several of the weaknesses noted above include actionable suggestions. I encourage the authors to address those directly.
- As mentioned in the “Strengths” section, I believe the theoretical results have potential value. One way to improve the paper would be to clearly delineate what is new relative to prior work, define a specific theoretical goal, and develop results that directly support that goal. To make these results accessible and meaningful, the paper would benefit greatly from a comprehensive background section. This section should carefully define the mathematical objects in use, state results precisely and correctly, and provide intuition to help the reader understand their significance.
- The paper draws on a wide range of advanced mathematical concepts. I strongly encourage the authors to expand the appendix with background material and intuition for at least the core topics necessary to follow the main text. At a minimum, any non-standard or technical mathematical claim should be accompanied by a precise reference (textbook, appendix section, or paper). While some background is included in the current appendix, its clarity and integration with the main text could be significantly improved. In many cases, useful background content is present but not referenced — forcing the reader to look elsewhere (e.g., in Karimi et al., 2024) to understand the claims.

    Some examples:

    - The paragraph between Eqs. (2) and (3), which discusses the relationship between the primal and dual solutions, either needs a citation or should be derived in the appendix.
    - The sentence “The fundamental equivalence between static and dynamic SBPs (Pavon & Wakolbinger, 1991; Léonard, 2012) allows us to consider the optimal coupling when finding the SB process, vice versa” is unclear. Is the citation meant to support the equivalence claim? If so, it should be placed accordingly — otherwise, an additional reference is needed.
    - The discussion spanning pages 4 and 5 lacks coherence and contextualization. For instance, "locally convex topological vector space" is introduced without definition or justification. (Side note: is this level of abstraction necessary for the practical goals of the paper, or can the framework be simplified?) This section appears to follow presentation and results from other papers with minimal explanation. A self-contained and clearly motivated presentation would be much more helpful to the reader.
- The assumptions underpinning the theoretical results are quite dense and abstract. It would be helpful to explain what they mean in practice: What is their intuition? Are they likely to hold in common SB settings? Can they be easily verified or approximated in applications? A more precise and practical discussion of these assumptions would significantly improve the paper’s clarity and usefulness.
- Finally, please make it explicit in the main text that the appendix includes extensive tables and figures related to the experiments, including full results and standard errors (e.g. Tables 9 and 10). These are important for evaluating the claims and should be more clearly referenced.

### Minor comments:

- There are quite a bit of typos and syntactic or grammatical issues throughout the manuscript. A thorough proofreading would be beneficial. Some specific examples include:
    - The first sentence in the “Wasserstein gradient flows” paragraph of the *Related Work* section.
    - The phrasing in *Lemma 1* is unclear. is → are.
    - *Assumption 2* is written in a confusing way.
- Figure 1 is visually informative and well-designed. However, it is underutilized in the text. I strongly suggest incorporating a detailed discussion of this figure after the introduction of the main conceptual framework or the algorithm. Right now it is unfortunately barely referenced.
- Several mathematical concepts and notations are introduced before they are properly defined — or are not defined at all. Some examples include:
    - In the *“MD and Sinkhorn”* paragraph of the *Related Work* section: the phrase “our noisy setup” appears without prior mention or context. What is noisy here?
    - At the start of *Section 3*: the term “transportation plan” is used — presumably referring to a coupling — but this should be made explicit. For clarity, I recommend using consistent terminology across the manuscript.
    - The *KL functional* is referenced without definition. While the Kullback-Leibler divergence may be familiar to many readers, its precise form — especially in the infinite-dimensional setting — should be stated clearly.
    - After Equation (2): function spaces like L1 are invoked but never defined.
    - At the top of page 5, the statement “which enforces the Gibbs parameterization for the couplings” → what is this Gibbs parametrization?

Overall, I believe the paper addresses a very interesting topic and brings together ideas from multiple literatures in an intriguing way. However, its impact is significantly limited by the quality of the exposition and the lack of clear goalposts, background, and guiding intuition. In its current form, the paper is accessible only to a narrow audience already well-versed in the technical details. Given the broader appeal and relevance of the topic, I strongly encourage the authors to revise the manuscript with a focus on clarity and accessibility, so that a wider range of readers — e.g., within the TMLR community — can engage with and appreciate the contributions.

**Strengths And Weaknesses:**

**Strengths:**

- The main strength of the paper is the theoretical development in Section 4. Though my understanding is limited (see below for reasons), the results seem promising from a theoretical standpoint.
- The figures are of high quality and, in most cases, effectively support the arguments presented in the corresponding sections of the paper.

**Weaknesses:**

- The presentation is challenging to follow. Key definitions, assumptions, and motivations are frequently introduced without sufficient context or are buried in dense mathematical formalism, making it difficult to understand the core ideas and contributions. The conference papers that this work builds upon — such as Aubin-Frankowski et al. (2022) and Karimi et al. (2024) — despite stricter page limits, do a better job at providing background, intuition, and a clear narrative. In contrast, this paper is difficult to parse unless the reader is already quite familiar with the relevant literature and technical machinery. As a result, I found it hard to identify the main message, the specific contributions, and the practical implications of the theoretical framework. It is also unclear which assumptions are necessary for the theory to hold, whether they are easily satisfied in practice, or whether approximations are always required (as seems to be the case in the proposed approach).
- The motivation for introducing an *online* formulation of the Schrödinger Bridge (SB) problem is not clearly articulated. The paper does not convincingly explain why online updates are necessary in this context, and the experiments do not isolate or demonstrate the practical benefits of adopting an online approach. Moreover, the use of variational approximations and mirror descent are not themselves novel, so it remains unclear what the distinctive contribution of this work is. I do think the idea of applying online learning to SB is intriguing, and the theoretical results could be valuable — but it's not clear why or when this approach is needed in practice. What concrete advantages does it provide? What problems does it solve better than existing methods? While the authors seem to be aiming to address these questions, the current presentation makes it difficult to extract a clear answer, and I am left unsure whether I fully understood the core motivation.
- The paper lacks a clear narrative arc: the motivation, methodology, and experimental results do not come together to form a cohesive story. This is, in my view, the paper’s most significant weakness. The work could be substantially improved by first offering an intuitive discussion of the shortcomings of current methods — accessible to a general TMLR reader — and then clearly stating the goals of the paper in the introduction. The rest of the paper should be structured around achieving these goals. In particular, the experimental section should directly support the core narrative. Additionally, I strongly recommend that the authors avoid introducing dense mathematical constructs without first explaining the problem being addressed, the rationale for the approach (i.e. why do we need this mathematical object that we’re defining?), and some guiding intuition.
- Many of the technical sections draw heavily from prior work — particularly Aubin-Frankowski et al. (2022), Korotin et al. (2024), and Karimi et al. (2024) — without clearly distinguishing which parts are novel. The paper would benefit significantly from a more explicit separation between previously established results and new contributions. For instance, the authors could adopt a more active voice when introducing their own results and ensure that references are consistently provided when building on prior work. While this is done well in several places, some sections — such as section 4.1 — remain ambiguous in this regard and could use more clarity.
- The experiments do not clearly validate the theory. While results are reported on standard benchmarks, they are not clearly tied back to the claimed theoretical contributions (e.g., regret bounds, online robustness). This goes back to my previous point, i.e. the fact that the paper is missing clear goalposts and thus I am not sure what these experiments are showing. In particular, if the online idea is the main innovation here, I would expect some experiments explicitly targeting the gain in performance that we have by having online MD. For example an ablation study on this component of the algorithm. Also: I am not sure in general the results are very convincing. The authors report mean and standard errors over few seeds, and boldface the “best method”, but it is not clear how the best method is defined. From my understanding, they highlight the best method to be the method with best mean, but there are — in many experiments — other methods whose results’ confidence interval overlaps with the best method’s one. Why are these not in boldface too? I don’t think we can be so sure that just having a better mean, especially if overlapping intervals, must define one to be the best method. But also, in general I would appreciate if the authors made more precise how they color in bold the results (right now it only says “Our VMSB and VMSB-M results are highlighted in bold when VMSB methods exceed their reference algorithm.”)
- The experiments do not clearly validate the theoretical claims. While results are reported on standard benchmarks, they are not explicitly connected to the core theoretical contributions — such as the regret bounds or the robustness of the online formulation. This reflects a broader issue noted earlier: the lack of clear goalposts makes it difficult to understand what the experiments are meant to demonstrate. If the online mirror descent aspect is intended to be the key innovation, then I would expect targeted experiments that directly assess its impact — for instance, an ablation study isolating the benefits of the online component. More broadly, I find the empirical results somewhat unconvincing. The authors report means and standard errors across a small number of random seeds, and highlight the “best” method in boldface. However, it is unclear how “best” is defined. From what I can infer, it seems to be based on the lowest mean value, but in many cases, the confidence intervals of other methods overlap with the reported best. In such cases, I do not think it is appropriate to highlight one method as clearly superior. If the difference is not statistically significant, this should be acknowledged. Additionally, the current explanation — “Our VMSB and VMSB-M results are highlighted in bold when VMSB methods exceed their reference algorithm” — is ambiguous. I recommend the authors clarify precisely how boldfaced entries are determined.

---

> ### Author Response · Authors · 2025-06-13
> **Response to Reviewer QGaD**
>
> We especially thank Reviewer **QGaD** for the insightful feedback and detailed suggestions, and we appreciate the encouraging remarks on our theory and figures in the **Strengths** section. We have uploaded a comprehensive revision that addresses the concerns outlined in the **Weaknesses** and **Requested Changes** sections. We respond to your comments below.
>
> ## **Weaknesses**
>
> > **The presentation is challenging to follow.**
>
> Since this point has been raised by multiple reviewers, we have revised the manuscript extensively to improve clarity.
>
> - We expanded the volume of **Sections 1 and 3,** with more detailed explanations of key concepts and contextual background, ensuring that general readers can fully grasp our contributions.
> - In **Section 3**, we have restated **Assumption 1** more clearly and, in **Section 5**, added intuitive commentary to explain why each approximation step is necessary.
> - **Section 6** now includes a clearer statement of our experimental objectives and more precise descriptions of the setup and evaluation metrics.
>
> Please let us know if there are remaining concerns on the presentation style.
>
> > **The motivation for online learning formulation of the SB problem is not clearly articulated.**
>
> We thank the reviewer for bringing out this topic to our attention. We would like to address your concern with the following points.
>
> - We agree that the motivation for online learning for the SB problem is not fully expressed in the submission. To resolve this issue, we improved readability of **Sections 1 and 4** to help readers grasp the core arguments. Notably, we included a succinct discussion of why online learning at the beginning of **Section 4.2**.
> - Existing SB algorithms exhibit idealized (wishful) assumptions that learning objectives can be precisely approximated. This work bets against such an idea, and we emphasize that this should be resolved by more than a minor algorithmic tweak. Our online learning hypothesis addresses a fundamental gap between theory and real‐world applications of SB solvers.
> - We believe the necessity of the learning theoretical direction comes both theory and practice.
>     - From a theoretical perspective, our convergence analysis in **Section 4** establishes convergence analysis with a regret bound for OMD applied to SB, situating the problem within the statistical learning framework rather than treating it as an ad-hoc problem.
>     - Practically, by drawing on recent studies of Sinkhorn algorithms and flows [1, 2], we implement an OMD solver whose performance we verify on both our custom experiments and standard SB benchmarks.
> - We respectfully claim the novelty of our approach. The VMSB algorithm emerges as a robust solver that embeds OMD within a variational formulation, offering both rigorous theoretical guarantees and clear computational advantages. To the best of our knowledge, some variational approaches for mirror descent were present, but our approaches used a complete computational OMD algorithm that finds a strong connection between classical learning theory and Schrödinger bridge.
>
> > **The paper lacks a clear narrative arc.**
>
> Thank you for the great suggestions. We also believe ensuring the paper’s coherence and cohesion is essential for making our paper more accessible to general audience. To fully resolve this issue, we made significant revision regarding clarity. Here is a summary of modifications.
>
> - **Section 1.** We clarified limitations of existing methods, which were a bit nuanced in the submission. We also put more description on mathematical concepts and our intention associated with our motivation.
> - **Section 3.** We significantly increased the sheer volume of the background of Schrödinger bridge. We included Wasserstein-2 distance, the duality lemma, justifications on dynamic SB problem and Bregman divergence.
> - **Sections 4 & 5.** We improved the flow of theory and  helpful narrative at the start or end of each methodological step, and we also enhanced the readability of assumptions.
> - **Section 6.** We enhanced the readability of experimental design and description of particular experiments.
>
> > **Many of the technical sections draw heavily from prior work. … The rest of the paper should be structured around achieving these goals.**
>
> Although our methodology and computational strategy build on established ideas, this paper deliberately integrates them into a unified framework under the singular framework of VOMD. We thank the reviewer for bringing us to the attention that the submission needs more emphasis as a cohesive story. Following your fruitful suggestion, we  enhanced structural integrity of writing, provide missing rationale for the approach, provide useful guidelines and reasons for pursuing each step.

---

> > ### Author Response · Authors · 2025-06-13
> > **Response to Reviewer QGaD (2/2)**
> >
> > > **The experiments do not clearly validate the theory. … The experiments do not clearly validate the theoretical claims.**
> >
> > We would like to highlight the following points.
> >
> > - We respectfully emphasize that our theory of OMD was directly validated by the actual experiment in **Figures 4, 6, 7** where each experiment effectively demonstrates a dedicated part on the theoretical assumptions or claims, respectively.
> > - We consider the necessity of OMD from the fundamental problem of SBP as an optimization problem; hence, we argue that the performance gains from standard benchmark in **Section 6** are important to verify the generality and significance of our online learning hypothesis.
> > - To fully resolve this issue, we have enhanced the readability of our experiments clarifying our experimental goals in the revision.
> >
> > ## **Requested Changes**
> >
> > - **Clearly delineate what is new related to prior work, define specific theoretical goals, and develop results that directly support that goal.** As we mentioned, we enhanced the overall readability and structural integrity to represent our theoretical goals.
> > - **I strongly encourage the authors to expand the appendix with background material and intuition for at least the core topics necessary to follow the main text.** We are pleased to report that **Section 3**  has been overhauled to incorporate foundational material on Itô’s lemma, the Feynman–Kac formula, and SB–FBSDE theory with new **Appendix C**.  We initially thought such derivations that might burden non‐experts, but we have decided to provide the full theory.
> > - **While some background is included in the current appendix, its clarity and integration with the main text could be significantly improved.**
> >     - *The paragraph between Eqs. (2) and (3).* We included the definition of KL functional, the duality lemma of SBP, and more detailed clarification of static SBP.  **
> >     - *“The fundamental equivalence between static and dynamic SBPs …, vice versa”* ****We significantly enhanced the description of dynamic SBP with theoretical support from the appendix.
> >     - *The discussion spanning pages 4 and 5.* We split the discussion into two parts: **the Bregman divergence** and **Asymptotically log-concave distributions.** For each part, we clarify the intention for introducing new arguments.
> > - **The assumptions underpinning the theoretical results are quite dense and abstract. It would be helpful to explain what they mean in practice.** We have fixed some grammatical errors and eased wordings for the assumptions and provide implication using Fig. 1. As we tried our best to explain our assumptions with helpful figures and plain terminology, there still can be some limitations in our description, as we believe our work deals with abstract concepts and brings us down to practical algorithms. Therefore, we think the practicality of our approach will be eventually evaluated by the correctness of our theoretical practice and experimental evaluation.
> > - **Please make it explicit in the main text that the appendix includes extensive tables and figures related to the experiments, including full results and standard errors (e.g. Tables 9 and 10).** As suggested, we added several pointers to the tables in appendix that has complete statistics of experiments in **Section 6**.
> >
> > ## **Minor comments**
> >
> > All grammatical issues, notational clarifications (e.g. L1, Gibbs parameterization), and small-detail suggestions have been addressed throughout.  We thank Reviewer **QGaD** for these helpful pointers, which have substantially improved readability and rigor.
> >
> > We again thank Reviewer QGaD for constructive feedback and considerate suggestions, providing us a good opportunity to improve our paper. We believe our submission now can contribute to a broader audience in the machine learning community, with a coherent narrative and rich, relevant context. Please contact us during the discussion phase if there are remaining concerns.
> >
> > Sincerely,
> >
> > Authors of Submission #4592
> >
> > [1] Pierre-Cyril Aubin-Frankowski, Anna Korba, and Flavien Léger. Mirror descent with relative smoothness in measure spaces, with application to Sinkhorn and EM. Advances in Neural Information Processing Systems, 35:17263–17275, 2022.
> >
> > [2] Mohammad Reza Karimi, Ya-Ping Hsieh, and Andreas Krause. Sinkhorn flow as mirror flow: a continuous-time framework for generalizing the sinkhorn algorithm. In International Conference on Artificial Intelligence and Statistics, pp. 4186–4194. PMLR, 2024.

---

### Decision · Action_Editor_adND · 2025-08-12

**Recommendation:** Reject

**Audience:**

Yes

**Audience Explanation:**

Schrödinger bridge problems are of interest to the community, and the reviewers agree this paper is appropriate for the TMLR audience.

**Claims And Evidence:**

No

**Claims Explanation:**

This paper proposes a variational online mirror descent framework for solving Schrödinger bridge problems. The reviewers found the proposed framework to be an interesting and promising direction, and appreciated the convergence guarantees established in the paper.

Following the TMLR review criteria, acceptance requires that the claims made in the submission be supported by accurate, convincing, and clear evidence, including the _clarity of the narrative and arguments presented_. Overall, the reviewers agreed that the paper does not yet meet this bar at its current stage.

**Summary of reviewer concerns**

The reviewers identified several major weaknesses:

* **Clarity and narrative:** The manuscript lacks a cohesive narrative. The introduction does not clearly communicate the problem or its significance, and it is not accessible to a general machine learning audience.

* **Contributions:** The contributions are not stated clearly, particularly in relation to prior work. While novelty is not part of the acceptance criteria at TMLR, it should be clearly stated how this paper advances the literature.

* **Accessibility of background:** The background material is presented in a way that is unnecessarily complex and difficult to follow.

* **Theory:** The theoretical results have insufficient discussion of assumptions and practical implications.

* **Experiments:** The experiments are not explicitly connected to the core theoretical contributions. Reviewers also noted that the criteria for highlighting “best” results could be clarified, and that in some cases the reported performance differences may not be statistically significant.



**Response and revision evaluation**

The reviewers appreciated the authors’ efforts in revising the manuscript, including improvements in distinguishing the contributions from prior work, updates to the experimental results, and the expanded background in the appendix.

However, after considering the revision and author response, while some concerns were addressed, the reviewers stated that several issues from the original reviews remain.

* The reviewers remained primarily concerned about the overall clarity of the manuscript, and felt the revision did not address many of their original concerns.
* After reading the response, the reviewers felt the paper was not yet ready for acceptance due to notational issues and other specific issues, such as concerns about Theorem 1 (raised by Reviewer X26F) and a comment that the statement of Lemma 14 is unclear and missing a statement.
* After the revision, the reviewers remain unconvinced by the experimental results and the extent to which they advance current methodologies --- especially in the single-cell dynamics experiment.

**Summary**

Given the reviewer comments and discussion, I do not recommend acceptance at this time. For a future revision, the reviewers suggest placing particular emphasis on strengthening the early framing, presenting the core problem, motivation, and contributions clearly in the introduction, in a way that will be accessible to the broader machine learning community. In addition, increasing accessibility may require reworking and restructuring some parts of the text, rather than simply adding further material. Finally, the reviewers also encourage addressing the specific theoretical and experimental concerns discussed above and in their reviews.

**Resubmission Of Major Revision:**

The authors may consider submitting a major revision at a later time.